# Species composition and forest structure explain the temperature sensitivity patterns of productivity in temperate forests.

Friedrich J. Bohn[1], Felix May[2], and Andreas Huth[1,2,3]

[1]Helmholtz Centre for Environmental Research - UFZ / Permoserstr. 15 / 04318 Leipzig / Germany
[2]German Centre for Integrative Biodiversity Research (iDiv) Halle-Jena-Leipzig / Deutscher Platz 5e / 04103 Leipzig / Germany
[3]University of Osnabrück / Barbarastr. 12 / 49076 Osnabrück / Germany

*Correspondence to:* Friedrich J. Bohn (friedrich.bohn@ufz.de)

**Abstract.** Rising temperatures due to climate change influence the wood production of forests. Observations show that some temperate forests increase their productivity, whereas others reduce their productivity. This study focuses on how species composition and forest structure properties influence the temperature sensitivity of above-ground wood production (AWP). It further investigates which forests will increase their productivity the most with rising temperatures. We described forest
structure by leaf area index, forest height and tree height heterogeneity. Species composition was described by a functional diversity index (Rao's Q) and a species distribution index ($\Omega_{AWP}$). $\Omega_{AWP}$ quantifid how well species are distributed over the different forest layers regarding AWP. We analysed 370,170 forest stands, generated with a forest gap model. These forest stands covered a wide range of possible forest types. For each stand we estimated annual above-ground wood production and performed a climate sensitivity analysis based on 320 different climate time series (of one year length). The scenarios differed
in mean annual temperature and annual temperature amplitude. Temperature sensitivity of wood production was quantified as the relative change in productivity resulting from a $1°C$ rise in mean annual temperature or annual temperature amplitude. Increasing $\Omega_{AWP}$ positivly influenced both temperature sensitivity indices of forest, whereas forest height showed a bell-shaped relationship with both indices. Further, we found forests in each successional stage that are positively affected by temperature rise. For such forests, large $\Omega_{AWP}$-values were important. In case of young forest, low functional diversity and small tree
height heterogeneity was associated with a positive effect of temperature on wood production. During later successional stages, higher species diversity and larger tree height heterogeneity was an advantage. To achieve such a development, one could plant below the closed canopy of even-aged, pioneer trees a climax-species-rich understory that will build the canopy of the mature forest. This study highlights that forest structure and species composition are both relevant for understanding the temperature sensitivity of wood production.

## 1 Introduction

Climate change alters wood production by modifying the rates of photosynthesis and respiration rates of trees (Barber et al., 2000; Luo, 2007; Peñuelas and Filella, 2009; Reyer et al., 2014). Changes in forest productivity have been observed in past decades all over the world (Nemani et al., 2003; Boisvenue and Running, 2006; Seddon et al., 2016). The carbon stock of forests and their role as carbon sinks are therefore changing. These findings have stimulated discussions about whether forest management strategies can be adapted to reduce forest vulnerability to climate change, to support recovery after extreme events and foster the carbon sink function of forests (Spittlehouse and Stewart, 2004; Spittlehouse, 2005; Bonan, 2008).

Wood production is influenced by several factors, such as $CO_2$ fertilization, nitrogen deposition, precipitation, and temperature. (Barford et al., 2001). For instance, rising $CO_2$ increases wateruse efficiency of forests (Keenan et al., 2013), which could compensate negative effects of climate change on European forest growth (Reyer et al., 2014). Another important process is fertilization (De Vries et al., 2006, 2009). Due to depositions of nitrogen in the second half of the last century, wood production had increased in European forests (Solberg et al., 2009). However, temperature modifies photosynthesis, respiration and growth rates of trees (Dillon et al., 2010; Piao et al., 2010; Wang et al., 2011; Jeong et al., 2011; Heskel et al., 2016). In the temperate biome, positive effects on wood production (Bontemps et al., 2010; Delpierre et al., 2009; Pan et al., 2013; McMahon et al., 2010, e.g.) as well as negative ones have been found (Barber et al., 2000; Jump et al., 2006; Charru et al., 2010, e.g.). However, it remains unclear why forests react differently to temperature change.

In addition to the influence of climate variables, wood production is also affected by internal forest properties. These properties can be grouped into two types: properties which describe forest structure, and those which describe species composition (Fig. 1). For instance, changes in productivity can result from changes in basal area (Vilà et al., 2013), in leaf area index (Asner et al., 2003) or in the heterogeneity of tree heights within a forest (Bohn and Huth, 2017). Furthermore, wood production often increases with the increasing number of species (Zhang et al., 2012; Vilà et al., 2007).

Forest stands, which differ in their forest properties, might respond differently to the same climate change (Huete, 2016). For instance, the positive effect of increasing temperature on wood production fades with forest age in temperate deciduous forest (McMahon et al., 2010; Bontemps et al., 2010, e.g.) and Morin et al. (2014) showed that higher diversity buffers the effect of inter-annual variability on wood production. However, these studies include only a few forest properties and rarely include properties related to both species composition and forest structure. Hence, it is unclear, how these forest properties influence wood production change due to temperature rise and which forests will benefit from rising temperatures.

As far as we know, there is no data set available that covers forests, differing in structure and diversity, under almost identical climatic conditions. Even if a larger number of forest stands were available, it would be difficult to manipulate for instance temperature while keeping all other climate variables constant. Forest simulations models offer an alternative to the analysis of field experiments. Such models are able to estimate wood production under different climate conditions (Lasch et al., 2005; Bohn et al., 2014, e.g.). For instance, Reyer et al. (2014) investigated the effect of climatic change on forests by simulating 30-years time slices of a range of different future climates for 135 inventoried forest stands. There are also model-based studies, which systematically analysed the effect of species diversity on productivity and stability over long periods (Morin et al.,

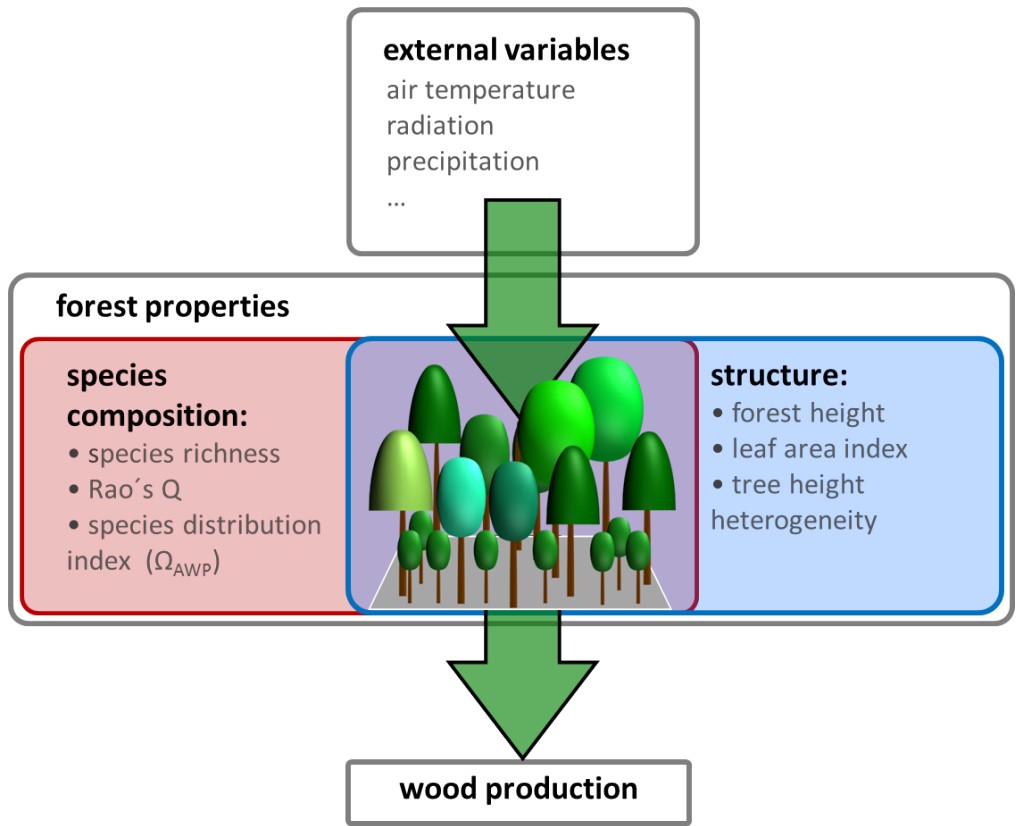

**Figure 1.** Overview of drivers influencing wood production. External variables in this study are temperature, radiation, and precipitation. Forest properties are divided into two groups: species composition properties (e.g., Rao's Q as a measure of functional diversity and species distribution index $\Omega_{AWP}$) and forest structure properties (e.g., forest height, leaf area index and tree height heterogeneity).

2011, 2014). However, disturbed or managed forest stands and the influence of climate change have not been included in these analyses.

In this study we therefore propose a new simulation-based approach. First, we generate a large number of forest stands covering various forest structures and species compositions (for up to eight temperate tree species). Annual above-ground wood production (AWP) is then calculated for all forest stands based on climate time series. These time series differ in the mean annual temperature and the intra-annual temperature amplitude. We aim to analyze (i) how productivity of forest stands (AWP) is influenced by increasing mean annual temperature and (ii) by increasing intra-annual temperature amplitude? Furthermore, we address the question (iii) of which forest stands will benefit most from rising temperatures.

## 2 Method

To analyse the effect of temperature on the productivity of forest stands, we applied the "forest factory" model approach (Bohn and Huth, 2017). The forest factory generated 370,170 different forest stands (see section 2.1) and allowed the estimation of aboveground wood production (AWP) under various climate time series (see section 2.2). The 320 scenarios differed in mean annual temperature and annual temperature amplitude. Finally, we calculated the forest stand-specific sensitivity of productivity to temperature change as the relative change of wood production per temperature change of 1 °C (see section 2.2). To relate these sensitivities to forest structure and species composition, we characterised every forest stand with five properties (see section 2.4). We analysed the influence of the five forest properties on temperature sensitivity using boosted regression trees (see section 2.5). Finally, we analysed which combination of forest properties resulted in the highest sensitivity values for different successional stages (see section 2.6).

### 2.1 The forest factory approach

The forest factory creates forest patches based on different stem size distributions and species mixtures. We used 15 stem size distributions covering a gradient from young to old and disturbed to undisturbed forests. Species mixtures included all 256 possible combinations of *Pinus sylvestris*, *Picea abies*, *Fagus sylvatica*, *Quercus robur*, *Fraxinus excelsior*, *Populus x canadensis*, *Betula pendula* and *Robinia pseudostuga*. We used the species parameter set and algorithms of the FORMIND model version for temperate forests within the forest factory (Bohn et al., 2014; Fischer et al., 2016). 100 forests patches of each combination were built.

To generate forest patches, the forest facotry randomly choose trees from the stem size distribution, assigned a species identity and plant them within a patch of 400 $m^2$ size. To place a tree within a patch the following rules must be met: (i) there must be enough space available for crowns of every tree and (ii) every tree in the forest must have a positive productivity under its environmental conditions (light, temperature, water).

We used climate time series from the year 2007, measured at Hainich National Park, central Germany. We assumed this time series to be a typical example for a temperate year (in principle it possible to use climate data from any other location). In contrast to an artificially generated climate, this climate is perfectly physically consistent (regarding light, air temperature and precipitation).

In a few cases not all species of the mixture could be placed within a patch by the algorithm, so we rejected such forests. We ended up with 370,100 forest stands. For more details regarding the forest factory see Bohn and Huth (2017).

### 2.2 Wood production

The calculation of above-ground wood production (AWP) of trees was based on algorithms of the model FORMIND (Bohn et al., 2014; Fischer et al., 2016). In this model, the wood production of a single tree is calculated as the difference between climate variables driven respiration rates and photosynthesis. The photosynthesis rate ($P_{tree}$) results from the crown size, self-shading within the crown and available light at the top of the tree. The available light depends on the radiation above the canopy,

reduced by the shading of larger trees within the forest stand. Furthermore, productivity can be limited due to air temperature and available soil water, which is expressed by the photosynthesis-limiting factor $\phi$ for each tree (Gutiérrez, 2010; Fischer, 2013; Bohn et al., 2014). Available soil water within the stand results from precipitation, interception, evapotranspiration of trees and run-off.

One part of the photosynthesis production of a tree ($P_{tree}$) is allocated to its maintenance respiration (and to non-wood tissues; $R_m$). Maintenance respiration depends on tree biomass and temperature $\psi$ (Piao et al., 2010). The remaining organic carbon is transformed into newly grown above-ground wood ($AWP_{tree}$) and a proportional growth respiration ($r_g$).

$$AWP_{tree} = (\phi P_{tree} - \psi R_m)(1 - r_g) \tag{1}$$

$AWP_{tree}$ was summed over all trees to obtain the productivity of the modeled forest stand - AWP (for a more detailed
description of growth processes, see Bohn et al. (2014); Bohn and Huth (2017) ).

## 2.3   Climate sensitivity

To generate a set of 320 annual climate time series, we selected daily climate measurements of the Hainich station in central Germany between the years 2000 and 2004. This time series includes mean daily radiation, precipitation and air temperature (see Appendix A1; Fig. A1)). We separated these time series into five distinct time series of one-year length. First, we increased
or decreased the mean annual temperature of each year by adding or subtracting 0.5 °C steps between -1.5 °C and +2 °C. Second, we changed the amplitude of the annual temperature cycle for these time series variation of each year. To do so, we modified the standard deviation of each year by 4% steps between -12 % and +16 %. We ended up with five sets of climate times series (of one-year length) that differ in temperature, precipitation and radiation. Each of these five sets includes 64 time series, which differ only in temperature (see Appendix A1 , Fig. A2). Temperature change was quantified using two indices:
(i) mean annual temperature and (ii) annual temperature amplitude, which described the 95 % inter-quantile range of all daily temperature values of a given year. We did not model the effects of nitrogen and $CO_2$ fertilization (as both do not vary strongly within one year) or extreme anomalies (e.g., pathogen attacks) on wood production. Figure 2 (a-c) shows the above-ground wood production (AWP) for different annual temperatures for three different forest stands.

We analysed the sensitivity of every forest stand to temperature change following the approach of Piao et al. (2010). For
every forest stand, a general linear model was fitted relating wood production mean annual temperature (MAT) and intra-annual temperature amplitude (Q95), as well as the nuisance parameter year.

$$AWP = \alpha x_{MAT} + \beta x_{Q95} + \gamma x_{year} + \epsilon \tag{2}$$

For every forest, we calculated the relative change of productivity resulting from an increase of 1 °C:

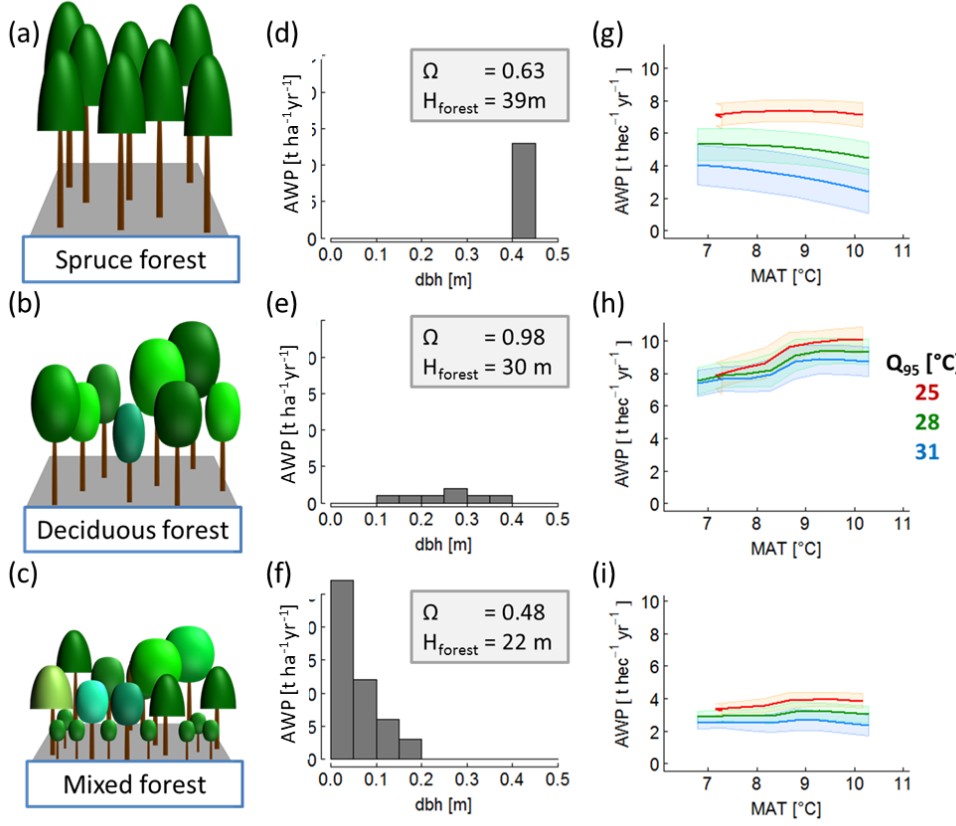

**Figure 2.** Overview of forest properties and resulting temperature sensitivity of above-ground wood production (AWP) of three exemplary forests: a) old even-aged spruce forest; b) mature deciduous forest; c) a quite young mixed species forest. The middle panel (subfigures d, e & f) shows the corresponding stem size distributions and provides information on the highest tree in the forest ($H_{forest}$) and species distribution index $\Omega_{AWP}$ (which quantifies the suitability of a species distributed within the forest structure regarding AWP). Each forest is treated with 320 climate time series: The last panel shows the AWP as a a function of mean annual temperature (MAT). The colours indicate different inter-annual temperature amplitudes (Q95) of the used time series. (The coloured bands show the standard deviation due to the variability of the five different time series that exist for each combination of mean annual temperaturemean and intra-annual temperature amplitude).

$$SI_{MAT} = \frac{\alpha}{AWP} \qquad (3)$$

$$SI_{Q95} = \frac{\beta}{AWP} \qquad (4)$$

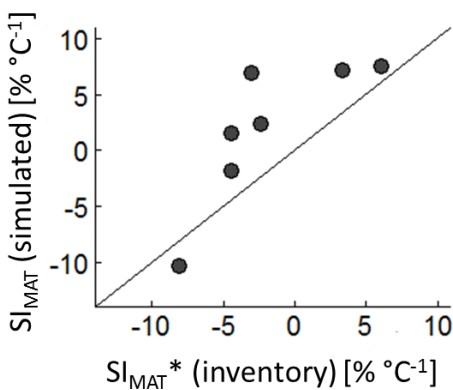

**Figure 3.** $SI_{MAT}$ values of seven different forest types derived from the analysis of the German forest inventory vs. $SI_{MAT}$ values derived from corresponding forest types of the forest factory. Only those $SI_{MAT}$ values of the field data are analysed, which showed p-values smaller then 0.05.

In our analysis we excluded all forests stands for which AWP turns negative if the temperature rises by 1 °C (This occurs in 2 % of all stands).

We also determined the sensitivity of forests to temperature change using the German forest inventory to validate our results. However, the inventory does not include leaf area index (LAI) measurements. We therefore assumed the basal area as a proxy for LAI, and we selected subsamples of forests stands with similar structure (basal area, tree height heterogeneity, forest height, and same species mixtures). In addition, we used elevation as a proxy for mean annual temperature, assuming temperature changes of 0.65 °C per 100 metres on average (Foken and Nappo, 2008). Only in the case of spruce and beech monocultures did we find enough data to calculate $SI_{MAT}$ values for several forest structures (for more details see Appendix A3, Fig. A3).

The comparison between the $SI_{MAT}$-estimation based on the German forest inventory with $SI_{MAT}$ values of corresponding forests from the forest factory showed quite good agreement ($R^2$ = 0.65). However, the simulated $SI_{MAT}$ values of the forest factory slightly overestimated the sensitivity compared to the inventory-based values (Fig. 3). This might be explained by the difference in the methods used because, in case of the inventory, we used basal area instead of LAI and altitude instead of temperature. Another explanation could be that in our approach the climate time series showed relatively high and regular precipitation. In the German forest inventory warmer sites might be more frequently exposed to water stress, which than reduced the SI values.

### 2.4 Five forest properties to describe forest stands

We used three indices to describe the forest structure: leaf area index (LAI), maximum forest height ($H_{forest}$), which corresponds to the height of the largest tree in a forest stand, and tree height heterogeneity ($\theta$), which was quantified by the standard

deviation of the tree heights. To describe species composition, we used Rao's Q and species distribution index ($\Omega_{AWP}$). Rao's Q quantified functional diversity based on species abundances and differences in species traits (Botta-Dukát, 2005, for details see Appendix A2). $\Omega_{AWP}$ analysed the optimal location of species within the forest structure. $\Omega_{AWP}$ is defined as the ratio of the forest's productivity to the maximum possible productivity of the forest without changing tree sizes or number (Bohn and Huth, 2017). Hence, the maximum productivity can be obtained by varying only the species identities of trees in the forest stand. We changed the assigned species of each tree until we found the optimal species for each individual tree and its specific environmental condition. All five indices were nearly uncorrelated for the investigated forest stands ( Appendix A2 Table A1).

## 2.5 Boosted regression trees

We applied boosted regression trees to quantify the influence of the five forest properties on $SI_{MAT}$ and $SI_{Q95}$. Boosted regression trees are a machine learning algorithm using multiple decision (or regression) trees. It is able to address unidentified distributions (De´Ath, 2007; Elith et al., 2008). Each model was fitted in a forward stage-wise procedure to predict the response of the dependent variable on ($SI_{MAT}$ or $SI_{Q95}$) to multiple predictors tree height heterogeneity, forest height, LAI, Rao's Q, and $\Omega_{AWP}$). To omit an over-fitting regarding maximal forest height, we classified forest stands into 18 classes. Each class had a width of 2 metres, starting with 4 to 6 metres and finishing with 36 to 38 metres. The boosted regression trees tried an iterative process to minimise the squared error between predicted SI values and those of the data set. Hereby, part of the data was used for a fitting procedure and the other part was used for computing out-of-sample estimates of the loss function (Ridgeway, 2015). This boosted regression tree analysis was performed in the R-package gbm 2.1.1 (Ridgeway, 2015).

We used a quarter of the data (randomly sampled) for the machine learning procedure. To get the best model, we varied the following four parameters of the boosted regression tree algorithm: learning rate (0.1, 0.05 and 0.01), the bag-fractions (0.33, 0.5 and 0.66), the interactions depth (1, 3 and 5) and the cross-validation (3-, 6- and 9-fold) assuming a Gaussian error structure (the default setting). The best fitted boosted regression tree for both $SI_{MAT}$ and $SI_{Q95}$ showed a learning rate of 0.1, a bag-fraction of 0.66, an interaction depth of 5 and a 3-fold cross validation. These two models were used for all further analyses. The remaining 75% of the data were used to validate the fitted boosted regression tree algorithm.

## 2.6 Finding the forest stands for different successional stages that benefit the most increasing temperatures

Here, we assumed forest height as a proxy for the successional stage of a forest. In every height class, we selected those 5% of forests that showed the highest sensitivity values ($SI_{MAT}$ and $SI_{Q95}$). We removed the forest height classes between 10 and 14 metres, as they only contained a few forests (15). For all other classes, we analysed the relationship between height class and the forest properties ($\Omega_{AWP}$, Rao's Q, LAI and tree height heterogeneity).

## 3 Result

We analysed the sensitivity of productivity (AWP) to temperature for forest stands that differ in forest properties (species distribution index ($\Omega_{AWP}$), functional diversity (Rao's Q), tree height heterogeneity ($\theta$), forest height class and LAI). The

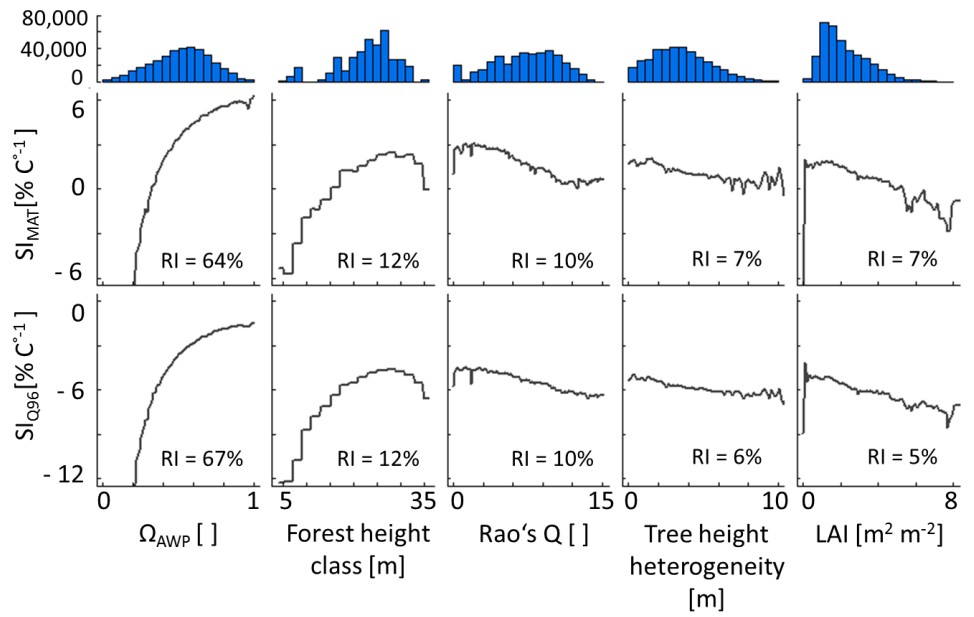

**Figure 4.** Partial dependency plots of the five forest properties $\Omega_{AWP}$ (species distribution index), forest height class, Rao's Q (functional diversity), tree height heterogeneity and LAI (leaf area index) for $SI_{MAT}$ (sensitivity to changes inthe mean annual temperature) and $SI_{Q95}$ (sensitivity to changes in annual temperature amplitude). Relative importance (RI) compares the influence of different input variables on the variability of a target variable. Histograms show the frequency of forest property values in the analysed data set. Note, a $\Omega_{AWP}$ is the ratio of the current AWP of a forest and the highest possible AWP optained by shuffling ony species identies without changing the forest structure.

annual above-ground wood production (AWP) was estimated for each forest stand using 320 different climate time series. We then quantified the changes in productivity resulting from changes in mean annual temperature ($SI_{MAT}$) and intra-annual amplitude ($SI_{Q95}$). For the analysed forest stands, the average $SI_{MAT}$ is 1.5 % °C $^{-1}$ and the average $SI_{Q95}$ is -5.4 % °C $^{-1}$ (see also the frequency distribution in Appendix B1, Fig. B1).

5  With a boosted regression tree algorithm, we analysed how the five forest properties influence the temperature sensitivity of forests. To validated the fitted boosted regression tree algorithm, we compared SI values, which are not used for the fitting, with the SI value predicted by the boosted regression tree algorithm (Fig. 4). The sensitivities to mean annual temperature change ($SI_{MAT}$) correlated very well ($R^2$ of 0.84) and showd a low RMSE of $\pm$ 2.9 % °C $^{-1}$ (see Appendix B2 Fig. B3).The RMSE even decreased to $\pm$ 1.5 % °C $^{-1}$ if a subset of the forest stands was analysed that showed $SI_{MAT}$ values larger than -5 % °C

10  $^{-1}$ (90 % of the data). The accuracy of the sensitivities to temperature amplitude change ($SI_{Q95}$) was even slightly better. In addition, a subset that included $SI_{Q95}$ values larger than -15 % °C $^{-1}$ (93% of the data) showed a RMSE of only $\pm$ 1.1 % °C $^{-1}$ (see Appendix B2 Fig. B4).

According to boosted regression tree analysis, $\Omega_{AWP}$ was the most relevant forest property to explain temperature sensitivities (relative influence of 87 % for $SI_{MAT}$ and 89 % for $SI_{Q95}$; see also Appendix B2, Fig. B2). However, the influence of

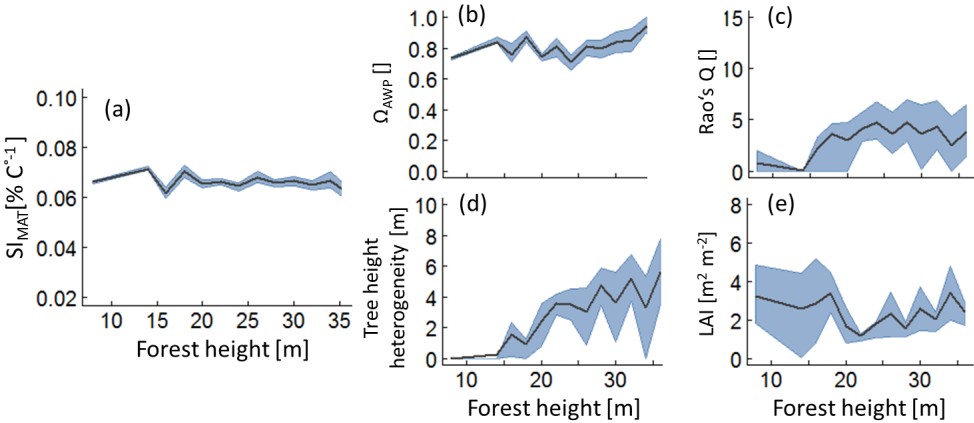

**Figure 5.** Analysis of those forests that show the highest 5 % of the SI values depending on forest height. Lines indicate mean values of the forest subsamples which includes the best 5% regarding $SI_{MAT}$ of each hight class. The grey band indicates the inter quartile range. Figure a) shows temperature sensitivity of above-ground wood production over forest height, analysing only the best the forest subsample. b) to d) shows the change of the remaining forest properties within the forest subsamples ($\Omega_{AWP}$ = optimal species distribution; LAI = leaf area index; Rao s Q quantifies functional diversity).

$\Omega_{AWP}$ on temperature sensitivity flattened out for high $\Omega_{AWP}$ levels (Fig. 5). The second relevant forest property was forest height ($H_{forest}$). Forests with heights between 25 and 30 m benefited the most from increasing mean annual temperatures. The other three properties (LAI, Rao's Q, and tree height heterogeneity) had a low influence on $SI_{MAT}$.

Both sensitivity indices showed similar relationships to the five forest properties. However, an increase in annual temperature amplitude always reduced productivity, whereas increasing mean annual temperature could result in a positive effect on wood production. To detect those stands that benefit the most from increasing temperature, we selected the 5 % of forest stands that showed the highest $SI_{MAT}$ values in each forest height class (Fig. 5). In all forests classes, we found forest stands that would benefit from increasing temperatures. Analyses of their forest properties revealed that the $\Omega_{AWP}$ levels were always high. Young forests (low forest height), which had a positive temperature sensitivity, show low functional diversity and low tree height heterogeneity ($\theta$). For older forests (of intermediate and high forest height) with positive temperature sensitivity, we found an intermediate level of functional diversity. Interestingly, for three variables (Rao's Q, tree height heterogeneity and LAI), the relationships changed their character between young and intermediate forest heights. We obtained similar simulation patterns for $SI_{Q95}$ ( Appendix B3 Fig. B5).

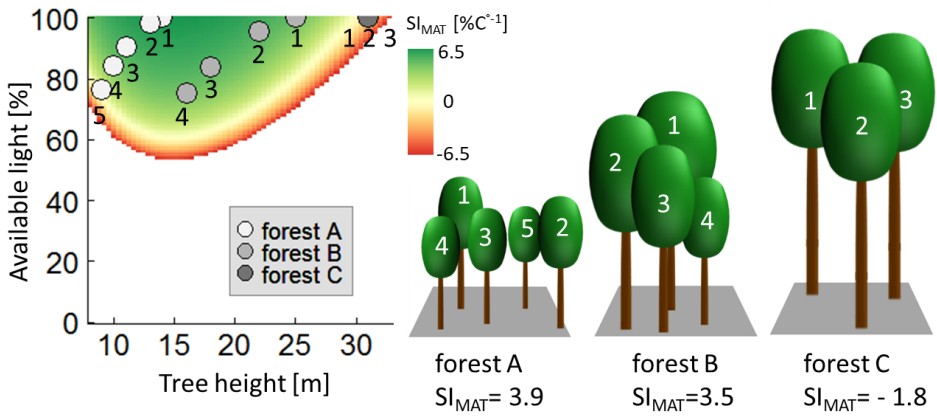

**Figure 6.** Analysis of the sensitivity index of AWP against mean annual temperature ($SI_{MAT}$) values of single trees within three different forests. The diagram shows the calculated $SI_{MAT}$ value of individual trees for every combination of tree height and available light (for *pinus sylvestris* between $SI_{MAT}$-levels of 6.5 and - 6.5; other species show similar patterns). The dots indicate the different trees of the three forest examples. The white dots belong to trees with the corresponding number of forest A, grey dots belong to the trees of forest B and dark grey dots belong to forest C. Note that in the case of forest C, all trees have the same height and the same light, so that all three dots are at the same place in the diagram.

## 3.1 Understanding the patterns

### 3.1.1 The influence of forest structure on temperature sensitivity

Forest structure affects the wood production of single trees in two ways. First, it determines the amount of light available to each individual tree and second, the size of trees influences their photosynthesis and respiration rates (Fig. B6). Hence, based
on the height of a tree and the amount of light available to it, it was possible to calculate its SI values (for a detailed discussion of these calculations, see Appendix B4).

In even-aged forests, all trees have the same height and receive full light (e.g., Fig. 6, forest C). In our study, such forests showed a bell-shaped relationship between forest height and temperature sensitivity (Fig. 6, SI values for 100 % available light depending on tree height).

In case of a forest consisting of trees of different heights smaller trees receive less light due to shading. Note that, even if trees receive less light, the bell-shaped relationship between tree height and productivity persisted (Fig. 6). Two cases will be discussed (assuming identical LAI as forest C, Fig. 6). In the first case all trees have not yet reached their maximal SI values (Fig. 6, forest A,); and in the second case all trees have already passed their maximal SI values (Fig. 6, forest B). In the case of forest A, trees in the shade of larger trees always had lower SI values if they belong to the same species (see Appendix B4).

Hence, the temperature sensitivity level of this forest was lower than the sensitivity of an even-aged forest, whose trees have

the same size as the largest tree in forest A (Fig. 6, tree 1). Hence, if maximal SI values were not reached, increasing height heterogeneity decreases SI values of a forest.

In forest B (Fig. 6), SI values of the shaded trees can be similar (or even higher) than the SI value of the largest trees in the forest (SI values of tree 1 show similar levels to tree 2, 3 and 4 in forest B, Fig. 6). Hence, if maximal SI values were 5 passed, increasing tree height heterogeneity resulted in similar (or even more positive) temperature sensitivity levels compared to even-aged forest trees (an even-aged forest consisting only of trees similar to tree 1 of forest B in Fig. 6). These general considerations explain the change from low levels of height heterogeneity in young forests to a more heterogeneous structure in the analysis of those forests, which will benefit from increasing temperature (see Fig. 5 d).

### 3.1.2 The effect of species composition on temperature sensitivity

10 In this study, we use the new index $\Omega_{AWP}$ called the species distribution index (Bohn and Huth, 2017). $\Omega_{AWP}$ is the ratio between current AWP and the highest possible AWP of the forest which can be reached due to shuffling of species identities. Its huge importance on forest temperature sensitivity might be illustrated by the following considerations: If species are un-favourably distributed within the forest (low $\Omega_{AWP}$), the AWP of the forest is low and, in consequence, the SI values are low as well (see Appendix). If the AWP is low the forest will suffer from increasing temperatures, which results in negative slopes 15 (Equation 2). These values are then divided by low AWP values (Equation 4), which results in large negative values of $SI_{MAT}$ and $SI_{Q95}$. (See Appendix B5).

Increasing functional diversity (Rao's Q) stabilised the forests' sensitivity to temperature. This corresponds to results of Morin et al. (2014) and the theoretical consideration of Yachi and Loreau (1999). The analysis of the single species can give additional insight into the mechanisms behind those species that benefited the most from temperature increase, which were 20 deciduous trees under most conditions. This is reasonable as warmer regions host more deciduous species than needle-leaf species. The highest functional diversity (Rao's Q), one the other hand, occured in mixtures of deciduous and needle-leaf trees (Appendix B5 Fig. B7). As only two needle-leaf species were considered here in the species pool, low Rao's Q values were dominated by mixtures of deciduous trees. Such deciduous tree mixtures mostly benefited from temperature increases. In consequence, mixtures with high Rao's Q values, which mostly included both functional types, reacted more poorly (Fig. 4; 25 Appendix B5 Fig. B7).

We developed two diagrams that show the species with the highest temperature sensitivity and with the highest productivity for different conditions (available light and height of a tree) (Fig. 7). Interestingly, the species with the highest productivity differed from the species that benefit most from rising temperatures in many cases. This has important implications. The highest benefit due to increasing temperatures was optained by forests with high but not maximal $\Omega_{AWP}$ (Fig. 5). Additionally, 30 deciduous trees benefited more than coniferous trees from rising temperatures (Fig. 7, Appendix B5, Fig. B7). Hence, young forests should consist of deciduous trees (compare Fig. 6, forest A, and Fig. 7), although the highest productivity values are found for coniferous trees (Fig. 7; forest A). Forests including large trees obtained the highest sensitivity values if intermediate sized trees differed in their species identity from the largest trees (Fig. 7).

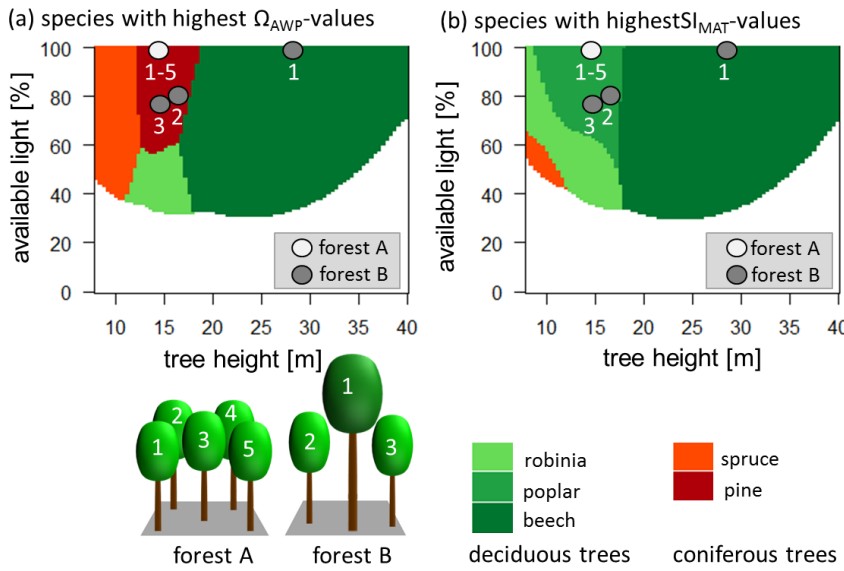

**Figure 7.** Graphic (a) shows which species have the highest productivity ( $\Omega_{AWP}$ value of 1) under the current climate for different heights and different light conditions. Graphic (b) shows which species shows the highest increase in productivity due to rising temperatures for different heights und different light conditions. Red colours indicate coniferous trees, whereas green colours indicate deciduous trees. Darker colours indicate late successional species, whereas lighter colours indicate pioneers. The dots indicate the different trees of the two forest examples (A and B). The white dots belong to trees with the corresponding number of forest A. Note, that all trees have the same height and the same light, so all five dots are at the same place in the diagram. Grey dots belong to the corresponding trees with the same number of forest B.

## 4 Discussion

### 4.1 The study design

In this theoretical study, we present a new climate sensitivity analysis (regarding temperature) of AWP. This approach extends field observations and long-term model simulations, as it allows the analysis of exsisting forests, but also of those that might

5   exist in the future due to management changes and/or disturbances. Our approach includes only forest stands in which every tree has positive productivity and enough space for its crown. Hence, it is impossible, for instance, that light-demanding species grow below a closed canopy or forests are overcrowded. However, the data set also include a few very unusual stands structures or species combinations, which can not emerge in a natural system, but may result from disturbances or management. In the case of field observations, it is difficult to explore the influence of a single climate variable (e.g., temperature) on one target

10   variable (e.g., AWP), as in most cases, several variables are altered at the same time (see also Appendix A3). Process-based models are one option to analyse such relationships and separate these effects. The simulation of AWP with the FORMIND-

model in temperate forests has been successfully compared to Eddy flux sites (Rödig et al., 2017b), the national German forest inventory (Bohn and Huth, 2017), and European yield tables Bohn et al. (2014).

An advantage of the forest factory approach is the huge set of various forests stands that can be analysed. The dataset includes forest stands that often occur in temperate forests (even-aged spruce, pine and beech stands). However, it also includes hypothetical ones that could occur through alternative forest management or disturbances (fire, bark beetles, etc.). Hence, our data set of forest stands covers a much larger variety of forest property combinations compared to long-term forest simulations with the focus on natural forests in their equilibrium state (Morin et al., 2011, e.g.) or on monocultures (Reyer et al., 2014, e.g.). Long term simulations with ecosystem models, which process modelled climate projections, face a trade-off between cascade uncertainty and path dependency (Wilby and Dessai, 2010; Reyer et al., 2014). The accumulations of model uncertainties over such a process chain result in increasing uncertainty. Our study design tries to minimise this uncertainty and omit path dependencies by including only those processes that might be relevant for the research question. In this study, for instance, we omit the effect of climate change on regeneration and mortality. Furthermore, using several climate variables as model inputs but only analysing the effect of one variable might lead to incorrect interpretations of its effect. For example, temperature and radiation often correlate, and both might increase productivity. Therefore, in this study, we only vary one variable in all 5 sets of time series. This guarantees that there are no relationships between the target climate variable and the remaining climate variables.

As an increase in global mean temperature of $1.5\,°C$ to $2\,°C$ can hardly be avoided, even under the RCP 2.6 climate scenarios (IPCC, 2013), this study focuses on temperature change. This RCP scenario predicts only small changes in annual precipitation levels for temperate regions. Hence, our approach focuses only on the effect of temperature change on wood production. However, this might be critical for the analysis of strong temperature changes (e.g. RCP 8.5) which will result in an increased incidence of drought and changes in the annual temperature cycles and a strong change in $CO_2$. Such more complex scenarios should be analysed in future studies. Further, we neglect the effect of time lags (e.g. bud building in the previous year). However, it is possible to extend the used time series to analyse the behaviour of the forest over longer time periods and study not only productivity, but also effects on regeneration or mortality.

To characterise the annual temperature cycles we used two variables: mean annual temperature and intra-annual temperature amplitude. Both variables can be varied independently. In case of higher mean annual temperature we observe an elongation of the vegetation period. This leads to higher forest productivity (if other resources are not limiting (Luo, 2007) and explains why $SI_{MAT}$ is often positive. However, warmer summer temperatures can also lead to a decline in wood production due to an increase in respiration. In case of increasing intra-annual temperature amplitude, more days with extreme temperatures will occur in a year. Thus, an increase of $1\,°C^{-1}$ of intra-annual temperature amplitude will increase respiration more strongly compared to an increase of $1\,°C^{-1}$ of mean annual temperature. Hence, the increase of intra-annual temperature amplitude normally has negative effects on the productivity (negative SI values).

The temperature sensitivity values obtained here are in the same range as those found for temperate ecosystems in heating experiments (Lu et al., 2013, $4.4 \pm 2.2\,\%\,°C^{-1}$). Within the 16 analysed studies reviewed by Lu et al. (2013), the experimental plots show almost identical environmental conditions (soil, radiation, and precipitation) and species composition. To heat the

plots, greenhouses or infrared heaters were used. Another study, based on natural forest stands in New Zealand, found an AWP increase of between 5 and 20 % °C$^{-1}$ for forest, assuming no change in forest structure and species composition (Coomes et al., 2014). The analysed plots were spread throughout New Zealand, and warmer temperatures coincide with higher radiation (Mackintosh, 2016). Hence, the analysed temperature effect also includes the influence of radiation. In our setting, however,

the influence of temperature is independent of radiation (Lu et al., 2013, as in). We also found a good correlation between SI values derived from growth measurements of the German forest inventory and simulated SI values based on the forest factory ( Appendix A3 Fig. A3 & 3 ).

## 4.2 Implications for forest management

Our findings might be relevant for future management strategies for temperate forests. Specifically, our new understanding of

10 which species benefit most from rising temperatures (Fig 6) suggests possible strategies, e.g. replacing spruce monocultures with mixtures of deciduous trees. Further, based on the analysis of which forest structure benefits most from rising temperatures (Fig. 4, Fig 5, Fig 6), early-stage even-aged forests should include mainly pioneer species. In the mature stage, we predict a positive effect of temperatures on wood production for a mixture of climax species including different tree sizes. These climax species could be planted below the canopy of the pioneer species in young forests. In our approach, we do not simulate the

15 establishment of very young trees. However, during the conversion between these two forest types one big challenge might be the removal of the pioneer trees without damaging the young trees that will build the mature forest.

## 4.3 Implications for global vegetation modelling

Most global vegetation models represent vegetation as fractional cover of different plant functional types within a grid cell (e.g. LPJ Sitch et al., 2003). Only a few global vegetation models include a more detailed representation of vegetation structure

and functional diversity (Sato et al., 2007; Scheiter et al., 2013; Sakschewski et al., 2016). It would be interesting to perform the analysis presented here with global vegetation models which include structure to better understand the mechanisms driving forest systems' sensitivity to climate change.

Besides the global vegetation models, forest gap models, which have been restricted to local stands, are now able to simulate forest dynamics in regions or even entire continents (Seidl and Lexer, 2013; Rödig et al., 2017a). Studies using global veg-

25 etation models or large-scale forest gap models simulate natural succession. Our analysis indicates that natural and managed (or disturbed) forest systems, which differ in forest structure, might react differently to climate change. Hence, we suggest considering forest structure in future analyses of global vegetation. Such information on forest structure might be derived from remote sensing.

## 5 Conclusions

The temperature sensitivity of wood production in temperate forests is influenced by forest structure and species diversity as our study showed. The species distribution index ($\Omega_{AWP}$) and forest height seems to be the most important forest properties influencing temperature sensitivity.

5     Temperate forests that benefit most from temperature rise are those which consist of even-aged deciduous pioneer species in the case of young forests; mature forests benefit most if tree height heterogeneity is large and the forest includes different deciduous climax species.

This study also attemps to explain why certain forests types will decrease their productivity and others not. Our findings highlight the importance of forest structure for future studies investigating wood production under climate change.

10   *Data availability.* The R-workspace which includes the dataset of the analysed forests "foreststands" and the calculated SI values "SIValues" can be found in the online supplement.

## Appendix A:  Additional information regarding methods and validation

### A1    Climate data

The construction of the 320 climate time series is based on measured climate time series of the eddy-flux station Hainich in Central Germany (Knohl et al., 2003) for the years 2000-2004 (Fig. A1). Mean annual temperature of these five years does not correlate with the annual precipitation sum, nor with the mean annual radiation (Fig. A2). Radiation and precipitation within these years correlate quite well (Pearson's r = 0.73).

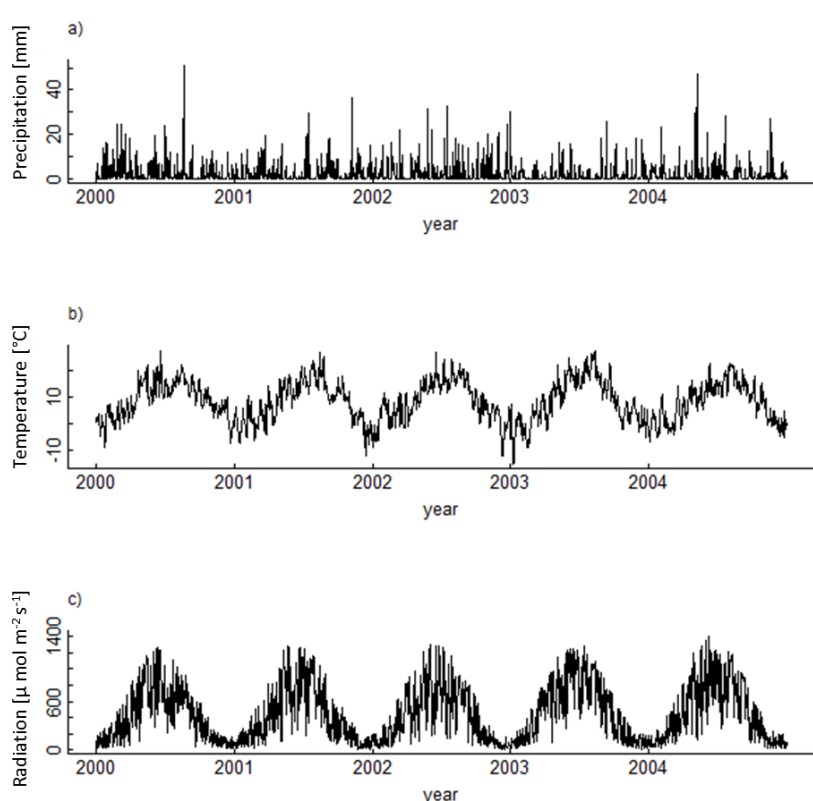

**Figure A1.** The climate time series measured at FLUXNET station Hainich from 2000 to 2004 which are used to generate the 320 climate time series: (a) daily precipitation [mm], (b) daily air temperature [% °C $^{-1}$], (c) daily incoming radiation [photoactive photon flux density $\mu\ mol\ m^{-1}s^{-1}$].

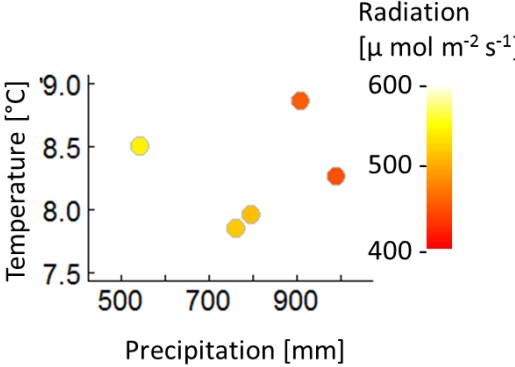

**Figure A2.** Mean annual temperature, annual precipitation sum and mean annual radiation of the five climate time series measured at Hainich station from 2000 to 2004.

## A2   Forest properties

We use three forest properties to describe forest structure (tree height heterogeneity (*theta*), forest height class and LAI) and two properties to describe species diversity (Rao's Q describes functional diversity and $\Omega_{AWP}$ describes suitability). The calculation of Rao's Q is based on 12 species-specific parameters which are relevant for productivity (AWP) and species abundance (based on crown area). None of the properties correlate (table A1).

**Table A1.** Coefficient of Determination ($R^2$) between all used internal forest properties for 370,170 stands of the forest factory. $\theta$ = tree height heterogeneity; LAI = leaf area index; $\Omega_{AWP}$ = species distribution index

| Variables | Rao's Q | $\theta$ | forest height class | LAI |
|---|---|---|---|---|
| $\Omega_{AWP}$ | 0 | 0.02 | 0 | 0.2 |
| LAI | 0 | 0.23 | 0.06 | |
| forest height class | 0.01 | 0.2 | | |
| $\theta$ | 0.02 | | | |

## A3     Validation with the German forest inventory

We analysed the influence of forest structure on temperature sensitivity within the German forest inventory (beech monocultures and spruce monocultures. Tree height was used to calculate forest height ($H_{forest}$) and tree height heterogeneity ($\theta$). We replaced LAI, which is not measured, by basal area (both properties correlate quite well in the forest factory data set; $R^2 = 0.74$). The forest stands of each species were classified into six structure classes: three classes which are based on the height of the largest tree in the forest stand (10-15 m, 20-25 m and 30-35 m), and two classes representing different tree height heterogeneities (0-1 and >1.6 m). Only plots that are located on flat terrain (slop less than 15 %) and have a maximum dbh of 0.5 m) were analysed. A linear model was fitted to the data of every class using basal area and elevation as input variables to predict above-ground wood productivity (AWP).

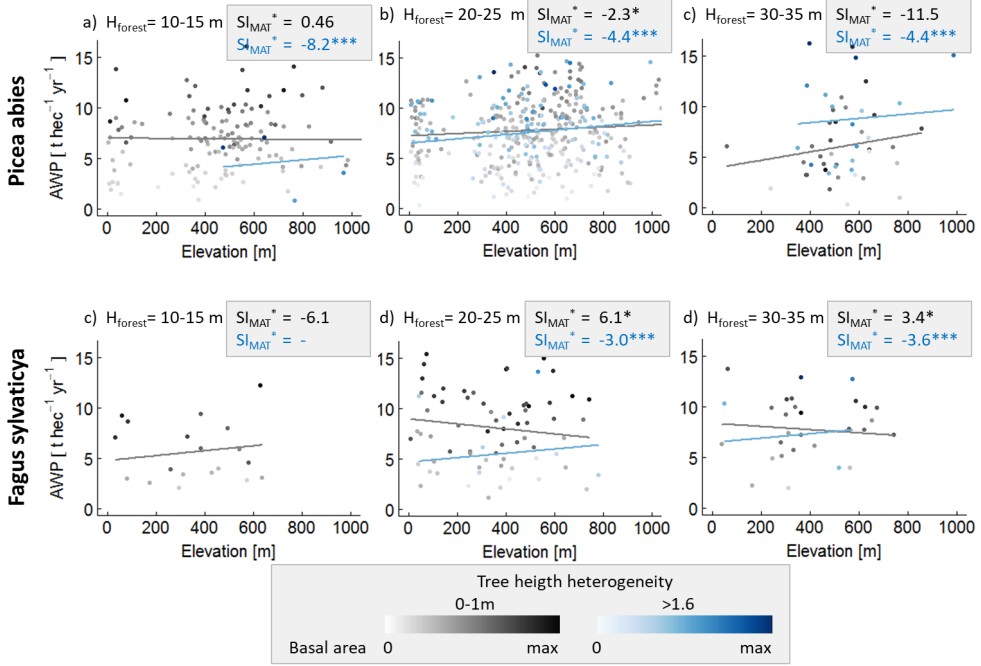

**Figure A3.** Analysis of the influence of forest structure on the relationship between elevation and above-ground wood production (AWP). Figures (a) - (c) are based on spruce monocultures and d)-e) on beech monocultures. For each species, forest stands were classified into three forest height classes which were based on the largest tree ($H_{max}$) in a forest stand. These forest stand classes were additionally separated into two tree height heterogeneity classes (0-1 m in grey and >1.6 m in blue). Intensities of the colours indicate the ratio between basal area of the stand and maximal basal area found within one class. Lines show the results of the linear model with mean basal area. The amount of stars behind the SI values indicates the significance of the slope within a linear model: (***) indicate a p-value below 0.001 and (*) indicates a p-value between 0.01 and 0.05. No star indicates p-values above 0.1. The unit of $SI^*_{MAT}$ is % °C$^{-1}$ .

## Appendix B

### B1 Frequency distribution of sensitivity values

The analysed forest stands show a large range of temperature sensitivity levels, which reach up to 8.5 % °C $^{-1}$ in case of $SI_{MAT}$ (Fig. ref4Bf1a). This means that one forest increases its productivity by 8.5 % due to an increase in the mean annual

temperature of one °$C$. In case of the annual temperature amplitude, the best forest reduces its productivity by -0.5 % °$C^{-1}$ (Fig. ref4Bf1b). The mean $SI_{MAT}$ is 1.5 % °C $^{-1}$ and the interquartile range (iqr) ranges from 1.6 % °C $^{-1}$ to. 5.2 % °C $^{-1}$. The mean $SI_{Q95}$ is -5.4 % °C $^{-1}$ and the iqr ranges from -5.2 % °C $^{-1}$ to -2.2 % °C $^{-1}$.

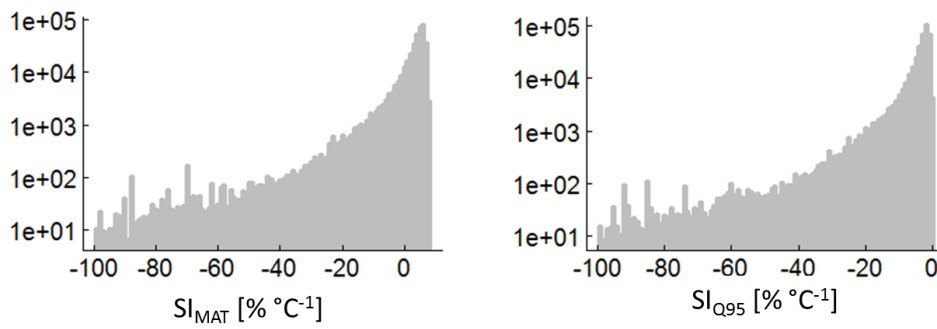

**Figure B1.** Frequency distribution of $SI_{MAT}$ values (a) and $SI_{Q95}$ values (b) of all forest stands.

### B2 Analysis with boosted regression trees

Boosted regression trees provide information about the underlying relationship between input variables (here forest properties)

and output variables (here SI values). Several techniques were developed to visualise and interpret the high-dimensional relationship of input and target variables (Friedman, 2001). The comparisons between SI values of the forest factory and predicted SI values (based on the five properties as input), show a very high agreement (Fig. B2 & B3). The obtained vertical patterns for $SI_{Mat} = 0$ % °C $^{-1}$ and $SI_{Q95} = -6$ % °C $^{-1}$ are probably artefacts of the boosted regression tree algorithm.

Other commonly used visualization of the relationship of input and target variable are partial dependency plots (Fig. 4).

These plots show the influence of an input variable on the target variable considering the influence of all input variables which have higher relative importance. In our study, the most important variable is $\Omega_{AWP}$, hence the first plot shows the relationship between suitability and SI values. The second relationship (forest height on SI values) based on the residuals of the first relationship (here between SI values and $\Omega_{AWP}$; Becker et al., 1996). Although a collection of such plots can seldom provide a comprehensive analysis of the boosted regression trees, it can often produce helpful hints, especially if variables show very

low correlations, as in this study.

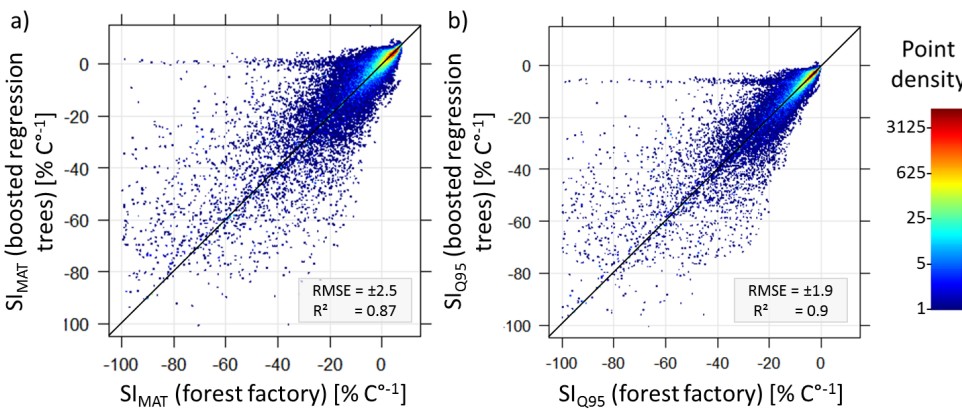

**Figure B2.** Comparisons of temperature sensitivity ($SI_{MAT}$ and $SI_{Q95}$) based on the forest factory and boosted regression tree model. Colours indicate point density. Diagonal is the 1:1 line.

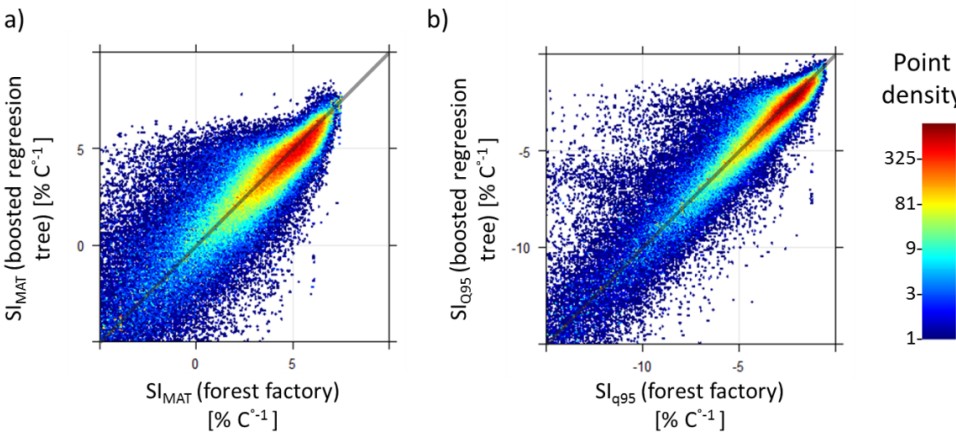

**Figure B3.** Comparison of temperature sensitivity calculations ($SI_{MAT}$ and $SI_{Q95}$) based on the forest factory and boosted regression tree model. Colours indicate point density. Diagonal is the 1:1 line. a) Contains 90% of the forest factory data set and b) contains 93% of the forest factory data set.

## B3   Forest stands properties with highest $SI_{Q95}$ values over a forest height gradient

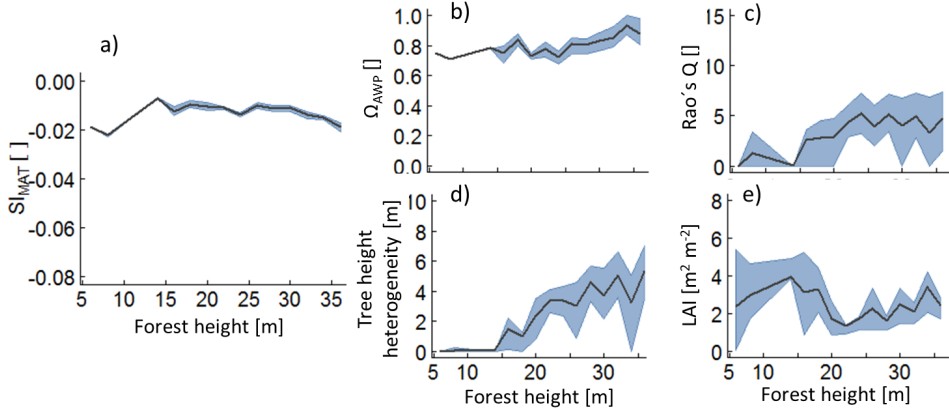

**Figure B4.** Analysis of those forests which lie above the 95% percentile of $SI_{MAT}$, depending on forest height $H_{forest}$. Lines indicate mean values of the subsamples and the grey bands indicate the inter quartile range. Figure a) shows the temperature sensitivity of productivity to forest height, analysing only values above the 95% percentile b) to d) shows the change of the remaining forest properties within the subsamples.

## B4   SI values of single trees

To understand the origin of the SI values, we make the following assumptions: An increase of 1 % °C [-1] always results in an increase of  8.6 % of the respiration rate in the model (Fig. B5 b; Piao et al. (2010)). The positive effect of a temperature increase of 1 % °C [-1] on the photosynthesis rate varies between the years due to the assumed species-specific bell-shaped relationship

5   (Fig. B5 a). In case of deciduous trees the length of the vegetation period (leaf onset to fall) additionally affects the annual photoproduction (e.g., Haxeltine and Prentice, 1996; Luo, 2007; Horn and Schulz, 2011; Gutiérrez and Huth, 2012; Sato et al., 2007). If the photosynthesis rate is much larger than the respiration rate (high AWP; for instance, low ratio of maintenance respiration to photosynthesis under full light in Fig. B6 b), the positive effect of temperature on photosynthesis causes an increase of AWP in some simulated years. If both rates show the same magnitude (ratio of maintenance respiration to photosynthesis

10   under full light is closed to 1 in Fig. B6 b), higher temperatures increase respiration stronger than photoproduction (in most years), which results in a decrease of AWP.

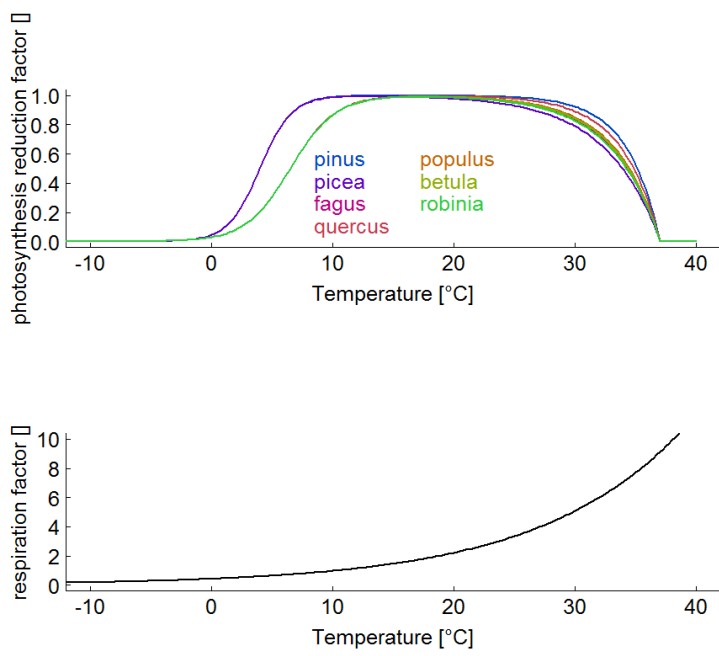

**Figure B5.** a) Species-specific reduction factor of photosynthesis due to a change in air temperature. b) Species-unspecific correction factor for maintenance respiration due to a change in air temperature.

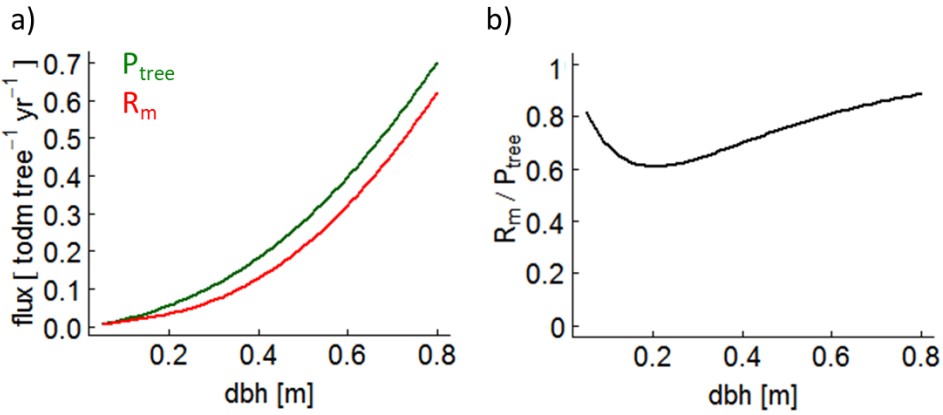

**Figure B6.** a) Photosynthesis (green line - $P_{tree}$) and maintenance respiration (red line - $R_m$) rates of a single beech tree over stem diameter (dbh) under full light. b) The ratio between maintenance respiration and photosynthesis of the same beech tree.

## B5   Functional diversity and temperature sensitivity

To analyse the effect of functional diversity on temperature sensitivity, we first calculated $SI_{MAT}$ for every species depending on tree height and light availability (as done for pine trees in figure 6). Then, we calculated a mean $SI_{MAT}$ value for each species mixture for all light-height combinations . Finally, we averaged those SI-values which are larger than -7.5 % °C $^{-1}$ ($\overline{SI}_{MAT}$) and calculate the Rao's Q of the mixtures (based on equal abundances). The highest $\overline{SI}_{MAT}$ values were found for deciduous forests (Fig. B7). Mixed forests with deciduous and needle leaf trees showed lower values than the deciduous forests, but higher Rao's Q-values.

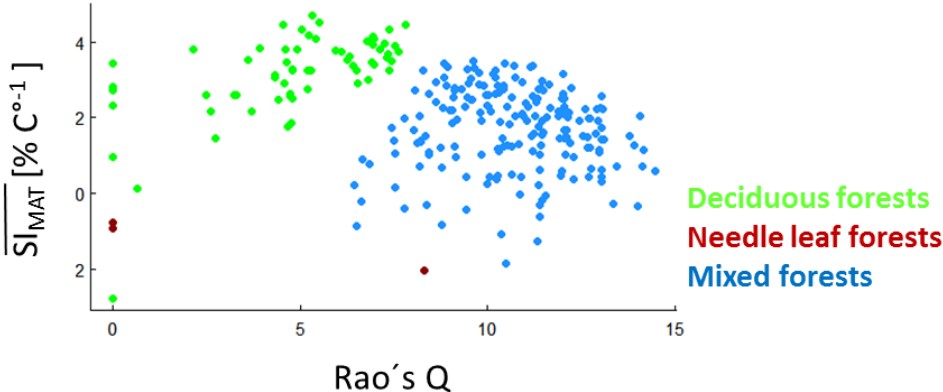

**Figure B7.** Rao's Q (with equal abundances) against $\overline{SI}_{MAT}$ values of all possible species mixtures (from the forest factory). The $\overline{SI}_{MAT}$ values are the average over all $SI_{MAT}$ values for all light-height combinations and with values larger than -7.5 % °C$^{-1}$. For mixtures, we assumed equal abundances and calculate the mean over the $SI_{MAT}$ values of all species within the mixture. Green dots indicate forests that consist only of deciduous trees; red dots indicate forests that consist only of needle leaf trees; blue dots indicate forests that contain both tree types.

*Author contributions.* F.J.B. F.M. and A.H. conceived of the study. F.J.B. implemented and analysed the simulation model and wrote the first draft of the manuscript. A.H. and F.M contributed to the text. All authors gave their final approval for publication.

*Competing interests.* We have no competing interests.

*Acknowledgements.* We thank Edna Rödig, Franziska Taubert, Nikolaj Knapp, Rico Fischer and Kristin Bohn and for providing many helpful
5  suggestions and comments. We also thank the Department of Bioclimatology of the University Göttingen and the Max Planck Institute of Biogeochemistry for providing climate data and the administration of Hainich National Park for permission to conduct research there.

**References**

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

# Species composition and forest structure explain the temperature sensitivity patterns of productivity in temperate forests.

Friedrich J. Bohn[1], Felix May[2], and Andreas Huth[1,2,3]

[1]Helmholtz Centre for Environmental Research - UFZ / Permoserstr. 15 / 04318 Leipzig / Germany
[2]German Centre for Integrative Biodiversity Research (iDiv) Halle-Jena-Leipzig / Deutscher Platz 5e / 04103 Leipzig / Germany
[3]University of Osnabrück / Barbarastr. 12 / 49076 Osnabrück / Germany

*Correspondence to:* Friedrich J. Bohn (friedrich.bohn@ufz.de)

**Abstract.** Rising temperatures due to climate change influence the wood production of forests. Observations show that some temperate forests increase their productivity, whereas others reduce their productivity. This study focus[c1]es on how species composition and forest structure properties influence[c2] the temperature sensitivity of above-ground wood production (AWP). It further investigates[c3] which forests will increase [c4] their productivity the most with rising temperatures. We describe[c5]ed forest structure by leaf area index, forest height and tree height heterogeneity. Species composition [c6]described by a functional diversity index (Rao's Q) and [c7]a species distribution index ($\Omega_{AWP}$). $\Omega_{AWP}$ quantifi[c8]ed how well species are distributed over the different forest layers regarding AWP. We analy[c9]sed 370,170 forest stands, generated with a forest gap model. These forest stands cover[c10]ed a wide range of possible forest types. For each stand we estimate[c11]d annual above-ground wood production and perform[c12]ed a climate sensitivity analysis based on 320 different climate time series (of one year length). The scenarios differ[c13]ed in mean annual temperature and annual temperature amplitude. Temperature sensitivity of wood production [c14]was quantified as [c15]the relative change [c16]in productivity [c17]resulting from a $1°C$ [c18] rise in mean annual temperature or [c19] annual

---

[c1] *Text added.*
[c2] s
[c3] ;
[c4] in
[c5] *Text added.*
[c6] is
[c7] *Text added.*
[c8] es
[c9] z
[c10] *Text added.*
[c11] *Text added.*
[c12] *Text added.*
[c13] *Text added.*
[c14] is
[c15] *Text added.*
[c16] of
[c17] due to
[c18] temperature
[c19] rather

temperature amplitude. Increasing $\Omega_{AWP}$ addpositivly influence[c20]ed [c21] both temperature sensitivity indices of forest, whereas forest height show[c22]ed a bell-shaped relationship with both indices. Further, we [c23]found forests in each successional stage that are positively affected by temperature rise. For such forests, large $\Omega_{AWP}$-values [c24]were important. In case of young forest, low functional diversity and small tree height heterogeneity [c25]was associated with a positive effect of temperature on wood production. During later successional stages, higher species diversity and larger tree height heterogeneity [c26]was an advantage. To [c27]achieve such a development, one could plant below the closed canopy of even-aged, pioneer trees a [c28]climax-species-rich understory that will build the canopy of the mature forest. This study highlights that forest structure and species composition are both relevant [c29]for understanding the temperature sensitivity of wood production.

*Copyright statement.* TEXT

# 1 Introduction

Climate change alters wood production by modifying the rates of photosynthesis and respiration rates of trees (Barber et al., 2000; Luo, 2007; Peñuelas and Filella, 2009; Reyer et al., 2014). Changes in forest productivity have been observed in past decades all over the world (Nemani et al., 2003; Boisvenue and Running, 2006; Seddon et al., 2016). The carbon stock of forests and their role as carbon sink[c1]s are therefore changing. These findings have stimulated discussions about whether forest management strategies can be adapted to reduce forest vulnerability to climate change, to support recovery after extreme events and foster the carbon sink function of forests (Spittlehouse and Stewart, 2004; Spittlehouse, 2005; Bonan, 2008).

Wood production is influenced by several factors, such as $CO_2$ fertilization, nitrogen deposition, precipitation, and temperature. (Barford et al., 2001). For instance, rising [c2]$CO_2$ increases wateruse efficiency of forests (Keenan et al., 2013), which could compensate negative effects of climate change on European forest growth (Reyer et al., 2014). Another important process is [c3] fertilization (De Vries et al., 2006, 2009). Due to depositions of nitrogen in the second half of the last century, wood production had increased in European forests (Solberg et al., 2009). However, temperature modifies photosynthesis, respiration and growth rates of trees (Dillon et al., 2010; Piao et al., 2010; Wang et al., 2011; Jeong et al., 2011; Heskel et al., 2016). In

---

[c20] s
[c21] ~~positively~~
[c22] *Text added.*
[c23] ~~reveal that there are~~
[c24] ~~are~~
[c25] ~~is~~
[c26] ~~is~~
[c27] ~~archive~~
[c28] ~~climx~~
[c29] ~~in~~
[c1] *Text added.*
[c2] ~~CO2~~
[c3] ~~the~~

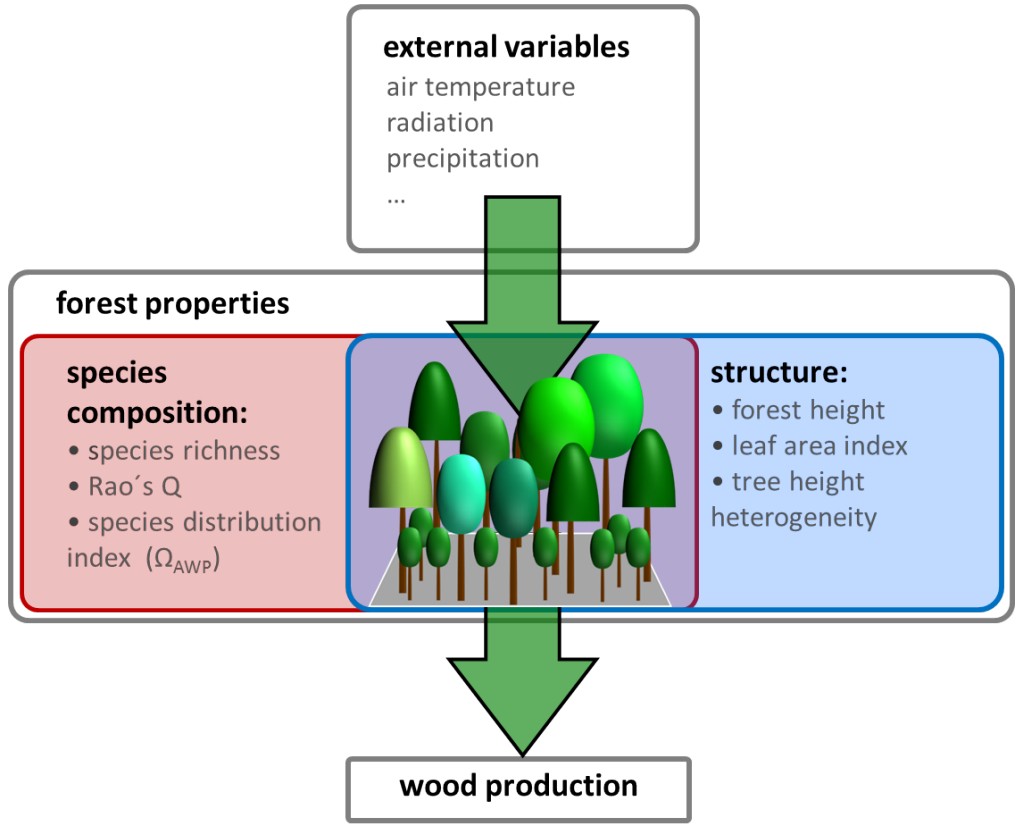

**Figure 1.** Overview of drivers influencing wood production. External variables in this study are temperature, radiation, and precipitation. Forest properties are divided into two groups: species composition properties (e.g., [c1] Rao's Q as a measure of functional diversity and species distribution index $\Omega_{AWP}$) and forest structure properties (e.g., forest height, leaf area index and tree height heterogeneity).

the temperate biome, positive effects on wood production (Bontemps et al., 2010; Delpierre et al., 2009; Pan et al., 2013; McMahon et al., 2010, e.g.) as well as negative ones have been found (Barber et al., 2000; Jump et al., 2006; Charru et al., 2010, e.g.). However, it remains unclear why forests react differently to temperature change.

In addition to the influence of climate variables, wood production is also affected by internal forest properties. These properties can be grouped into two types: properties which describe forest structure, and those which describe species composition (Fig. 1). For instance, changes in productivity can result from changes in basal area (Vilà et al., 2013), in leaf area index (Asner et al., 2003) or in the heterogeneity of tree heights within a forest (Bohn and Huth, 2017). Furthermore, wood production often increases with the increasing number of species (Zhang et al., 2012; Vilà et al., 2007).

Forest stands, which differ in their forest properties, might respond differently to the same climate change (Huete, 2016). For instance, the positive effect of increasing temperature on wood production fades with forest age in temperate deciduous forest (McMahon et al., 2010; Bontemps et al., 2010, e.g.) and Morin et al. (2014) showed that higher diversity buffer[c2]s the effect

---

[c2] *Text added.*

of inter-annual variability on wood production. However, these studies include only a few forest properties and rarely include properties related to both species composition and forest structure. Hence, it is unclear, how these forest properties influence wood production change due to temperature rise and which forests will benefit from rising temperatures.

As far as we know, there is no data set available that covers forests, differing in structure and diversity, under almost identical climatic conditions. Even if a larger number of forest stands were available, it would be difficult to manipulate for instance temperature while keeping all other climate variables constant. [c1]Forest simulations models offer an alternative to the analysis of field experiments Such models are able to estimate wood production under different climate conditions (Lasch et al., 2005; Bohn et al., 2014, e.g.). For instance, Reyer et al. (2014) investigated the effect of climatic change on forests by simulating 30-years time slices of a range of different future climates for 135 inventoried forest stands. There are also model-based studies, which systematically analy[c2]sed the effect of species diversity on productivity and stability over long periods (Morin et al., 2011, 2014). However, disturbed or managed forest stands and the influence of climate change have [c3]not been included in these analyses.

In this study we therefore propose a new simulation-based approach. First, we generate a large number of forest stands covering various forest structures and species compositions (for up to eight temperate tree species). Annual above-ground wood production (AWP) is then calculated for all forest stands based on climate time series. These time series differ in the mean annual temperature [c4] and the intra-annual temperature amplitude [c5]. We aim to analyze (i) how productivity of forest stands (AWP) is influenced by increasing [c6]mean annual temperature and (ii) by increasing intra-annual temperature amplitude? Furthermore, we address the question (iii) of which forest stands will benefit most from rising temperatures.

## 2  Method

To analyse the effect of temperature on the productivity of forest stands, we appl[c7]ied the "forest factory" model approach (Bohn and Huth, 2017). The forest factory generate[c8]d 370,170 different forest stands (see section 2.1) and allow[c9]ed the estimation of aboveground wood production (AWP) under various climate time series (see section 2.2). The 320 scenarios differ[c10]ed in mean annual temperature [c11] and annual temperature amplitude [c12]. Finally, we calculate[c13]ed the forest stand-specific sensitivity of

---

[c1] ~~An alternative option to such field experiments analysis is offered by forest simulation models.~~

[c2] ~~z~~

[c3] ~~been not~~

[c4] ~~(MAT)~~

[c5] ~~(Q95)~~

[c6] ~~temperature (MAT)~~

[c7] ~~y~~

[c8] ~~s~~

[c9] ~~s~~

[c10] *Text added.*

[c11] ~~(MAT)~~

[c12] ~~(Q95)~~

[c13] *Text added.*

productivity [c14]to temperature change [c15] as the relative change of wood production per temperature change of 1 °C (see section 2.2). To relate these sensitivities to forest structure and species composition, we characteri[c16]sed every forest stand with five properties (see section 2.4). We analyse[c17]d the influence of the five forest properties on temperature sensitivity using boosted regression trees (see section 2.5). Finally, we analyse[c18]d which combination of forest properties result[c19]ed in the highest

sensitivity values for different successional stages (see section 2.6).

## 2.1   The forest factory approach

The forest factory creates forest patches [c1] based on [c2] different stem size distributions and [c3] species mixtures. [c4]

[c5]We used 15 stem size distributions cover[c6]ed a gradient from young to old and disturbed to undisturbed forests. Species mixtures include[c7]d all possible [c8]256 combinations of *Pinus sylvestris*, *Picea abies*, *Fagus sylvatica*, *Quercus robur*, *Fraxinus*

*excelsior*, *Populus x canadensis*, *Betula pendula* and *Robinia pseudostuga*. We use[c9]d the species parameter set and algorithms of the FORMIND model version for temperate forests within the forest factory (Bohn et al., 2014; Fischer et al., 2016). [c10]100 forests patches of each combination were built.

To generate forest patches, [c11]the forest factory randomly choose trees from the stem size distribution, assign a species identity and plant them within a patch of 400 $m^2$ size. To place a tree within a patch the following rules must be me[c12]t: (i)

there must be [c13] enough space [c14]available for crowns of every tree [c15]and (ii) every tree in the forest must have a positive productivity under its environmental conditions (light, temperature, water).

---

[c14] ~~against~~

[c15] (~~$SI_{MAT}$ and $SI_{Q95}$~~)

[c16] ~~ze~~

[c17] *Text added.*

[c18] *Text added.*

[c19] s

[c1] ~~which is~~

[c2] ~~15~~

[c3] ~~256~~

[c4] ~~100 forests patches of each combination are built.~~

[c5] ~~The~~

[c6] *Text added.*

[c7] *Text added.*

[c8] *Text added.*

[c9] *Text added.*

[c10] *Text added.*

[c11] ~~we~~

[c12] ~~e~~

[c13] ~~available~~

[c14] *Text added.*

[c15] *Text added.*

We use[c16]ed climate time series [c17]from the year 2007, measured at Hainich National Park, central Germany. We assume[c18]d this time series [c19]to be a typical example for a temperate year (in principle it possible to use climate data [c20]from [c21]any other location). In contrast to an artificially generated climate, this climate is perfectly physically consistent (regarding light, air temperature and precipitation).

5    In a few cases not all species of the mixture could be placed within a patch by the algorithm, so we rejected such forests. We end[c1]ed up with 370,100 forest stands. For more details regarding the forest factory see Bohn and Huth (2017).

## 2.2 Wood production

The calculation of above-ground wood production (AWP) of trees was based on algorithms of the model FORMIND (Bohn et al., 2014; Fischer et al., 2016). [c2]TIn this model, the wood production of a single tree is calculated as the difference between 10   climate variables driven respiration rates and photosynthesis. The photosynthesis rate ($P_{tree}$) results from the crown size, self-shading within the crown and available light at the top of the tree. The available light depends on the radiation above the canopy, reduced by the shading of larger trees within the forest stand. Furthermore, productivity can be limited due to air temperature and available soil water, which is expressed by the photosynthesis-limiting factor $\phi$ for each tree (Gutiérrez, 2010; Fischer, 2013; Bohn et al., 2014). Available soil water within the stand results from precipitation, interception, evapotranspiration of 15   trees and run-off.

One part of the photosynthesis production of a tree ($P_{tree}$) is allocated to its maintenance respiration (and to non-wood tissues; $R_m$). [c3]Maintenance respiration depends on tree biomass and temperature $\psi$ (Piao et al., 2010). The remaining organic carbon is transformed into newly grown above-ground wood ($AWP_{tree}$) and a proportional growth respiration ($r_g$).

$$AWP_{tree} = (\phi P_{tree} - \psi R_m)(1 - r_g) \tag{1}$$

20   $AWP_{tree}$ [c4]was summed over all trees to obtain the productivity of the [c5]modeled forest stand - AWP (for a more detailed description of growth processes, see Bohn et al. (2014); Bohn and Huth (2017) ).

---

[c16] *Text added.*
[c17] ~~of~~
[c18] *Text added.*
[c19] ~~as~~
[c20] ~~of~~
[c21] ~~every~~
[c1] *Text added.*
[c2] *Text added.*
[c3] ~~$R_m$~~
[c4] ~~is~~
[c5] *Text added.*

## 2.3 Climate sensitivity

To generate a set of 320 annual climate time series, we selected daily climate measurements of the Hainich station in central Germany between the years 2000 and 2004. This time series includes mean daily radiation, precipitation and air temperature (see Appendix A1; Fig. A1)). We separated these time series into five distinct time series of one-year length. First, we increase[c6]d or decrease[c7]d the mean annual temperature of each year by adding or subtracting 0.5 °C steps between -1.5 °C and +2 °C. Second, we change[c8]d the amplitude of the annual temperature cycle for these time series variation of each year. To do so, we modif[c9]ied the standard deviation of each year by 4% steps between -12 % and +16 %. We end[c10]ed up with five sets of climate times series (of one-year length) that differ in temperature, precipitation and radiation. Each of these five sets includes 64 time series, which differ only in temperature (see Appendix A1 , Fig. A2). Temperature change [c11]was quantified using two indices: (i) mean annual temperature [c12] and (ii) annual temperature amplitude [c13], which describe[c14]d the 95 % inter-quantile range of all daily temperature values of a given year. We [c15]did not model the effects of nitrogen and $CO_2$ fertilization (as both do not vary strongly within one year) or extreme anomalies (e.g., pathogen attacks) on wood production. Figure 2 (a-c) shows the above-ground wood production (AWP) for different annual temperatures for three different forest stands.

We analysed the sensitivity of every forest stand to temperature change [c6] following the approach of Piao et al. (2010). For every forest stand, a general linear model [c7]was fitted relating wood production [c8]mean annual temperature (MAT) and intra-annual temperature amplitude (Q95), as well as the nuisance parameter year.

$$AWP = \alpha x_{MAT} + \beta x_{Q95} + \gamma x_{year} + \epsilon \tag{2}$$

For every forest, we calculate[c9]d the relative change of productivity [c10]resulting from an increase of 1 °C:

$$SI_{MAT} = \frac{\alpha}{AWP} \tag{3}$$

$$SI_{Q95} = \frac{\beta}{AWP} \tag{4}$$

---

[c6] *Text added.*
[c7] *Text added.*
[c8] *Text added.*
[c9] y
[c10] *Text added.*
[c11] is
[c12] (MAT)
[c13] (Q95)
[c14] s
[c15] do
[c6] by
[c7] is
[c8] and the two temperature indices MAT and Q95
[c9] *Text added.*
[c10] due to

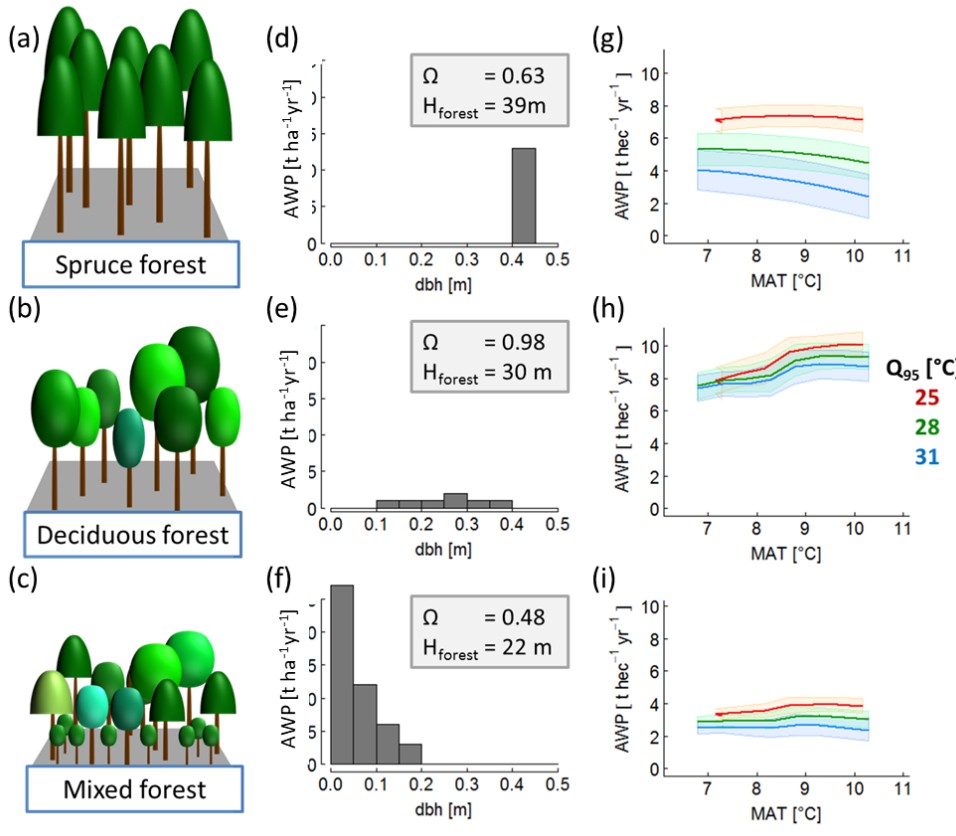

**Figure 2.** Overview of forest properties and resulting temperature sensitivity of above-ground wood production (AWP) of three exemplary forests: a) old even-aged spruce forest; b) mature deciduous forest; c) a quite young mixed species forest. The middle panel (subfigures d, e & f) shows the corresponding stem size distributions and [c1]provides information on the [c2]highest tree in the forest($H_{forest}$) and species distribution index $\Omega_{AWP}$ (which quantifies [c3]the suitability of a species distributed within the forest structure regarding AWP). Each forest is treated with 320 climate time series: The last panel shows the AWP as a a function of mean annual temperature (MAT). The colours indicate different inter-annual temperature amplitudes (Q95) of the used time series. (The coloured bands show the standard deviation due to the variability of the five different time series [c4]that exist for each combination of mean annual temperaturemean and [c5]intra-annual temperature amplitude).

In our analysis we exclude[c11]d all forests stands for which AWP turns negative if the temperature rises by 1 °C (This occurs in 2 % of all stands).

---

[c11] *Text added.*

We also determine[c1]d the sensitivity of forests [c2]to temperature change using the German forest inventory to validate our results. However, the inventory does not include [c3]leaf area index (LAI) measurements. We therefore assume[c4]d the basal area as a proxy for LAI, and we select[c5]ed subsamples of forests stands with similar structure (basal area, tree height heterogeneity, forest height, and same species mixtures). In addition, we use[c6]d elevation as a proxy for mean annual temperature, assuming temperature changes of 0.65 °C per 100 metres on average (Foken and Nappo, 2008). Only in the case of spruce and beech monocultures did we find enough data to calculate $SI_{MAT}$ values for several forest structures ([c7]for more details see Appendix A3, Fig. A3). [c8]

The comparison between the $SI_{MAT}$-estimation based on the German forest inventory with $SI_{MAT}$ values of corresponding forests from the forest factory show[c1]ed quite good agreement ($R^2 = 0.65$). However, the simulated $SI_{MAT}$ values of the forest factory slightly overestimate[c2]d the sensitivity compared to the inventory-based values (Fig. 3). [c3]This might be explained by the difference in the methods [c4]used because, in case of the inventory, we use[c5]ed basal area instead of LAI and altitude instead of temperature. Another [c6]explanation could be that in our approach the climate time series show[c7]ed relatively high and regular precipitation. In the German forest inventory [c8]warmer sites might [c9]be more frequently exposed to water stress, which than reduce[c10]ed the SI values.

## 2.4 Five forest properties to describe forest stands

We use[c12]d three indices to describe the forest structure: leaf area index (LAI), maximum forest height [c13]($H_{forest}$) [c14], which corresponds to the height of the largest tree in a forest stand , and [c15]tree height heterogeneity ($\theta$), which [c16]was quantified by the standard deviation of the tree heights. To describe species composition, we use[c17]ed Rao's Q and species distribution index

---

[c1] *Text added.*

[c2] ~~against~~

[c3] ~~LAI~~

[c4] *Text added.*

[c5] *Text added.*

[c6] *Text added.*

[c7] *Text added.*

[c8] ~~The correlation of the sensitivity values based on field data and simulation data was quite high.~~

[c1] *Text added.*

[c2] *Text added.*

[c3] ~~The reason might be~~

[c4] ~~as~~

[c5] *Text added.*

[c6] ~~reason~~

[c7] *Text added.*

[c8] ~~instead,~~

[c9] ~~exposed more often~~

[c10] ~~s~~

[c12] *Text added.*

[c13] *Text added.*

[c14] ~~($h_{max}$ ) and tree height heterogeneity ($\theta$). $h_{max}$~~

[c15] ~~$\theta$~~

[c16] ~~is~~

[c17] *Text added.*

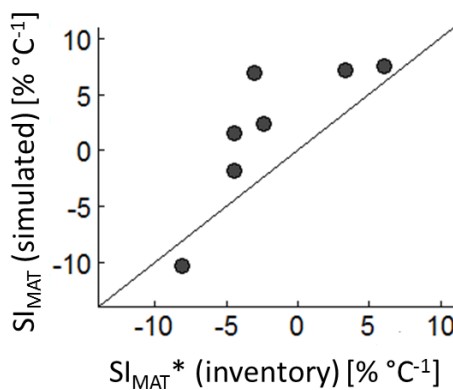

**Figure 3.** $SI_{MAT}$ values of seven different forest types derived from the analysis of the German forest inventory vs. $SI_{MAT}$ values derived from corresponding forest types of the forest factory. Only those $SI_{MAT}$ values of the field data are analysed, which show[c11]ed p-values smaller then 0.05.

($\Omega_{AWP}$). Rao's Q quantifie[c18]d functional diversity based on species abundances and differences in species traits (Botta-Dukát, 2005, for details see Appendix A2). $\Omega_{AWP}$ analyse[c19]d the optimal location of species within the forest structure. $\Omega_{AWP}$ is defined as the ratio of the forest's productivity to the maximum possible productivity of the forest without changing tree sizes or number (Bohn and Huth, 2017). Hence, the maximum productivity can be obtained by varying only the species identities

5 of trees in the forest stand. We change[c20]d the assigned species of each tree until we [c21]found the optimal species for each individual tree and its specific environmental condition . All five indices [c22]were nearly uncorrelated for the investigated forest stands ( Appendix A2 Table A1).

## 2.5 Boosted regression trees

We applied boosted regression trees to quantify the influence of the five forest properties on $SI_{MAT}$ and $SI_{Q95}$. [c1]Boosted

10 regression trees are a machine learning algorithm using multiple decision (or regression) trees. It is able to address unidentified distributions (De´Ath, 2007; Elith et al., 2008). Each model [c2]was fitted in a forward stage-wise procedure to predict the response of the dependent variable on ($SI_{MAT}$ or $SI_{Q95}$) to multiple predictors ([c3]tree height heterogeneity, forest height,

[c18] s
[c19] s
[c20] *Text added.*
[c21] find
[c22] are
[c1] BRT
[c2] is
[c3] $\theta, h_{max},$

LAI, Rao's Q, and $\Omega_{AWP}$). To omit an over-fitting regarding maximal forest height, we classif[c4]ed forest stands into 18 classes [c5]. Each class [c6]had a width of 2 metres, starting with 4 to 6 metres and finishing with 36 to 38 metres. The [c7]boosted regression trees [c8]tried an iterative process to minimi[c9]se the squared error between predicted SI values and those of the data set. Hereby, part of the data [c10]was used for a fitting procedure and the other part [c11]were used for computing out-of-sample estimates of the loss function (Ridgeway, 2015). This [c12]boosted regression tree analysis was performed in the R-package gbm 2.1.1 (Ridgeway, 2015).

We used a quarter of the data (randomly sampled) for the machine learning procedure. To get the best model, we var[c1]ied the following four [c2] parameters of the boosted regression tree algorithm: learning rate (0.1, 0.05 and 0.01), the bag-fractions (0.33, 0.5 and 0.66), the interactions depth (1, 3 and 5) and the cross-validation (3-, 6- and 9-fold) assuming a Gaussian error structure (the default setting). The best fitted [c3]boosted regression tree for both $SI_{MAT}$ and $SI_{Q95}$ show[c4]ed a learning rate of 0.1, a bag-fraction of 0.66, an interaction depth of 5 and a 3-fold cross validation. These two models were used for all further analyses. The remaining 75% of the data [c5]were used to validate the fitted [c6]boosted regression tree algorithm.

## 2.6 Finding the forest stands for different successional stages that benefit the most increasing temperatures

Here, we assume[c7]d forest height as a proxy for the successional stage of a forest. In every height class, [c8] we select[c9]ed those 5% of forests that show[c10]ed the highest sensitivity values ($SI_{MAT}$ and $SI_{Q95}$). We removed the forest height classes between 10 and 14 metres, as they only contain[c11]ed a few forests (15). For all other classes, we analyse[c12]d the relationship between [c13]height class and the forest properties ($\Omega_{AWP}$, Rao's Q, LAI and [c14]tree height heterogeneity).

---

[c4] y
[c5] $(H_{max})$
[c6] has
[c7] BRT
[c8] try
[c9] z
[c10] is
[c11] are
[c12] BRT
[c1] y
[c2] BRT parameters
[c3] BRT
[c4] *Text added.*
[c5] are
[c6] BRT
[c7] *Text added.*
[c8] $(H_{max})$
[c9] *Text added.*
[c10] *Text added.*
[c11] *Text added.*
[c12] *Text added.*
[c13] $H_{max}$
[c14] $\theta$

## 3 Result

We analysed the sensitivity of productivity (AWP) [c1]to temperature for forest stands that differ in forest properties (species distribution index ($\Omega_{AWP}$), functional diversity (Rao's Q), tree height heterogeneity ($\theta$), forest height class [c2] and LAI). The annual above-ground wood production (AWP) was estimated for each forest stand using 320 different climate time series. We

then quantified the changes in productivity [c3]resulting from changes in [c4]mean annual temperature ($SI_{MAT}$) and [c5]intra-annual amplitude ($SI_{Q95}$). For the analysed forest stands, the average $SI_{MAT}$ is 1.5 % °C $^{-1}$ and the average $SI_{Q95}$ is -5.4 % °C $^{-1}$ (see also the frequency distribution in Appendix B1, Fig. B1).

With a boosted regression tree algorithm, we analysed how the five forest properties influence the temperature sensitivity of forests. To validate[c6]d the fitted [c7]boosted regression tree algorithm, we compare[c8]d SI values, which are not used for the

fitting, with the SI value predicted by the [c9]boosted regression tree algorithm (Fig. 4). The sensitivities [c10]to mean annual temperature change ($SI_{MAT}$) correlate[c11]d very well ($R^2$ of 0.84) and show[c12]ed a low RMSE of $\pm$ 2.9 % °C $^{-1}$ (see Appendix B2 Fig. B3).The RMSE even decrease[c13]ed to $\pm$ 1.5 % °C $^{-1}$ if a subset of the forest stands [c14]was analysed that show[c15]ed $SI_{MAT}$ values larger than -5 % °C $^{-1}$ (90 % of the data). The accuracy of the sensitivities [c16]to temperature amplitude change ($SI_{Q95}$) was even slightly better. In addition, a subset that include[c17]ed $SI_{Q95}$ values larger than -15 % °C $^{-1}$ (93% of the data)

show[c18]ed a RMSE of only $\pm$ 1.1 % °C $^{-1}$ (see Appendix B2 Fig. B4).

According to [c23]boosted regression tree analysis, $\Omega_{AWP}$ [c24]was the most relevant forest property to explain temperature sensitivities (relative influence of 87 % for $SI_{MAT}$ and 89 % for $SI_{Q95}$; see also Appendix B2, Fig. B2). However, the influence of $\Omega_{AWP}$ on temperature sensitivity flatten[c25]ed out for high $\Omega_{AWP}$ levels (Fig. 5). The second relevant forest

---

[c1] ~~against~~ to

[c2] ~~($H_{max}$)~~

[c3] ~~due to~~ resulting from

[c4] ~~MAT~~ mean annual temperature

[c5] ~~Q95~~ intra-annual

[c6] *Text added.*

[c7] ~~BRT~~ boosted regression tree

[c8] *Text added.*

[c9] ~~BRT~~ boosted regression tree

[c10] ~~against~~ to

[c11] *Text added.*

[c12] *Text added.*

[c13] s

[c14] ~~is~~ was

[c15] s

[c16] ~~against~~ to

[c17] s

[c18] s

[c23] ~~BRT~~ boosted regression tree

[c24] ~~is~~ was

[c25] s

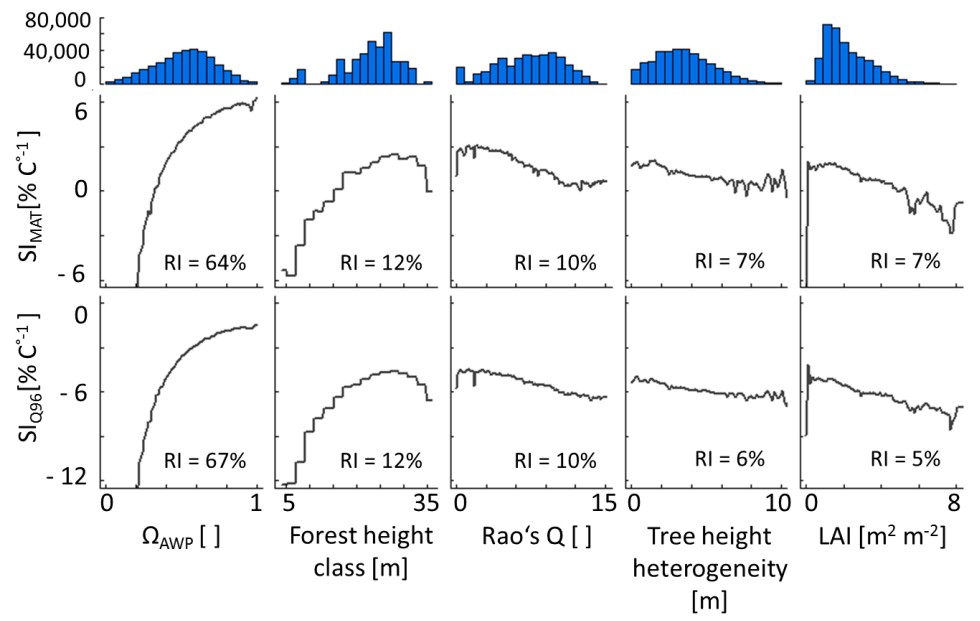

**Figure 4.** Partial dependency plots of the five forest properties $\Omega_{AWP}$ (species distribution index), forest height class [c19], Rao's Q (functional diversity), tree height heterogeneity [c20] and LAI (leaf area index) for $SI_{MAT}$ (sensitivity [c21]to changes inthe mean annual temperature) and $SI_{Q95}$ (sensitivity [c22]to changes in annual temperature amplitude). Relative importance (RI) compares the influence of different input variables on the variability of a target variable. Histograms show the frequency of forest property values in the analysed data set. Note, a $\Omega_{AWP}$ is the ratio of the current AWP of a forest and the highest possible AWP optained by shuffling ony species identies without changing the forest structure.

property [c26]was forest height [c27]($H_{forest}$). Forests with heights between 25 and 30 m benefit[c28]ed the most from increasing mean annual temperatures. The other three properties (LAI, Rao's Q, and [c29]tree height heterogeneity) [c30]had a low influence on $SI_{MAT}$.

Both sensitivity indices show[c2]ed similar relationships to the five forest properties. However, an increase in annual tem-
5    perature amplitude always reduce[c3]d productivity, whereas increasing mean annual temperature [c4]could resulte in a positive effect on wood production. To detect those stands that benefit the most from increasing temperature, we select[c5]ed the 5 % of forest stands that showed the highest $SI_{MAT}$ values in each forest height class (Fig. 5). In all forests classes, we found forest

[c26] ~~is~~
[c27] ~~($H_{max}$)~~
[c28] *Text added.*
[c29] ~~$\theta$~~
[c30] ~~have~~
[c2] *Text added.*
[c3] s
[c4] ~~can~~
[c5] *Text added.*

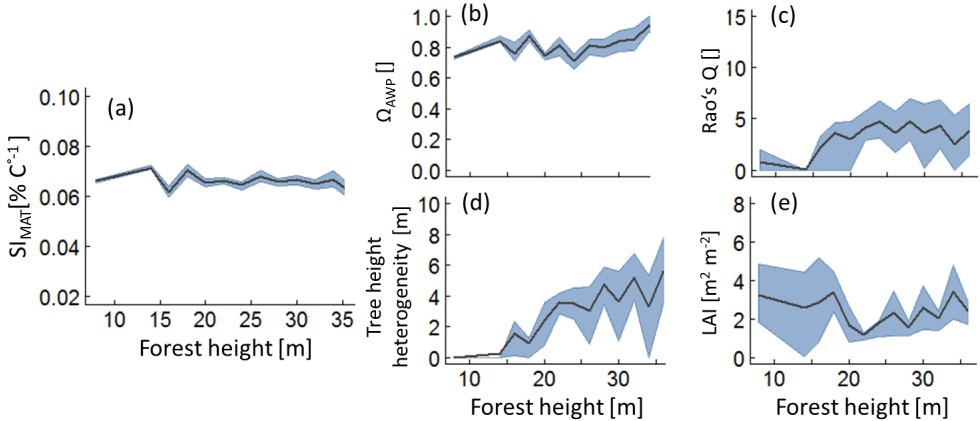

**Figure 5.** Analysis of those forests that show the highest 5 % of the SI values depending [c1]on forest height. Lines indicate mean values of the forest subsamples which includes the best 5% regarding $SI_{MAT}$ of each hight class. The grey band indicates the inter quartile range. Figure a) shows temperature sensitivity of above-ground wood production over forest height, analysing only the best the forest subsample. b) to d) shows the change of the remaining forest properties within the forest subsamples ($\Omega_{AWP}$ = optimal species distribution; LAI = leaf area index; Rao s Q quantifies functional diversity).

stands that would benefit from increasing temperatures. Analys[c6]es of their forest properties reveal[c7]ed that the $\Omega_{AWP}$ levels were always high. Young forests (low forest height), which [c8]had a positive temperature sensitivity, show[c9]ed low functional diversity and low tree height heterogeneity ($\theta$). For older forests (of intermediate and high forest height) with positive temperature sensitivity, we found an intermediate level of functional diversity. Interestingly, for three variables (Rao's Q, [c10]tree
5   height heterogeneity and LAI), the relationships change[c11]d their character between young and intermediate forest heights. We obtain[c12]ed similar simulation patterns for $SI_{Q95}$ ( Appendix B3 Fig. B5).

[c6] ~~ing~~
[c7] ~~s~~
[c8] ~~have~~
[c9] *Text added.*
[c10] ~~θ~~
[c11] *Text added.*
[c12] *Text added.*

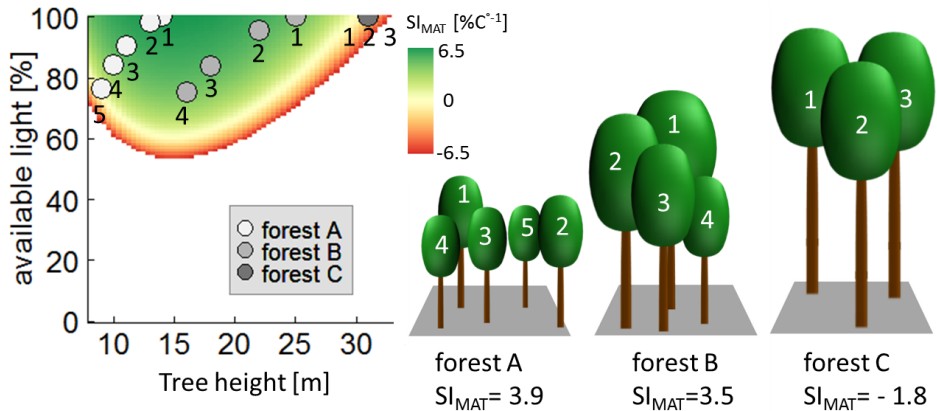

**Figure 6.** Analysis of [c3]the sensitivity index of AWP against mean annual temperature ($SI_{MAT}$) values of single trees within three different forests. The diagram shows the calculated $SI_{MAT}$ value of individual trees for every combination of tree height and available light (for *pinus sylvestris* between $SI_{MAT}$-levels of 6.5 and - 6.5; other species show similar patterns). The dots indicate the different trees of the three forest examples. The white dots belong to trees with the corresponding number of forest A, grey dots belong to the trees of forest B and dark grey dots belong to forest C. Note that in the case of forest C, all trees have the same height and the same light, so that all three dots are at the same place in the diagram.

## 3.1    Understanding the patterns

### 3.1.1    The influence of forest structure on temperature sensitivity

Forest structure affects [c1]the wood production of single trees in two ways. First, it determines [c2]the amount of light available to each individual tree and second, the size of trees influences their photosynthesis and respiration rates (Fig. B6). Hence,

5    based on the height of a tree and [c3]the amount of light available to it, it [c4]was possible to calculate its SI values (for a detailed discussion of these calculations, see Appendix B4).

In even-aged forests, all trees have the same height and receive full light (e.g., Fig. 6, forest C). [c1]In our study,  such forests show[c2]ed a bell-shaped relationship between forest height and temperature sensitivity (Fig. 6, SI values for 100 % available light depending on tree height).

10    In case of a forest consisting of trees of different heights smaller trees receive less light due to shading. Note that, even if trees receive less light, the bell-shaped relationship between tree height and productivity persist[c4]ed (Fig. 6). Two cases will be

---

[c1] *Text added.*
[c2] ~~the available light for each single tree~~
[c3] ~~its available light~~
[c4] ~~is~~
[c1] *Text added.*
[c2] *Text added.*
[c4] ~~s~~

discussed (assuming identical LAI as forest C, Fig. 6). In the first case all trees have not yet reached their maximal SI values (Fig. 6, forest A,); and in the second case all trees [c5]have already passed their maximal SI values (Fig. 6, forest B). In the case of forest A, trees in the shade of larger trees always [c6]had lower SI values if they belong to the same species (see Appendix B4). Hence, the temperature sensitivity level of this forest [c7]was lower than the sensitivity of an even-aged forest, whose trees

have the same size as the largest tree in forest A (Fig. 6, tree 1). Hence, if maximal SI values [c8]were not reached, increasing height heterogeneity decreases SI values of a forest.

In forest B (Fig. 6), SI values of the shaded trees can be similar (or even higher) than the SI value of the largest trees in the forest (SI values of tree 1 show similar levels [c1]to tree 2, 3 and 4 in forest B, Fig. 6). Hence, if maximal SI values [c2]were passed, increasing tree height heterogeneity result[c3]ed in similar (or even more positive) temperature sensitivity levels

compared to even-aged forest trees (an even-aged forest consisting only of trees similar to tree 1 of forest B in Fig. 6). These general considerations explain the change from low levels of height heterogeneity in young forests to a more heterogeneous structure in the analysis of those forests, which will benefit from increasing temperature (see Fig. 5 d).

### 3.1.2   The effect of species composition on temperature sensitivity

In this study, we use the new index $\Omega_{AWP}$ called [c4]the species distribution index (Bohn and Huth, 2017). $\Omega_{AWP}$ is the

ratio between current AWP and the highest possible AWP of the forest which can be reached due to shuffling of species identities. Its huge importance on forest temperature sensitivity might be illustrated by the following considerations: If species are unfavourably distributed within the forest (low $\Omega_{AWP}$), the AWP of the forest is low and, in consequence, the SI values are low as well (see Appendix). If [c5]the AWP is low the forest will suffer from increasing temperatures, which results in negative slopes [c6](Equation 2). These values are then divided by low AWP values (Equation 4), which results in large negative values

of $SI_{MAT}$ and $SI_{Q95}$. (See Appendix B5).

Increasing functional diversity (Rao's Q) stabili[c8]sedes the forests' sensitivity to temperature. This corresponds to results of Morin et al. (2014) and the theoretical consideration of Yachi and Loreau (1999). The analysis of the single species can give additional insight into the mechanisms behind those species that benefit[c9]ed the most from temperature increase, which [c10]were deciduous trees under most conditions. This is reasonable as warmer regions host more deciduous species than needle-leaf

---

[c5] *Text added.*
[c6] ~~have~~
[c7] ~~is~~
[c8] ~~are~~
[c1] ~~as~~
[c2] ~~are~~
[c3] *Text added.*
[c4] *Text added.*
[c5] *Text added.*
[c6] ~~(ΔAWP / ΔT)~~
[c8] ~~zes~~
[c9] *Text added.*
[c10] ~~are~~

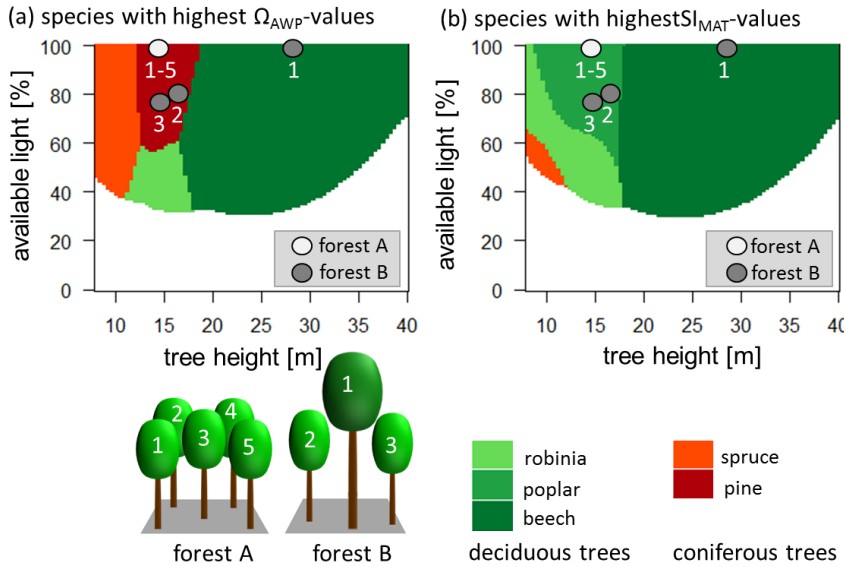

**Figure 7.** Graphic (a) shows which species have the highest productivity ($\Omega_{AWP}$ value of 1) under the current climate for different heights and different light conditions. Graphic (b) shows which species shows the highest increase [c7]in productivity due to rising temperatures for different heights und different light conditions. Red colours indicate coniferous trees, whereas green colours indicate deciduous trees. Darker colours indicate late successional species, whereas lighter colours indicate pioneers. The dots indicate the different trees of the two forest examples (A and B). The white dots belong to trees with the corresponding number of forest A. Note, that all trees have the same height and the same light, so all five dots are at the same place in the diagram. Grey dots belong to the corresponding trees with the same number of forest B.

species. The highest functional diversity (Rao's Q) [c11], one the other hand, occur[c12]ed in mixtures of deciduous and needle-leaf trees (Appendix B5 Fig. B7). As only two needle-leaf species [c13]were considered here in the species pool, low Rao's Q values [c14]were dominated by mixtures of deciduous trees. Such deciduous tree mixtures mostly benefit[c15]ed from temperature increases. In consequence, mixtures with high Rao's Q values, which mostly include[c16]d both functional types, react[c17]ed more poorly (Fig. 4; Appendix B5 Fig. B7).

We developed two diagrams that show the species with the highest temperature sensitivity and with the highest productivity for different conditions (available light and height of a tree) (Fig. 7). Interestingly, the species with the highest productivity

[c11] ~~instead~~
[c12] s
[c13] ~~are~~
[c14] ~~are~~
[c15] *Text added.*
[c16] *Text added.*
[c17] *Text added.*

differ[c1]ed from the species that benefit most from rising temperatures in many cases. This has important [c2]implications. The highest benefit due to increasing temperatures [c3]was optained by forests with high but not maximal $\Omega_{AWP}$ (Fig. 5). Additionally, deciduous trees benefit[c4]ed more than coniferous trees from rising temperatures (Fig. 7, Appendix B5, Fig. B7). Hence, young forests should consist of deciduous trees (compare Fig. 6, forest A, and Fig. 7), although the highest productivity values are found for coniferous trees (Fig. 7; forest A). Forests including large trees obtain[c5]ed the highest sensitivity values if intermediate sized trees differ[c6]ed in their species identity from the largest trees (Fig. 7).

## 4 Discussion

c1

### 4.1 The study design

In this theoretical study, we present a new climate sensitivity analysis (regarding temperature) [c2]of AWP. This approach extends field observations and long-term model simulations, as it allows the analysis of [c3]exsisting forests, but also of those that might exist in the future due to management changes and/or disturbances. Our approach includes only forest stands in which every tree [c4] has positive productivity and enough space for its crown. Hence, it is impossible, for instance, that light-demanding species grow below a closed canopy or forests are overcrowded. However, the data set [c5]also include a few very unusual stands structures or species combinations, which can not emerge in a natural system, but may result from disturbances or management. In the case of field observations, it is difficult to explore the influence of a single climate variable (e.g., temperature) on one target variable (e.g., AWP), as in most cases, several variables are altered at the same time (see also Appendix A3). Process-based models are one option to analyse such relationships and separate these effects. The simulation of AWP with the FORMIND-model in temperate forests has been successfully compared to Eddy flux sites (Rödig et al., 2017b), the national German forest inventory (Bohn and Huth, 2017), and European yield tables Bohn et al. (2014).

An advantage of the forest factory approach is the huge set of various forests stands that can be analysed. The dataset includes forest stands that often occur in temperate forests (even-aged spruce, pine and beech stands). However, it also includes

---

[c1] *Text added.*

[c2] ~~consequences~~

[c3] ~~obtain~~

[c4] *Text added.*

[c5] *Text added.*

[c6] *Text added.*

[c1] ~~In this study, we analyse how temperature changes affect above-ground wood production (AWP) and quantify the effect of five different forest properties on this relationship. The change of AWP was investigated for 370,170 forest stands under 320 different climate time series. Our analysis shows a high influence of $\Omega_{AWP}$ and $H_{max}$ on the temperature sensitivity of AWP. Further, for all successional stages of forests, we detect some forests with a specific set of forest properties which benefit from temperature rise. This specific combination varies with forest height.~~

[c2] ~~on~~

[c3] ~~forests, which already exist but also which~~

[c4] ~~in a forest~~

[c5] ~~include also~~

hypothetical ones that could occur through alternative forest management or disturbances (fire, bark beetles, etc.). Hence, our data set of forest stands covers a much larger variety of forest property combinations compared to long-term forest simulations with the focus on natural forests in their equilibrium state (Morin et al., 2011, e.g.) or on monocultures (Reyer et al., 2014, e.g.). Long term simulations with ecosystem models, which process modelled climate projections, face a trade-off between cascade

uncertainty and path dependency (Wilby and Dessai, 2010; Reyer et al., 2014). The accumulations of model uncertainties over such a process chain result in [c6] increasing uncertainty. Our study design tries to minimi[c7]se this uncertainty and omit path dependencies by including only those processes that might be relevant for the research question. In this study, for instance, we omit the effect of climate change on regeneration and mortality. Furthermore, using several climate variables as model inputs but only analysing the effect of one variable might lead to incorrect interpretations of its effect. For example, temperature and

radiation often correlate, and both might increase productivity. Therefore, in this study, we only vary one variable in all 5 sets of time series. This guarantees [c8]that there are no relationships between the target climate variable and the remaining climate variables.

     As an increase in global mean temperature of $1.5\,°C$ to $2\,°C$ can hardly be avoided, even under the RCP 2.6 climate scenarios (IPCC, 2013), this study focuses on temperature change. This RCP scenario predicts only small changes [c1]in annual

precipitation levels for temperate regions. Hence, our approach focuses only on the effect of temperature change on wood production. However, this might be critical for [c2]analysis of strong temperature changes (e.g. RCP 8.5) which will result in an [c3]increased incidence of drought and changes in [c4] annual temperature cycles and a strong change in $CO_2$. Such more complex scenarios should be analysed in future studies. Further, we neglect the effect of time lags (e.g. bud building in the previous year). However, it is possible to extend the used time series to analy[c5]se the behaviour of the forest over longer time periods

and study not only productivity, but also effects on regeneration or mortality.

     To characteri[c6]se the annual [c7]temperature cycles we used two variables: mean annual temperature [c8] and intra-annual temperature amplitude [c9]. Both variables can be varied independently. In case of higher [c10]mean annual temperature we observe an elongation of the vegetation period. This leads to higher forest productivity (if other resources are not limiting (Luo, 2007) and explains why $SI_{MAT}$ is often positive. However, warmer summer temperatures can also lead to a decline in wood production

due to an increase in respiration. In case of increasing [c11]intra-annual temperature amplitude, more days with extreme temper-

---

[c6] an

[c7] z

[c8] *Text added.*

[c1] of

[c2] the

[c3] increase

[c4] the

[c5] z

[c6] z

[c7] cycles of temperature

[c8] (MAT)

[c9] (Q95)

[c10] MAT

[c11] Q95

atures will occur in a year. Thus, an increase of $1\,°C^{-1}$ of [c12]intra-annual temperature amplitude will increase respiration more strongly compared to an increase of $1\,°C^{-1}$ of [c13]mean annual temperature. Hence, the increase of [c14]intra-annual temperature amplitude [c15]normally has negative effects on the productivity (negative SI values).

The temperature sensitivity values obtained here are in the same range as [c1]those found for temperate ecosystems in heating experiments (Lu et al., 2013, $4.4 \pm 2.2\,\%\,°C^{-1}$). Within the 16 analysed studies reviewed by Lu et al. (2013), the experimental plots show almost identical environmental conditions (soil, radiation, and precipitation) and species composition. To heat the plots, greenhouses or infrared heaters were used. Another study, based on natural forest stands in New Zealand, found an AWP increase [c2]of between 5 and $20\,\%\,°C^{-1}$ for forest, assuming no change in forest structure and species composition (Coomes et al., 2014). The analysed plots were spread [c3]throughout New Zealand, and warmer temperatures coincide with higher radiation (Mackintosh, 2016). Hence, the analysed temperature effect also includes the influence of radiation. In our setting, however, the influence of temperature is independent [c4]of radiation (Lu et al., 2013, as in). We also found a good correlation between SI values derived from growth measurements of the German forest inventory and simulated SI values based on the forest factory ( Appendix A3 Fig. A3 & 3 ).

### 4.2 Implications for forest management

Our findings might be relevant for future management strategies [c5]for temperate forests. Specifically, our new understanding of which species benefit most from rising temperatures (Fig 6) suggests possible strategies, e.g. replacing spruce monocultures with mixtures of deciduous trees. Further, based on the analysis of which forest structure benefits most from rising temperatures (Fig. 4, Fig 5, Fig 6), early-stage even-aged forests should include mainly pioneer species. In the mature stage, we predict a positive effect of temperatures on wood production for a mixture of climax species including different tree sizes. These climax species could be planted below the canopy of the pioneer species in young forests. In our approach, we do not simulate the establishment of very young trees. However, during the conversion between these two forest types one big challenge might be the removal of the pioneer trees without damaging the young trees[c6]that will build the mature forest.

### 4.3 Implications for global vegetation modelling

Most global vegetation models [c7] represent vegetation as fractional cover of different plant functional types within a grid cell (e.g. LPJ Sitch et al., 2003). Only a few global vegetation models include a more detailed representation of vegetation structure

---

[c12] ~~Q95~~
[c13] ~~MAT~~
[c14] ~~Q95~~
[c15] ~~has normally~~
[c1] ~~that~~
[c2] *Text added.*
[c3] ~~all over~~
[c4] ~~from~~
[c5] ~~of~~
[c6] ~~, which~~
[c7] ~~(DGVM)~~

and functional diversity (Sato et al., 2007; Scheiter et al., 2013; Sakschewski et al., 2016). It would be interesting to perform the [c8] analysis [c9]presented here [c10] with global vegetation models which include structure to better understand the mechanisms driving [c11]forest systems' sensitivity to climate change.

Besides the global vegetation models, forest gap models, which [c1]in the past have been restricted to local stands, [c2] are now able to simulate forest dynamics in regions or even entire continents (Seidl and Lexer, 2013; Rödig et al., 2017a). Studies using [c3]global vegetation models or large-scale forest gap models simulate natural succession. Our analysis indicates that natural and managed (or disturbed) forest systems, which differ in forest structure, might react differently [c4]to climate change. Hence, we suggest considering forest structure in future analyses of global vegetation. Such information on forest structure might be derived from remote sensing.

## 5  Conclusions

The temperature sensitivity of wood production in temperate forests is influenced by forest structure and species diversity as our study showed. The species distribution index ($\Omega_{AWP}$) and forest height seems to be the most important forest properties influencing temperature sensitivity.

Temperate forests that benefit most from temperature rise are those which consist of even-aged deciduous pioneer species in the case of young forests; [c5]mature forests benefit most if tree height heterogeneity is large and the forest includes different deciduous climax species.

This study [c6]also attemps to explain why certain forests types will decrease their productivity and others not. Our findings highlight the importance of forest structure for future studies investigating wood production under climate change.

*Data availability.* [c7]The R-workspace which includes the dataset of the analysed forests "foreststands" and the calculated SI values "SIValues" [c8]can be found in the online supplement.

---

[c8] here presented

[c9] *Text added.*

[c10] also

[c11] the sensitivity of forest systems against climate change

[c1] *Text added.*

[c2] in the past

[c3] DGVMs

[c4] on

[c5] M

[c6] tries also

[c7] In the online supplement you find the a

[c8] *Text added.*

## Appendix A: Additional information regarding methods and validation

### A1  Climate data

The <sup>c1</sup>construction of the 320 climate time series is based on measured climate time series of the eddy-flux station Hainich in Central Germany (Knohl et al., 2003) for the years 2000-2004 (Fig. A1). Mean annual temperature of these five years does not correlate with the annual precipitation sum, nor with the mean annual radiation (Fig. A2). Radiation and precipitation within these years correlate quite well (Pearson's r = 0.73).

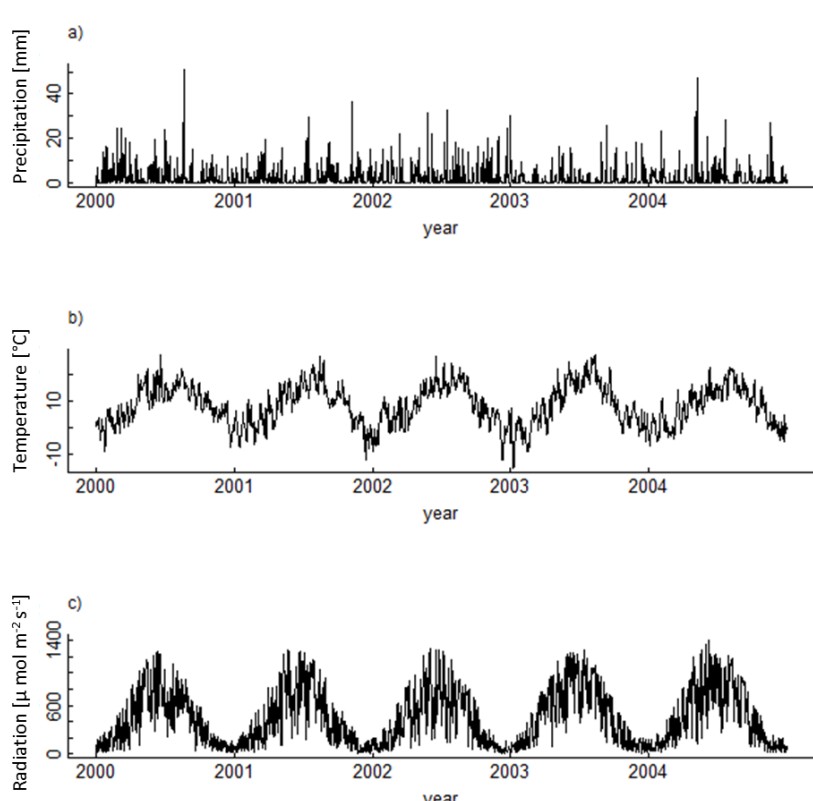

**Figure A1.** The climate time series measured at FLUXNET station Hainich from 2000 to 2004 which are used to generate the 320 climate time series: (a) daily precipitation [mm], (b) daily air temperature [% °C $^{-1}$], (c) daily incoming radiation [photoactive photon flux density $\mu\,mol\,m^{-1}s^{-1}$].

---

<sup>c1</sup> ~~development~~

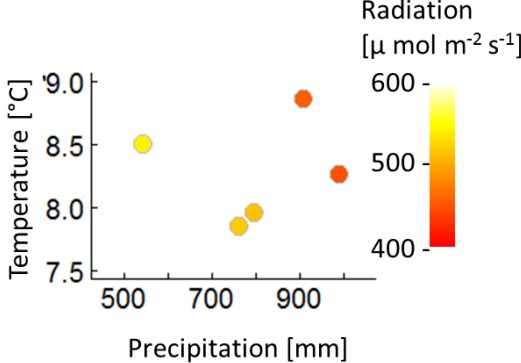

**Figure A2.** Mean annual temperature, annual precipitation sum and mean annual radiation of the five climate time series measured at Hainich station from 2000 to 2004.

## A2   Forest properties

We use three forest properties to describe forest structure (tree height heterogeneity ($\theta$), forest height class [c1] and LAI) and two properties to describe species diversity (Rao's Q describes functional diversity and [c2]the species distribution index $\Omega_{AWP}$ [c3]). The calculation of Rao's Q is based on 12 species-specific parameters which are relevant for productivity [c4](AWP) and [c5] species abundance (based on crown area). None of the properties correlate (table A1).

**Table A1.** Coefficient of Determination ($R^2$) between all used internal forest properties for 370,170 stands of the forest factory. $\theta$ = tree height heterogeneity; $H_{max}$ = forest height; LAI = leaf area index; $\Omega_{AWP}$ = species distribution index

| Variables | Rao's Q | $\theta$ | forest height class | LAI |
|---|---|---|---|---|
| $\Omega_{AWP}$ | 0 | 0.02 | 0 | 0.2 |
| LAI | 0 | 0.23 | 0.06 | |
| forest height class | 0.01 | 0.2 | | |
| $\theta$ | 0.02 | | | |

---

[c1] *$H_{max}$*
[c2] *Text added.*
[c3] ~~describes suitability~~
[c4] *Text added.*
[c5] ~~the~~

## A3  Validation with the German forest inventory

We analysed the influence of forest structure on temperature sensitivity within the German forest inventory[c1](beech monocultures and spruce monocultures). Tree height [c2] [c3]was used to calculate forest height [c4]$H_{forest}$ and tree height heterogeneity ($\theta$). We replaced LAI, which is not measured, by basal area (both properties correlate quite well in the forest factory data set; $R^2 = 0.74$). The forest stands of each species were classified into six structure classes: three [c5]classes which are based on the height of the largest tree in the forest stand (10-15 m, 20-25 m and 30-35 m), and two classes representing different tree height heterogeneities (0-1 and >1.6 m). [c6]Only plots that are located on flat terrain (slop[c7] less than 15 %) and have a maximum dbh of 0.5 m) [c8]were analysed.[c9]A linear model [c10]was fitted to the data of every class using basal area and elevation as input variables to predict above-ground wood productivity (AWP).

---

[c1] . We analyzed forest stands of beech monocultures and spruce monocultures.

[c2] data

[c3] are

[c4] ($h_{max}$)

[c5] forest height

[c6] We analyse o

[c7] ed at

[c8] *Text added.*

[c9] We fit a

[c10] *Text added.*

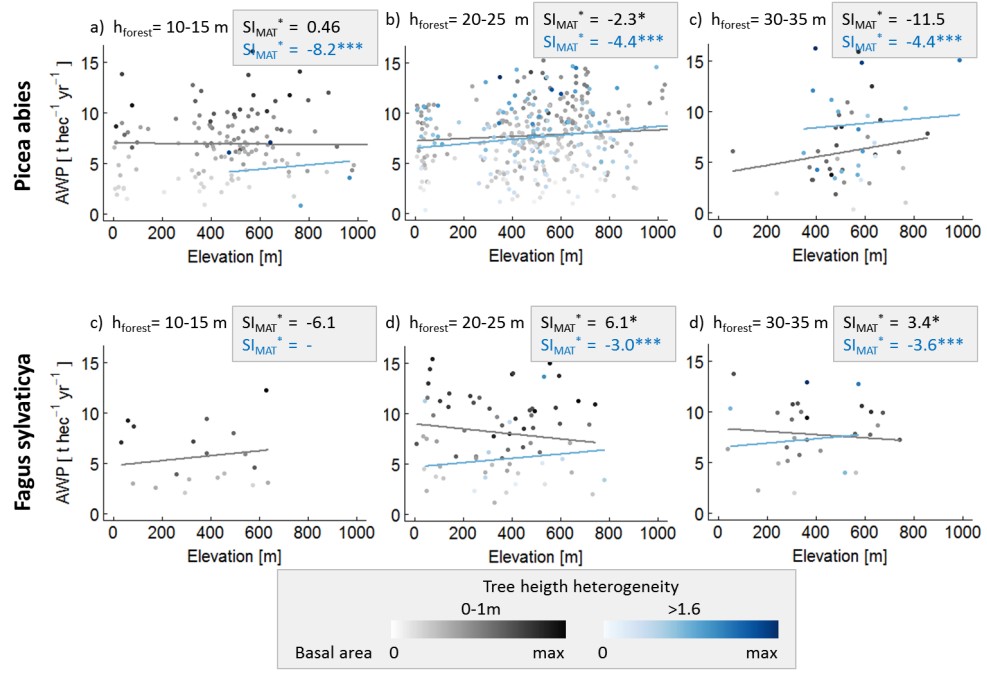

**Figure A3.** Analysis of the influence of forest structure on the relationship between elevation and above-ground wood production [c11] (AWP). Figures (a) - (c) are based on spruce monocultures and d)-e) [c12] on beech monocultures. For each species, forest stands [c13] were classified into three forest height classes which [c14] were based on the largest tree ($H_{forest}$) in a forest stand. These forest stand classes are additionally separated into two tree height heterogeneity classes (0-1 m in grey and >1.6 m in blue). Intensities of the colours indicate the ratio between basal area of the stand and maximal basal area found within one class. Lines show the results of the linear model with mean basal area. The amount of stars behind the SI values indicates the significance of the slope within a linear model: (***) indicate a p-value below 0.001 and (*) indicates a p-value between 0.01 and 0.05. No star indicates p-values above 0.1. The unit of $SI^*_{MAT}$ is % °C$^{-1}$ .

## Appendix B

### B1 Frequency distribution of sensitivity values

The analysed forest stands show a large range of temperature sensitivit[c1]y levels, which reach up to 8.5 % °C $^{-1}$ in case of $SI_{MAT}$ (Fig. ref4Bf1a). This means that [c2]a forest increases its productivity by 8.5 % due to an increase [c3]in the mean annual temperature [c4]of one °C. In case of the annual temperature amplitude, the best forest reduces its productivity by -0.5 % $°C^{-1}$ (Fig. ref4Bf1b). The mean $SI_{MAT}$ is 1.5 % °C $^{-1}$ and the interquartile range (iqr) ranges from 1.6 % °C $^{-1}$ to. 5.2 % °C $^{-1}$. The mean $SI_{Q95}$ is -5.4 % °C $^{-1}$ and the iqr ranges from -5.2 % °C $^{-1}$ to -2.2 % °C $^{-1}$.

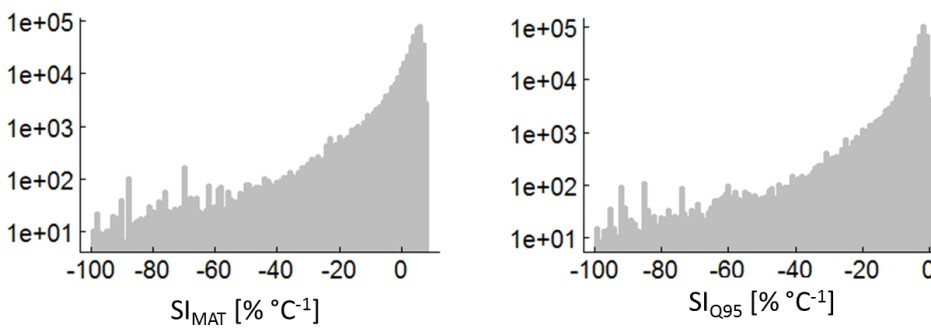

**Figure B1.** Frequency distribution of $SI_{MAT}$ values (a) and $SI_{Q95}$ values (b) of all forest stands.

### B2 Analysis with boosted regression trees

Boosted regression trees provide information about the underlying relationship between input variables (here forest properties) and output variables (here SI values). Several techniques were developed to visualise and interpret the high-dimensional relationship of input and target variables (Friedman, 2001). The comparisons between SI values of the forest factory and predicted SI values (based on the five properties as input), show[c5]ed a very high agreement (Fig. B2 & B3). The o[c6]btained vertical patterns for $SI_{Mat} = 0$ % °C $^{-1}$ and $SI_{Q95} = -6$ % °C $^{-1}$ are probably artefacts of the boosted regression tree algorithm.

Other commonly used visualization of the relationship of input and target variable are partial dependency plots (Fig. 4). These plots show the influence of an input variable on the target variable considering the influence of all input variables which

[c1] ~~ies~~
[c2] ~~one~~
[c3] ~~of~~
[c4] ~~by~~
[c5] *Text added.*
[c6] ~~p~~

have higher relative importance. In our study, the most important variable is $\Omega_{AWP}$, hence the first plot shows the relationship between suitability and SI values. The second relationship (forest height on SI values) [c7] based on the residuals of the first relationship (here between SI values and $\Omega_{AWP}$; Becker et al., 1996). Although a collection of such plots can seldom provide a comprehensive analysis of the [c8]boosted regression trees, it can often produce helpful hints, especially if variables show very low correlations, as in this study.

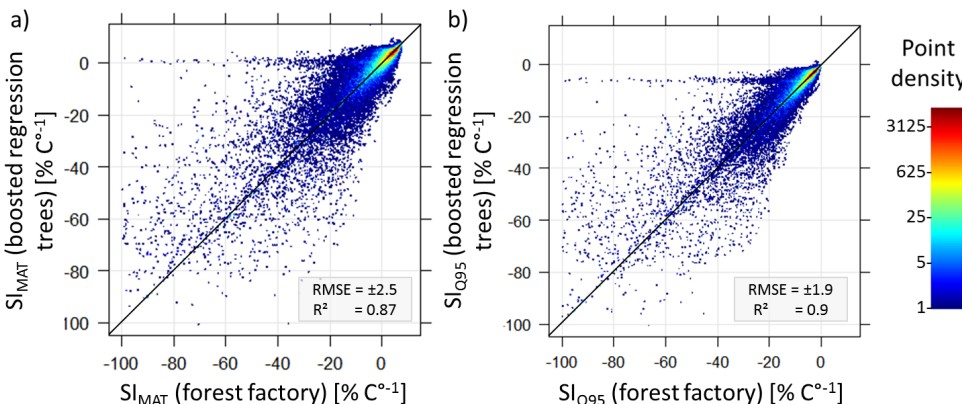

**Figure B2.** Comparisons of temperature sensitivity ($SI_{MAT}$ and $SI_{Q95}$) based on the forest factory and boosted regression tree model. Colours indicate point density. Diagonal is the 1:1 line.

---

[c7] is

[c8] BRT

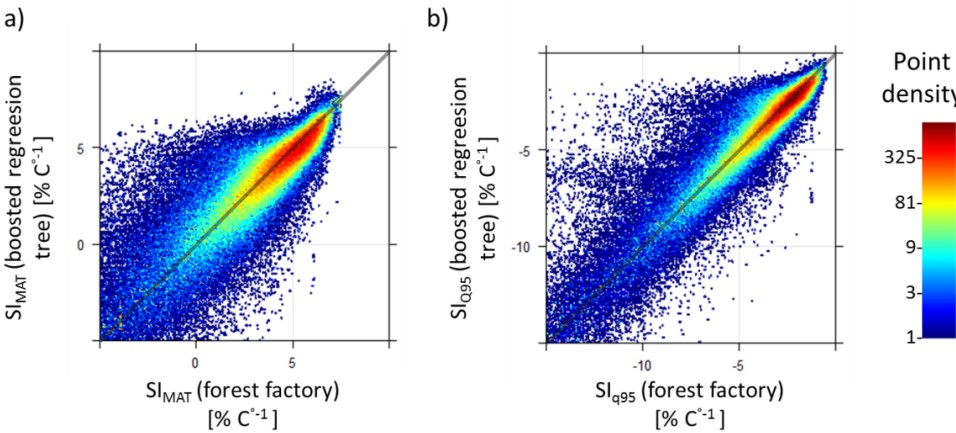

**Figure B3.** Comparison of temperature sensitivity calculations ($SI_{MAT}$ and $SI_{Q95}$) based on the forest factory and boosted regression tree model. Colours indicate point density. Diagonal is the 1:1 line. a) Contains 90% of the forest factory data set and b) contains 93% of the forest factory data set.

## B3    Forest stands properties with highest $SI_{Q95}$ values over a forest height gradient

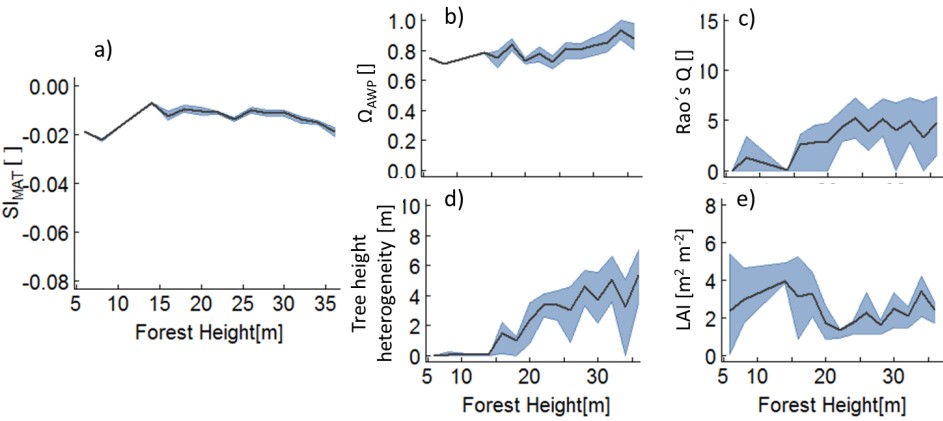

**Figure B4.** Analysis of those forests which lie above the 95% percentile of $SI_{MAT}$, depending on forest height $H_{forest}$. Lines indicate mean values of the subsamples and the grey bands indicate the inter quartile range. Figure a) shows the temperature sensitivity of productivity [c1]to forest height, analysing only [c2]values above the 95% percentile b) to d) shows the change of the remaining forest properties within the subsamples.

## B4    SI values of single trees

To understand the origin of the SI values, we make the following [c1]assumptions: An increase of 1 % °C [-1] always results in an increase of 8.6 % of the respiration rate in the model (Fig. B5 b; Piao et al. (2010)). The positive effect of a[c2] temperature increase of 1 % °C [-1] on the photosynthesis rate varies between the years due to the assumed species-specific bell-shaped
5  relationship (Fig. B5 a). In case of deciduous trees the length of the vegetation period (leaf onset to fall) [c3]additionally affects the annual photoproduction (e.g., Haxeltine and Prentice, 1996; Luo, 2007; Horn and Schulz, 2011; Gutiérrez and Huth, 2012; Sato et al., 2007). If the photosynthesis rate is much larger than the respiration rate (high AWP; for instance, low ratio of maintenance respiration to photosynthesis [c4] under full light in Fig. B6 b), the positive effect of temperature on photosynthesis causes an increase of AWP in some simulated years. If both rates show the same magnitude ([c5]ratio of maintenance respi-
10  ration to photosynthesis under full light is closed to 1 in Fig. B6 b), higher temperatures increase respiration stronger than photoproduction (in most years), which result[c6]s in a decrease of AWP.

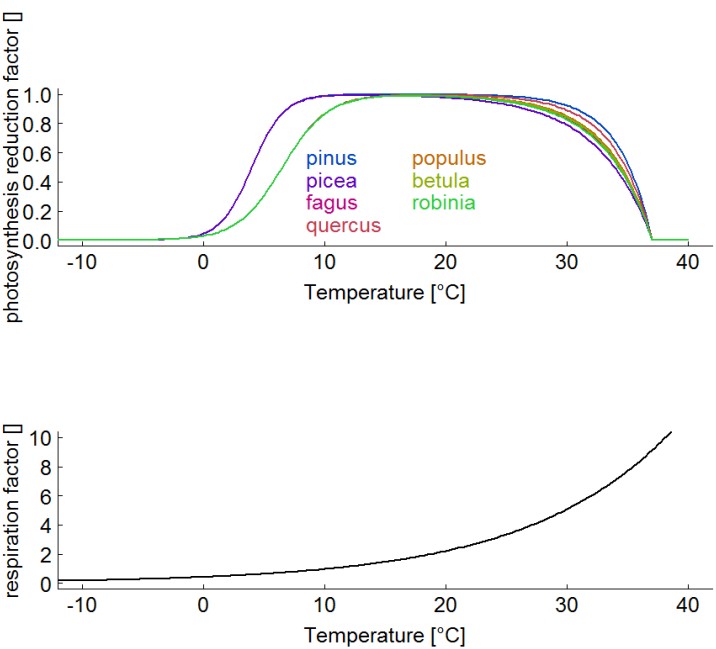

**Figure B5.** a) Species-specific reduction factor of photosynthesis due to a change in air temperature. b) Species-unspecific correction factor for maintenance respiration due to a change in air temperature.

---

[c1] ~~considerations~~
[c2] ~~n~~
[c3] ~~affects additionally~~
[c4] ~~(RMP)~~
[c5] ~~RMP~~
[c6] *Text added.*

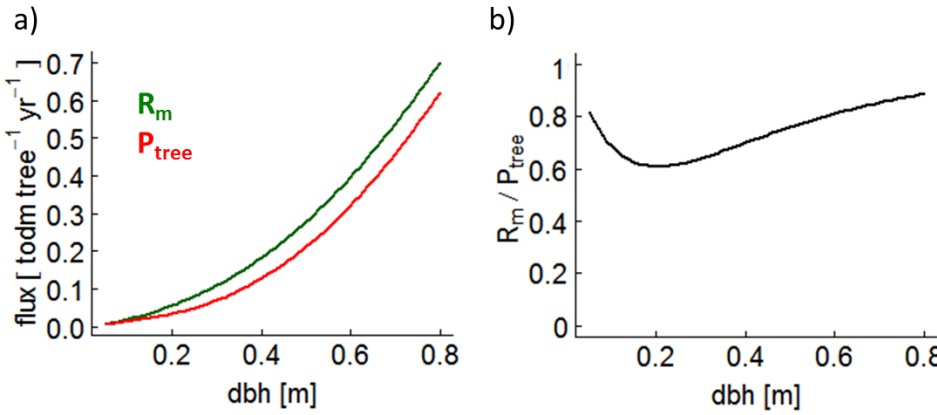

**Figure B6.** a) Photosynthesis (green [c7]line - $P_{tree}$) and maintenance respiration (red [c8]line - $R_m$) rates of a single beech tree over stem diameter (dbh) under full light. b) The ratio between maintenance respiration and photosynthesis of the same beech tree.

## B5  Functional diversity and temperature sensitivity

To analyse the effect of functional diversity on temperature sensitivity, we first calculate[c1]d [c2] $SI_{MAT}$ [c3] for every species depending on tree height and light availability (as done for pine trees in figure 6). Then, we calculate[c4]d a mean $SI_{MAT}$ value for each species mixture for all light-height combinations [c5]. Finally, we average[c6]d [c7]those SI-values which are larger than -7.5 % °C [-1] ($SI_{\overline{MAT}}$) and calculate the Rao's Q of the mixtures (based on equal abundances). The highest $SI_{\overline{MAT}}$ values were found for deciduous forests (Fig. B7). Mixed forests with deciduous and needle leaf trees show[c8]ed lower values than the deciduous forests, but higher Rao's Q-values.

[5]

---

[c1] *Text added.*

[c2] ~~the~~

[c3] ~~values~~

[c4] *Text added.*

[c5] ~~($SI_{h,t}$)~~

[c6] *Text added.*

[c7] ~~all $SI_{h,t}$~~

[c8] *Text added.*

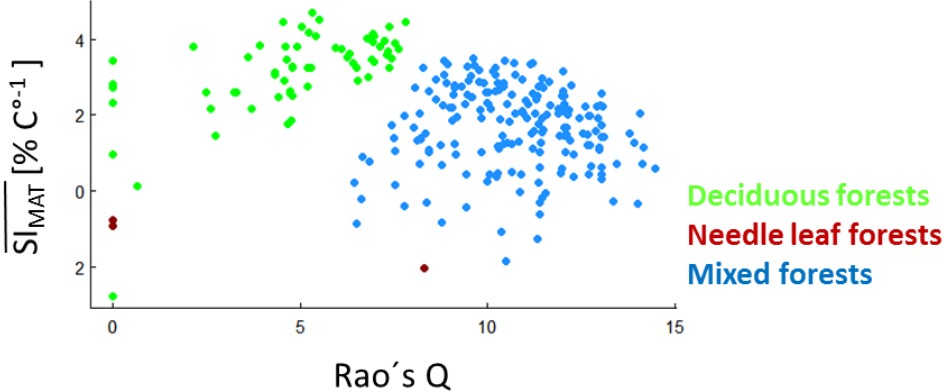

**Figure B7.** Rao's Q (with equal abundances) against $\overline{SI_{MAT}}$ values of all possible species mixtures (from the forest factory). The $\overline{SI_{MAT}}$ values are the average over all $SI_{MAT}$ values for all light-height combinations and with values larger than -7.5 % °C$^{-1}$. For mixtures, we assume[c9]d equal abundances and calculate the mean over the $SI_{MAT}$ values of all species within the mixture. Green dots indicate forests that consist only of deciduous trees; red dots indicate forests that consist only of needle leaf trees; blue dots indicate forests that contain both tree types.

*Author contributions.* F.J.B. F.M. and A.H. conceived of the study. F.J.B. implemented and analysed the simulation model and wrote the first draft of the manuscript. A.H. and F.M contributed to the text. All authors gave their final approval for publication.

*Competing interests.* We have no competing interests.

*Acknowledgements.* We thank Edna Rödig, Franziska Taubert, Nikolaj Knapp, Rico Fischer and Kristin Bohn and for providing many helpful
5 suggestions and comments. We also thank the Department of Bioclimatology of the University Göttingen and the Max Planck Institute of Biogeochemistry for providing climate data and the administration of Hainich National Park for permission to conduct research there.

**References**

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
