# Peer review of "Species composition and forest structure explain the temperature sensitivity patterns of productivity in temperate forests."

_Biogeosciences, 2017_

## Referee Comment (RC1) · C.P.O. Reyer (Referee) · 4 Oct 2017

I am sorry for being late. The manuscript uses an innovative modelling approach to assess the temperature sensitivity of above ground wood production. It is generally well written but there are quite a number of minor fixed that still need to be carried out. I have two major concerns that can be addressed in a thorough minor revision:

1) I think your "climate scenarios" are actually rather a "climate sensitivity test". Even though many things can be scenarios in the wider sense in the narrow sense, a scenario usually refers to an internally consistent projection while your scenarios systematically explore change temperature but do not adjust precip and radiation accordingly.

This may lead to physically inconsistent "climates" because under a given temperature pathway it might be impossible to get a certain precip or radiation behavior. Related to that, you should possibly also discuss that your approach basically ignores transient responses and time lags longer than a year that influence forests. So i think this "sensitivity test" aspect should be carved out more clearly. The title actually is fine but some of the other sections give the impression this is rather a scenario study.

2) I think, even though refering to central European, temperate forests, you are not putting enough emphasis on the interpretation of the results from a forest management perspective. However, your target variable "above ground wood production" as well as the stand densities, species choices etc. are subject to forest management and species mixing etc. are important elements of EU silvicultural strategies. I think the discussion of the influence of forest management on your results and their implications for forest management should be strengthened.

P1L1: Either "Observational studies discovered that" or "Observations show that..."

P1L4: "increase productivity the most"

P1L6: unclear: "within the forest structure" ==> don't you simply mean "within the forest given the environmental conditions of each..."?

P1L8: "cover a wide range of possible"

P1L11 and also L14: "increasing OmegaAWP" it sounds weird that an optimum can be increased or have "large values". Maybe I am too picky and I do not have a good alternative... maybe ask a native speaker...

P1L15: "heterogenity is associated with a positive"

P1L14-16: This sounds like quite a contradiction: for young forests low diversity and low height spread make the forest react positively to temperature while for older forests this is not the case. Could you add one sentence of explanation here and discuss potential implications for forest management in the discussion? This would mean foresters

should go for even aged, mono-species stands during establishment and then bring in other species later? or keep the canopy closed with one species while having other species in the undergrowth for some time?

P1L19: I have the feeling you are using "forest growth", "wood production" and "forest productivity" interchangeably. While this can be correct in some instances I wonder whether all the references you cite here actually refer to forest growth or rather productivity.

P1L21-P2L2: I think this sentence is imprecise. the observed changes in productivity are not the primary reason for discussing the compensation of co2. it is rather the overall high carbon stocks and sequestration rates (even without changes) that matter for this discussion. I see what you want to say but I think it is a bit too condensed here....

P2L3-9: I think this section should also mention the influence of other factors, at least briefly... Especially since you say that "productivity is influenced by several factors" in the first sentence of this paragraph...

P2L18/19: "rarely include properties related to both species composition. . ."

P2L22: "forests stands were available, it would"

P2L23.: "option to such field experiments is"

P2L26: "simulating 30 year time slices of a range of different future climates for 135..."

P2L27: "analyzed"

P2L29: "a large number"

P2L30: "species compositions"

Figure 1: I would delete the "..." in each box as your study does not cover more climate variables nor more stand structural or composition related variables. In the

caption, I would precise: "overview of drivers influencing forest forest productivity in this study" and also clearly state that only temperature is varied in a "temperature sensitivity analysis" or so.

P3L3: "2017). The forest factory generates 370,170... and allows to estimate"

P4L2: How to get from the 15 stem size distributions and 256 mixtures to the 370,170 stands?

P4L3/4: At some point you should give the latin names of the species to allow international readers to check which species you mean...

P4L6: I wonder how you can actually represent complex mixtures on a 400m$^2$ plot. This could be covered by one large beech tree? I think you need to discuss the implications of choosing this patch size. Or do you upscale to the ha or so?

P4L6: "space limits"

P4L8: You should discuss in detail why you think the year 2007 in Hainich is representative of temperate climatic conditions! I think you make two dangerous assumptions: 1) Hainich is somehow representative for "temperate climates" (it certainly is but only to a certain degree... and 2) the 2007 climate is somehow representative of the overall Hainich climate...

P4L10: "2.2 Forest productivity..."

P4L11-22: This is from FORMIND, right? You could say that...

P4L15: "by the photosynthesis-limiting..."

P4L17/18: I read this as if Rm was both maintenance respiration and allocation to non-woody tissues?

P4L23: I would introduce a new subheading here about the "climate sensitivity"

P4L25: "separated" The methods description should be in the past tense

Figure2: You should explain once more all the variables shown on the plots in the caption, AWP and MAT are not explained currently.

P5L1: unclear: do you include co2 and nitrogen in the model but keep them constant or are they not included at all? You should discuss that co2 will matter in a 2°C warmer world. . .

P6L6: Do you have any reference or argumentation to support using BA as a proxy for LAI? I could imagine this only works until canopy closure?

P6L13: "maximum forest height"

P7L1-2: Is "10.2015" the right citation format?

P7L5: Why Gaussian? Any deeper reasons or simply because it is the default?

P7L8: the heading is unclear. "benefit the most" ==>from what?

P7L18-19: I wonder if the mean is the appropriate measure here given that the distributions are so skewed (figure b1)?

P7L28: This has to be carefully discussed. It seems to be obvious that the species choice will have the strongest influence.

Figure 3: maybe explain somewhere (can be in the main text) how to interpret the scale from 0-1 of the omegaAWP

P8L12: "analysed how"

P8L12: Maybe recap here that AWP is your expression of productivity.

P9L1: "specific value combination of forest properties" ==> rephrase

P9L5/6 & 12-14: Certainly analyzing all possible combinations of species and structures etc as done in the Forest Factory approach is valuable but this approach will also generate a huge number of stands which are highly unrealistic and that will never be found in reality. So the discussion could be more balanced here highlighting that you

also produce quite a lot of "non-sense" forests as well.

P9L10: "FORMIND"

P10L2-3: This sentence needs to be rewritten for clarity

P10L11: RCP2.6

P10L15-28: I had the feeling this section needs to be rewritten for clarity and logical connection to the preceding sections.

P11L3-12: Also here maybe some rewriting is needed to better link the paragraph to the rest of the discussion.

P13L5: I find the conclusions too short and too close to the results. The conclusion in my view should clarify: Why do your results matter? What do we learn?

P13L12: I find the supporting material organized in a complicated way. Can you not provide the text plus the associated figures and then another piece of text etc. The online material is not meant to be read as one text but one should find things quickly.

Figure A2: I am pretty sure you are showing the annual precipitation sum and not the "mean" here.

P15L4/5: I do not understand this: it seems mean (1.5%) is outside of the interquartile range?

Figure A3: hec should be ha

P16l5: "the first plot" ==>not very precise. Do you mean plot 3a)? Then also include small letters in the plots!

P16L6: I would always refer to omegaAWP and not introduce any other terms such as suitability etc. it is getting too complicated. . .

Figure A4: To me it looks like You are overestimating SI MAT quite systematically but you never really discuss this?

P17L6: sentence misses a verb

Figure b3: I wonder why there are so distinct patterns of values with y=0 and also x=-100 in panel a and y=-8 (or so) and x=-100. Is this an artifact?

P18L2: "in some simulated"

P18L7: "we calculate a mean..."

Figure B5: panel "b)" is missing the "b)"

Figure B6: The layout of this figure makes it very hard to see which color overlays the other... so do most "brownish" species rather follow the green line or the dark blue for the left-side of the bell-shaped lines in panel a)?

Figure B7: Is nowhere referred to in the text. What is the unit of the y-axis?

Figure B8: The SIMAT values are so small, is that correct?

---

## Referee Comment (RC2) · Anonymous Referee #2 · 26 Oct 2017

**GENERAL COMMENTS**

The authors use an individual-based forest modelling approach to isolate the effect of five forest structure (LAI, maximum stand height, and canopy stratification) and composition (functional diversity and its optimal distribution) parameters on the overall sensitivity of simulated wood production to increasing mean annual temperature and seasonal amplitude in a European temperate setting. The model is integrated hundreds of times over a single year and forest stand using synthetic climate scenarios with perturbed temperature to simulate an ensemble of productivities for thousands of stands with a range of initial conditions of structure and composition, representative of different

stand development stages. This synthetic dataset, broadly consistent with ecological rules observed in the real world, is then analysed using a statistical regression method in order to quantify the relative effect of the five structural and compositional parameters on the simulated temperature sensitivity of forest productivity. Their results indicate that an index of optimal species distribution of the trees in a forest stand – the ratio of actual productivity of a forest stand to the maximum productivity achieved by changing only the species of the trees whilst keeping the same stand structure – explains ∼88% of the temperature sensitivity of forest productivity simulated by the model. Among the remaining four parameters, forest height (a proxy of stand development stage) is the most important variable in explaining the rest of the fractional variance of temperature sensitivities. Thus the authors conclude that the sensitivity of plant productivity in temperate forests to changes in temperature is driven by forest structure and species diversity.

I believe that the main scientific finding is of interest for the wider biogeosciences communities. The overall modelling experiment seems appropriated to disentangle the relative importance of forest structure and composition properties on patterns of temperature sensitivity of temperate forest productivity in the model world. However, the manuscript is very difficult to follow at times and the discussion of the main findings is simply too thin. I recommend major revisions in order to improve (i) the readability and English of the manuscript and (ii) to better discuss the underlying mechanisms and implications of the findings for the wider ecological theory.

SPECIFIC COMMENTS

Regarding the first point above, I have the following comments:

-Introduction It is informative but the English and grammar need revision.

-Method This section tends to be redundant with the opening paragraph and appendices. I recommend reorganising it and make it more concise. It is not clear how the stands at different stages of development were initialised. Were the structure, composition and development stage randomly generated or did you apply some spin-up? This has implications for the realised productivity when computing the $\Omega$_AWP index.

Why exactly 370,170 stands?

In Line 28 of page 4 it is said "We end up with five climate scenario sets of one-year length that differ in precipitation and radiation." But since these scenarios are derived from five different real years, they should also differ in absolute temperature values? As I understand it, within each synthetic scenario, radiation and precipitation are the same and only temperature changes at the specified steps. Among the five scenarios, the absolute values of all variables should be different. Please explain better this part to the reader.

In Fig. 2, what does the shading in the middle panels mean? The meaning of H_forest is not explained in the caption.

-Results and Discussion I found many passages in these sections very difficult to follow. There are result statements with no reference to the figures or tables that leave the reader guessing the corresponding figures. The results section is rather short and most of the discussion is still results.

Figure B1 shows a long negative tail in the distribution of the obtained sensitivities. The authors focus exclusively on the positive sensitivities and neglect negative values, despite declaring in the introduction that responses can be both positive and negative. Why the simulated sensitivities are so asymmetrical and the negative values are not discussed?

In the discussion I miss a more complete explanation of the underlying ecophysiological and metabolic mechanisms (e.g., Figs 5 and 6). Also, there is no discussion of the potential limitations of the model and the modelling experiments performed here. For instance, autotrophic respiration seems to be a critical factor affecting the response of net productivity to changes in temperature in the model. How well is this process

represented in the model version used? Would you get the same result if you account for uncertainties in parameterisation of this process?

Finally, what are the implications of the main finding for the wider ecosystem and climate modelling communities that usually rely on global models that have no explicit forest structure? Any recommendation?

-Conclusion Rather brief. Here the authors could wrap-up the wider implications of their main findings.

-Figures The figures are excellent but the captions are not sufficiently informative. Please improve the captions. Fig 3 could be merged with Fig B2. The latter is important to understand the overall result.

-English and grammar There are many grammar errors and typos through the text. Please revise English and correct typos.

---

## Author Comment (AC1) · 22 Nov 2017

Comments of reviewer 1 in blue, reply in black. Note that Appendix R is an online supplement of this reply.

*I am sorry for being late. The manuscript uses an innovative modelling approach to assess the temperature sensitivity of above ground wood production. It is generally well written but there are quite a number of minor fixed that still need to be carried out.*

Thank you very much for your detailed and helpful comments. We will include your recommendations carefully.

*I have two major concerns that can be addressed in a thorough minor revision: 1) I think your "climate scenarios" are actually rather a "climate sensitivity test". Even though many things can be scenarios in the wider sense in the narrow sense, a scenario usually refers to an internally consistent projection while your scenarios systematically explore change temperature but do not adjust precip and radiation accordingly. This may lead to physically inconsistent "climates" because under a given temperature pathway it might be impossible to get a certain precip or radiation behavior.*

Thanks for this comment. We agree. We will replace "climate scenarios" by "climate sensitivity test" or "climate time series" depending on the context. For instance, P1L9: For each stand we estimate annual above-ground wood production and perform a climate sensitivity test based on 320 different climate time series (of one year length).

*Related to that, you should possibly also discuss that your approach basically ignores transient responses and time lags longer than a year that influence forests. So i think this "sensitivity test" aspect should be carved out more clearly. The title actually is fine but some of the other sections give the impression this is rather a scenario study.*

This is a good point. Thank you. We will replace the sentences between P15L12 and P15L15 with the following text:

. . .This RCP scenario predicts only small changes of annual precipitation levels for many temperate regions. However, our approach analyzes only impacts of change in temperature. For instance, we do not simulate the effect of changing $CO_2$ in combination with an increase of temperature. This might be critical for analysis of strong temperature changes (e.g. RCP 8.5), which will result in an increase of droughts and changes in the annual temperature cycles.

Such more complex scenarios should be analyzed in future studies. Further, we neglect the effect of time lags (e.g. the bud building in the autumn in the previous year). However, it is possible to enlarge the used time series to analyze the behavior of the forest over longer time periods and study not only productivity, but also effects on regeneration or mortality.

*2) I think, even though referring to central European, temperate forests, you are not putting enough emphasis on the interpretation of the results from a forest management perspective. However, your target variable "above ground wood production" as well as the stand densities, species choices etc. are subject to forest management and species mixing etc. are important elements of EU silvicultural strategies. I think the discussion of the influence of forest management on your results and their implications for forest management should be strengthened. We agree that our result might be important for forest management strategies.*

We therefore will add a new section "4.4. Implications for forest management" in the Discussion section (see Appendix R 2).

Thank you for these detailed smaller points.

*P1L1: Either "Observational studies discovered that" or "Observations show that..."*

*P1L4: "increase productivity the most"*

*P1L8: "cover a wide range of possible"*

We will reformulate the phrases.

*P1L6: unclear: "within the forest structure" ==> don't you simply mean "within the forest given the environmental conditions of each..."?*

We will write: ... how well species are distributed over the different forest layers regarding...

*P1L11 and also L14: "increasing OmegaAWP" it sounds weird that an optimum can be increased or have "large values". Maybe I am too picky and I do not have a good alternative... maybe ask a native speaker...*

We will introduce a new term for $\Omega_{AWP}$: "species distribution index" but keep the acronym $\Omega_{AWP}$.

*P1L15: "heterogenity is associated with a positive"*

We will reformulate the sentence, following your suggestion.

*P1L14-16: This sounds like quite a contradiction: for young forests low diversity and low height spread make the forest react positively to temperature while for older forests this is not the case. Could you add one sentence of explanation here and discuss potential implications for forest management in the discussion? This would mean foresters should go for even aged, mono-species stands during establishment and then bring in other species later? or keep the canopy closed with one species while having other species in the undergrowth for some time?*

We recommend paying attention to young forests with low diversity and even-aged structure. A mixture of climax species should be planted below the main canopy. Later those climax species will replace the pioneer species in the canopy and build the mature forest with heterogeneous tree sizes.

We will add a new section regarding the implications on forest management (Appendix R1).

*P1L19: I have the feeling you are using "forest growth", "wood production" and "forest productivity" interchangeably. While this can be correct in some instances I wonder whether all the references you cite here actually refer to forest growth or rather productivity.*

We will focus on (above-ground) wood production in the whole manuscript. Barber et al. 2000 analyzed the change in radial growth of white spruce under climate change. We will replace the references of Cao and Woodard 1998 (as it refers more to carbon fluxes of ecosystem in general and not explicit wood production) by Reyer et al. 2013. The other two refer to the effect of climate change on photosynthesis or respiration of

trees (Luo 2007) or plants (Penuelas and Filella 2009).

*P1L21-P2L2: I think this sentence is imprecise. the observed changes in productivity are not the primary reason for discussing the compensation of co2. it is rather the overall high carbon stocks and sequestration rates (even without changes) that matter for this discussion. I see what you want to say but I think it is a bit too condensed here....*

We will reformulate the sentences:

Changes in forest productivity have been observed in past decades all over the world (Nemani et al., 2003; Boisvenue and Running, 2006; Seddon et al., 2016). The carbon stock of forests and their role as carbon sink are therefore in danger. These findings stimulated discussions about whether forest management strategies can be adapted to reduce forest vulnerability to climate change, to support recovery after extreme events and foster the carbon sink function of forests.

*P2L3-9: I think this section should also mention the influence of other factors, at least briefly... Especially since you say that "productivity is influenced by several factors" in the first sentence of this paragraph...*

We will add the following sentences:

. . . in addition to other climate variables (Barford et al. 2001). For instance, increasing $CO_2$ increases water use efficiency of forests (Keenan et al. 2013), which could compensate negative effects of climate change on European forest growth (whereas, with constant $CO_2$ at 350ppm, forest growth declines on several sites due to climate change - Reyer et al. 2014). Another important often investigated process is the fertilization effect of nitrogen (De Vries et al., 2006, 2009). For instance due to depositions in the second half of the last century, wood production had increased in European forests (Solberg et al. 2009). In case of temperature change, photosynthesis, respiration and growth rates are modified. . .

*P2L18/19: "rarely include properties related to both species composition..."*

*P2L22: "forests stands were available, it would"*

*P2L23.: "option to such field experiments is"*

*P2L26: "simulating 30 year time slices of a range of different future climates for 135..."*

*P2L27: "analyzed"*

*P2L29: "a large number"*

*P2L30: "species compositions"*

*Figure 1: I would delete the "..." in each box as your study does not cover more climate variables nor more stand structural or composition related variables. In the caption, I would precise: "overview of drivers influencing forest forest productivity in this study" and also clearly state that only temperature is varied in a "temperature sensitivity analysis" or so.*

*P3L3: "2017). The forest factory generates 370,170...and allows to estimate"*

We will realize all these suggestions. Thank you.

*P4L2: How to get from the 15 stem size distributions and 256 mixtures to the 370,170 stands?*

The forest factory creates forest patches which are based on 15 different stem size distributions and 256 species mixtures. 100 forests patches of each combination are built (in total 15*256*100 = 384,000 forests). In a few cases not all species of the mixture could be placed within a patch by the algorithm, so these forests are rejected. We end up with 370,170 forest stands. We will reformulate section 2.2 of the Method section to clarify these points.

*P4L3/4: At some point you should give the latin names of the species to allow international readers to check which species you mean...*

Good point. We will add the Latin names in the section 2.2.

*P4L6: I wonder how you can actually represent complex mixtures on a 400m2 plot. This could be covered by one large beech tree? I think you need to discuss the implications of choosing this patch size. Or do you upscale to the ha or so?*

Within an area of 400m$^2$ the trees compete for light as large trees shade the smaller ones. This is the typical plot size used in forest gap models. You are right, a very large tree could cover such a plot but below its crown there is space for smaller trees of other species. However, there are indeed a few combinations of species mixtures and stem size distribution that could not be represented within a plot because of limited space. Thus, these forests do not exist in the collection of forests generated by the forest factory.

We will reformulate the text in section 2.2. to clarifying these points.

*P4L6: "space limits"*

We will reformulate the text.

*P4L8: You should discuss in detail why you think the year 2007 in Hainich is representative of temperate climatic conditions! I think you make two dangerous assumptions: 1) Hainich is somehow representative for "temperate climates" (it certainly is but only to a certain degree and 2) the 2007 climate is somehow representative of the overall Hainich climate*

We agree. We use this year as an example of a temperate climate. (In principle, it is possible to use climate data of every other location). We will reformulate the text in section 2.2.

*P4L11-22: This is from FORMIND, right? You could say that*

We will add the following sentence at the beginning of the paragraph: The calculation of wood production of trees is based on algorithms of the model FORMIND (Fischer et

al. 2016,).

*P4L10: "2.2 Forest productivity..."*

*P4L15: "by the photosynthesis-limiting..."*

We will use your suggestions.

*P4L17/18: I read this as if Rm was both maintenance respiration and allocation to non-woody tissues?*

This is correct.

*P4L23: I would introduce a new subheading here about the "climate sensitivity"*

We will add it.

*P4L25: "separated" The methods description should be in the past tense*

We will change the tense.

*Figure2: You should explain once more all the variables shown on the plots in the caption, AWP and MAT are not explained currently.*

Good point. We will modify the caption of Figure 2.

*P5L1: unclear: do you include co2 and nitrogen in the model but keep them constant or are they not included at all? You should discuss that co2 will matter in a $2\,^{\circ}C$ warmer world...*

They are not included at all. We will modify the sentence. See also the reply to your first major comment above.

*P6L6: Do you have any reference or argumentation to support using BA as a proxy for LAI? I could imagine this only works until canopy closure?*

In the forest factory dataset LAI and basal area correlate quite well ($R^2$=0.74). A high correlation between leaf area and basal area ($R^2$>0.92) has been found for instance

also by Levi and Jarvis 1999.

*P6L13: "maximum forest height"*

We will replace the phrase.

*P7L1-2: Is "10.2015" the right citation format?*

No. We will correct the citation format to "2015".

*P7L5: Why Gaussian? Any deeper reasons or simply because it is the default?*

It is the default setting. We will modify the sentence: ... assuming a Gaussian error structure (default setting).

*P7L8: the heading is unclear. "benefit the most" ==>from what?*

We will add: ... the most from increasing temperatures.

*P7L18-19: I wonder if the mean is the appropriate measure here given that the distributions are so skewed (figure b1)?*

Figure B1 shows the distribution of the SI values. The shown SI values do not represent mean values; they are the slope of linear models, which relate temperature changes with AWP changes divided by the average AWP of the forest (see equation 2, 3 4). We will modify the sentence: We than quantified the changes in productivity due to changes in mean annual temperature ($SI_{MAT}$) and amplitude of inter-annual temperature ($SI_{Q95}$).

*P7L28: This has to be carefully discussed. It seems to be obvious that the species choice will have the strongest influence.*

Thanks for this point. We will modify the sentences at P11 L16ff: ... If species have an unfavorably distribution within the forest (low $\Omega_{AWP}$), the wood production (AWP) of the forest is low. Note, $\Omega_{AWP}$ is the ratio between current AWP and the highest possible AWP of the forest which can be reached due to shuffling of species identities. If AWP is

low the forest will suffer from increasing temperatures, which results in negative slopes ($\Delta$AWP / $\Delta$T). These values are than divided by low AWP values (Equation 4), which results in large negative values of $SI_{MAT}$ and $SI_{Q95}$. (See Appendix B5).

*Figure 3: maybe explain somewhere (can be in the main text) how to interpret the scale from 0-1 of the $\Omega_{AWP}$.*

We will add an explanation in the caption of figure 3.

*P8L12: "analysed how"*

We will reformulate the phrases.

*P8L12: Maybe recap here that AWP is your expression of productivity.*

We will replace forest productivity by (above-ground) wood production. See also reply to your comment P1 L19.

*P9L1: "specific value combination of forest properties" ==> rephrase*

We will write: . . .with a specific set of forest properties which. . .

*P9L5/6 12-14: Certainly analyzing all possible combinations of species and structures etc as done in the Forest Factory approach is valuable but this approach will also generate a huge number of stands which are highly unrealistic and that will never be found in reality. So the discussion could be more balanced here highlighting that you also produce quite a lot of "non-sense" forests as well.*

This is an important point of the forest factory approach. All forest patches could exist in reality, as every tree has a positive productivity and enough space for its crown. It is not possible that "non-sense" forests are generated as the used method prevents that a light demanding species occurs below a closed canopy or that forests are overcrowded. We will revise the text to makes this clearer (see Appendix R 5.6).

*P9L10: "FORMIND"*

We will replace the phrase.

*P10L2-3: This sentence needs to be rewritten for clarity*

We will remove this sentence.

*P10L11: RCP2.6*

We will correct this.

*P10L15-28: I had the feeling this section needs to be rewritten for clarity and logical connection to the preceding sections.*

Thanks for this comment. We will reformulate the section in the following way:

To characterize the annual cycles of temperature we select two variables: mean annual temperature (MAT) and inter annual temperature amplitude (Q95). Both variables can be varied independently. In case of higher MAT we observe an elongation of the vegetation period. This leads to higher forest productivity, if other resources are sufficiently available (Luo, 2007). This explains why $SI_{MAT}$ is often positive. However, warmer summer temperatures can also lead to a decline in wood production due to an increase in respiration. In case of increasing Q95, more days with extreme temperatures will occur in a year. Thus, an increase of one $°C-1$ of Q95 will increase respiration more strongly compared to an increase of one $°C^{-1}$ of MAT. Hence, the increase of Q95 has normally negative effects on the productivity (negative SI values).

*P11L3-12: Also here maybe some rewriting is needed to better link the paragraph to the rest of the discussion.*

We will modify the two paragraphs of section 4.2.

*P13L5: I find the conclusions too short and too close to the results. The conclusion in my view should clarify: Why do your results matter? What do we learn?*

We will extent the conclusion and include implications of our study for future forest

modelling and forest management. (See also Appendix R3: new conclusion)

*P13L12: I find the supporting material organized in a complicated way. Can you not provide the text plus the associated figures and then another piece of text etc. The online material is not meant to be read as one text but one should find things quickly.*

We will rearrange the text and figures as suggested.

*Figure A2: I am pretty sure you are showing the annual precipitation sum and not the "mean" here.*

Correct. We will revise the sentence.

*P15L4/5: I do not understand this: it seems mean (1.5%) is outside of the interquartile range?*

The unit is % $°C^{-1}$ . This means that on average forest increase their productivity by one % if temperature increases by $°C$ . The interquantile range of 95% (Q95) is used only to quantify the interannual temperature variability. We will modify the sentences to make this clearer:

*Figure A3: hec should be ha*

*P16l5: "the first plot" ==>not very precise. Do you mean plot 3a)? Then also include small letters in the plots!*

We will follow your suggestions.

*P16L6: I would always refer to $\Omega_{AWP}$ and not introduce any other terms such as suitability etc. it is getting too complicated...*

We will modify text and figure.

*Figure A4: To me it looks like You are overestimating SIMAT quite systematically but you never really discuss this?*

Good point. It is quite difficult to evaluate the theoretical analysis by using field data, as

a huge number of similar forest plots are needed, which cover a temperature gradient. In case of the German forest inventory only a few forest plots can be used for such an analysis. For spruce and beach monocultures we were able to select enough similar plots. Further, these inventoryplots provide no direct measurements of climate or LAI, which would be needed for an appropriate evaluation. We therefore used altitude as proxy for temperature (although other environmental variables like precipitation or soil attributes could change as well with altitude) and basal area as proxy for LAI. The reason for the overestimates of $SI_{MAT}$ is not clear, but might be a result of the differences between $SI_{MAT}$-calculations based on field data and the calculation of $SI_{MAT}$ based on the forest factory approach. We will add some sentences regarding this point to the Appendix.

*P17L6: sentence misses a verb*

We will revise the sentence.

*Figure b3: I wonder why there are so distinct patterns of values with y=0 and also x=-100 in panel a and y=-8 (or so) and x=-100. Is this an artifact?*

The vertical dots occur as all forests with SI values smaller than -100% $°C-1$ where set to -100% $°C-1$ for this graphic. To be consistent, we will remove them, also as they were removed for the final analysis, which is presented in the paper. See page 6, line 4 of the manuscript: ...In our analysis we exclude all forests stands for which AWP is negative if the temperature rises by 1 $°C-1$ ( 2% of all stands). I assume the horizontal structure is an artifact of the boosted regression tree algorithm. We will reformulate the corresponding sentences.

*P18L2: "in some simulated"*

*P18L7: "we calculate a mean..."*

*Figure B5: panel "b)" is missing the "b)"*

We will reformulate these phrases and add the "b)".

*Figure B6: The layout of this figure makes it very hard to see which color overlays the other ... so do most "brownish" species rather follow the green line or the dark blue for the left-side of the bell-shaped lines in panel a)?*

We will revise the figure based on a modified color-palette and enlarge the graphic.

*Figure B7: Is nowhere referred to in the text. What is the unit of the y-axis?*

It will be referred in the text and the text will be reformulated and we will revise the figure.

*Figure B8: The SIMAT values are so small, is that correct?*

We forgot conversion into % (multiplication by 100, as done in all the other graphics). We will also add the unit to the y-axes.

Please also note the supplement to this comment:
https://www.biogeosciences-discuss.net/bg-2017-335/bg-2017-335-AC1-supplement.pdf
* * *
[Figure]

**Supplement:**

**Appendix R**

**R 1 - 4.4. Implications for forest management.**

This new paragraph will be added as "4.4. Implications for forest management" at page 13 line 4:

Our findings might be relevant for future management strategies of temperate forests. Specifically, the analysis, which species benefit most from rising temperatures (Fig 6), suggests replacing spruce monocultures with mixtures of deciduous trees.

Further, based on the analysis of which forest structure benefits most from rising temperatures (Fig. 4, Fig 5, Fig 6), we would suggest that early stage even-aged forests include only pioneer species. In the mature stage, we predict a positive effect of temperatures on wood production for a mixture of climax species including different tree sizes. These climax species could be planted below the canopy of the pioneer species in young forests. In our approach, we do not simulate the establishment and initial growth of very young trees. This can be analyzed in future studies with our approach. However, during the conversion between these two forest types one big challenge might be the removal of the pioneer trees without damaging the young trees, which will build the mature forest.

**R2 - 4.5 implications for global vegetation modelling.**

This new paragraph will be added as "4.5 implications for global vegetation modelling" at page 13 line 4:

Most global vegetation models (DGVM) represent vegetation as fractional cover of different plant functional types within a grid cell (e.g. LPJ Sitch et al 2003). Only a few global vegetation models include a more detailed representation of vegetation structure and functional diversity (Sato et al. 2007, Scheiter et al. 2013, Sakschewsky et al. 2016). It would be interesting to perform the here presented analysis also with global vegetation models which include structure, to better understand the mechanisms driving the sensitivity of forest systems against climate change.

Beside the global vegetation models, forest gap models, which have been restricted to local stands in the past are now able to simulate regions or even entire continents (Seidl et al. 2013, Roedig et al. 2017). Many studies using DGVMs or large scale forest gap models simulate natural succession. Our analysis indicates that natural and managed or disturbed forest systems (which differ in forest structure) might react differently on climate change. Hence, we suggest considering forest structure in future analysis on forest systems. Such forest structure information might be derived for instance from remote sensing.

**R3 - enlarged conclusion:**

The new conclusion will replace the text on page 13 line 6-9:

The temperature sensitivity of wood production in temperate forests is driven by forest structure and species diversity as our study showed. The species distribution index ($\Omega_{AWP}$) and forest height seems to be the most important forest properties influencing temperature sensitivity.

Temperate forests that benefit most under temperature rise are (a) young forests including deciduous pioneer species with an even-aged structure and (b) old-growth forests with high tree height heterogeneity including different deciduous climax species.

This study tries also to give an explanation why forests suffer from rising temperatures in some cases and in others not. Finally, this work motivates the inclusion of forest structure and spatial distribution of species into future studies regarding wood production.

**Additional References:**

Reyer, Christopher, et al. "Projections of regional changes in forest net primary productivity for different tree species in Europe driven by climate change and carbon dioxide." *Annals of forest science* 71.2 (2014): 211-225.

Sitch, Stephen, et al. "Evaluation of ecosystem dynamics, plant geography and terrestrial carbon cycling in the LPJ dynamic global vegetation model." *Global Change Biology* 9.2 (2003): 161-185.

Sato, Hisashi, Akihiko Itoh, and Takashi Kohyama. "SEIB–DGVM: A new Dynamic Global Vegetation Model using a spatially explicit individual-based approach." *Ecological Modelling* 200.3 (2007): 279-307.

Scheiter, Simon, Liam Langan, and Steven I. Higgins. "Next-generation dynamic global vegetation models: learning from community ecology." *New Phytologist* 198.3 (2013): 957-969.

Sakschewski, Boris, et al. "Resilience of Amazon forests emerges from plant trait diversity." *Nature Climate Change* 6.11 (2016): 1032-1036.

Seidl, Rupert, et al. "An individual-based process model to simulate landscape-scale forest ecosystem dynamics." *Ecological Modelling* 231 (2012): 87-100.

Rödig, Edna, et al. "Spatial heterogeneity of biomass and forest structure of the Amazon rain forest: Linking remote sensing, forest modelling and field inventory." *Global Ecology and Biogeography* (2017).

Levy, P. E., and P. G. Jarvis. "Direct and indirect measurements of LAI in millet and fallow vegetation in HAPEX-Sahel." *Agricultural and Forest Meteorology* 97.3 (1999): 199-212.

---

## Author Response (AR1)

Compared to the announced text changes in our reply in the interactive discussion we slightly changed several text fragments to improve readability, English and grammar (especially in the three new sections: 4.2 Implications for forest management, 4.3 Implications for global vegetation modelling and the new Conclusion).. We mark major changes in our reply in purple.

**Comments of reviewer 1 in blue, reply in black.**

*I am sorry for being late. The manuscript uses an innovative modelling approach to assess the temperature sensitivity of above ground wood production. It is generally well written but there are quite a number of minor fixed that still need to be carried out.*

Thank you very much for your detailed and helpful comments. We will include your recommendations carefully.

*I have two major concerns that can be addressed in a thorough minor revision: 1) I think your "climate scenarios" are actually rather a "climate sensitivity test". Even though many things can be scenarios in the wider sense in the narrow sense, a scenario usually refers to an internally consistent projection while your scenarios systematically explore change temperature but do not adjust precip and radiation accordingly. This may lead to physically inconsistent "climates" because under a given temperature pathway it might be impossible to get a certain precip or radiation behavior.*

Thanks for this comment. We agree. We will replace "climate scenarios" by "climate sensitivity test" or "climate time series" depending on the context. For instance, P1L9: For each stand we estimate annual above-ground wood production and perform a climate sensitivity analysis analysis based on 320 different climate time series (of one year length).

*Related to that, you should possibly also discuss that your approach basically ignores transient responses and time lags longer than a year that influence forests. So i think this "sensitivity test" aspect should be carved out more clearly. The title actually is fine but some of the other sections give the impression this is rather a scenario study.*

This is a good point. Thank you. We will replace the sentences between P15L12 and P15L15 with the following text:

…This RCP scenario predicts only small changes of annual precipitation levels for many temperate regions. Hence, our approach focuses only on the effect of temperature change on wood production. For instance, we do not simulate the effect of changing $CO_2$ in combination with an increase of temperature. This might be critical for analysis of strong temperature changes (e.g. RCP 8.5), which will result in an increase of droughts and changes in the annual temperature cycles. Such more complex scenarios should be analyzed in future studies. Further, we neglect the effect of time lags (e.g. the bud building in the autumn in the previous year). However, it is possible to enlarge the used time series to analyze the behavior of the forest over longer time periods and study not only productivity, but also effects on regeneration or mortality.

*2) I think, even though referring to central European, temperate forests, you are not putting enough emphasis on the interpretation of the results from a forest management perspective. However, your target variable "above ground wood production" as well as the stand densities, species choices etc. are subject to forest management and species mixing etc. are important*

*elements of EU silvicultural strategies. I think the discussion of the influence of forest management on your results and their implications for forest management should be strengthened.*

We agree that our result might be important for forest management strategies. We therefore will add a new section "4.2. Implications for forest management" in the Discussion section
* * *
Thank you for these detailed smaller points.

*P1L1: Either "Observational studies discovered that" or "Observations show that..."*

*P1L4: "increase productivity the most"*

*P1L8: "cover a wide range of possible"*

We will reformulate the phrases.

*P1L6: unclear: "within the forest structure" ==> don't you simply mean "within the forest given the environmental conditions of each..."?*

We will write: … how well species are distributed over the different forest layers regarding…

*P1L11 and also L14: "increasing OmegaAWP" it sounds weird that an optimum can be increased or have "large values".  Maybe I am too picky and I do not have a good alternative... maybe ask a native speaker...*

We will introduce a new term for $\Omega_{AWP}$:  "species distribution index" but keep the acronym $\Omega_{AWP}$.

*P1L15: "heterogenity is associated with a positive"*

We will reformulate the sentence, following your suggestion.

*P1L14-16: This sounds like quite a contradiction: for young forests low diversity and low height spread make the forest react positively to temperature while for older forests this is not the case.  Could you add one sentence of explanation here and discuss potential implications for forest management in the discussion?  This would mean foresters should go for even aged, mono-species stands during establishment and then bring in other species later?  or keep the canopy closed with one species while having other species in the undergrowth for some time?*

We recommend paying attention to young forests with low diversity and even-aged structure. A mixture of climax species should be planted below the main canopy. Later those climax species will replace the pioneer species in the canopy and build the mature forest with heterogeneous tree sizes.

We will add a new section regarding the implications on forest management (4.2. Implications for forest management).

*P1L19: I have the feeling you are using "forest growth", "wood production" and "forest productivity" interchangeably. While this can be correct in some instances I wonder whether all the references you cite here actually refer to forest growth or rather productivity.}*

We will focus on (above-ground) wood production in the whole manuscript. Barber et al. 2000 analyzed the change in radial growth of white spruce under climate change. We will replace the references of Cao and Woodard 1998 (as it refers more to carbon fluxes of ecosystem in general and not explicit wood production) by Reyer et al. 2013.

The other two refer to the effect of climate change on photosynthesis or respiration of trees (Luo 2007) or plants (Penuelas and Filella 2009).

*P1L21-P2L2: I think this sentence is imprecise. the observed changes in productivity are not the primary reason for discussing the compensation of co2. it is rather the overall high carbon stocks and sequestration rates (even without changes) that matter for this discussion. I see what you want to say but I think it is a bit too condensed here....*

We will reformulate the sentences:

Changes in forest productivity have been observed in past decades all over the world (Nemani et al., 2003; Boisvenue and Running, 2006; Seddon et al., 2016). The carbon stock of forests and their role as carbon sink are therefore changing. These findings stimulated discussions about whether forest management strategies can be adapted to reduce forest vulnerability to climate change, to support recovery after extreme events and foster the carbon sink function of forests.

*P2L3-9: I think this section should also mention the influence of other factors, at least briefly... Especially since you say that "productivity is influenced by several factors" in the first sentence of this paragraph...*

We will add the following sentences: … in addition to other climate variables (Barford et al. 2001). For instance, increasing $CO_2$ increases water use efficiency of forests (Keenan et al. 2013), which could compensate negative effects of climate change on European forest growth  – (Reyer et al. 2014). Another important often investigated process is the fertilization effect of nitrogen (De Vries et al., 2006, 2009). For instance due to depositions in the second half of the last century, wood production had increased in European forests (Solberg et al. 2009). In case of temperature change, photosynthesis, respiration and growth rates are modified…

*P2L18/19: "rarely include properties related to both species composition . . .*

*P2L22: "forests stands were available, it would"*

*P2L23.: "option to such field experiments is"*

*P2L26: "simulating 30 year time slices of a range of different future climates for 135..."*

*P2L27: "analyzed"*

*P2L29: "a large number"*

*P2L30: "species compositions"*

*Figure 1: I would delete the "..." in each box as your study does not cover more climate variables nor more stand structural or composition related variables. In the caption, I would precise: "overview of drivers influencing forest forest productivity in this study" and also clearly state that only temperature is varied in a "temperature sensitivity analysis" or so.*

*P3L3: "2017). The forest factory generates 370,170...and allows to estimate"*

We will realize all these suggestions. Thank you.

*P4L2: How to get from the 15 stem size distributions and 256 mixtures to the 370,170 stands?*

The forest factory creates forest patches which are based on 15 different stem size distributions and 256 species mixtures. 100 forests patches of each combination are built (in total 15*256*100 = 384,000 forests). In a few cases not all species of the mixture could be placed within a patch by the algorithm, so these forests are rejected. We end up with 370,170 forest stands.

We will reformulate section 2.2 of the Method section to clarify these points.

*P4L3/4: At some point you should give the latin names of the species to allow international readers to check which species you mean...*

Good point. We will add the Latin names in the section 2.2.

*P4L6: I wonder how you can actually represent complex mixtures on a 400m$_2$ plot. This could be covered by one large beech tree? I think you need to discuss the implications of choosing this patch size. Or do you upscale to the ha or so?*

Within an area of 400m² the trees compete for light as large trees shade the smaller ones. This is the typical plot size used in forest gap models. You are right, a very large tree could cover such a plot but below its crown there is space for smaller trees of other species. However, there are indeed a few combinations of species mixtures and stem size distribution that could not be represented within a plot because of limited space. Thus, these forests do not exist in the collection of forests generated by the forest factory.

We will reformulate the text in section 2.2. to clarifying these points.

*P4L6: "space limits"*

We will reformulate the text.

*P4L8: You should discuss in detail why you think the year 2007 in Hainich is representtative of temperate climatic conditions! I think you make two dangerous assumptions: 1) Hainich is somehow representative for "temperate climates" (it certainly is but only to a certain degree and 2) the 2007 climate is somehow representative of the overall Hainich climate*

We agree. We use this year as an example of a temperate climate. (In principle, it is possible to use climate data of every other location). We will reformulate the text in section 2.2.

*P4L11-22: This is from FORMIND, right? You could say that*

We will add the following sentence at the beginning of the paragraph: The calculation of wood production of trees is based on algorithms of the model FORMIND (Fischer et al. 2016,).

*P4L10: "2.2 Forest productivity . . ."*

*P4L15: "by the photosynthesis-limiting . . ."*

We will use your suggestions.

*P4L17/18: I read this as if Rm was both maintenance respiration and allocation to non-woody tissues?*

This is correct.

*P4L23: I would introduce a new subheading here about the "climate sensitivity"*

We will add it.

*P4L25: "separated" The methods description should be in the past tense*

We will change the tense.

\emph{\color{blue}*Figure2: You should explain once more all the variables shown on the plots in the caption, AWP and MAT are not explained currently.*

Good point. We will modify the caption of Figure 2.

*P5L1: unclear: do you include co2 and nitrogen in the model but keep them constant or are they not included at all? You should discuss that co2 will matter in a 2 $^{\circ}C$ warmer world . . .*

They are not included at all. We will modify the sentence. See also the reply to your first major comment above.

*P6L6: Do you have any reference or argumentation to support using BA as a proxy for LAI? I could imagine this only works until canopy closure?*

In the forest factory dataset LAI and basal area correlate quite well ($R^2$=0.74). A high correlation between leaf area and basal area ($R^2$>0.92) has been found for instance also by Levi and Jarvis 1999.

*P6L13: "maximum forest height"*

We will replace the phrase.

*P7L1-2: Is "10.2015" the right citation format?*

No. We will correct the citation format to "2015".

*P7L5: Why Gaussian? Any deeper reasons or simply because it is the default?*

It is the default setting. We will modify the sentence: … assuming a Gaussian error structure (default setting).

*P7L8: the heading is unclear. "benefit the most" ==>from what?*

We will add: … the most from increasing temperatures.

*P7L18-19: I wonder if the mean is the appropriate measure here given that the distributions are so skewed (figure b1)?*

Figure B1 shows the distribution of the SI values. The shown SI values do not represent mean values; they are the slope of linear models, which relate temperature changes with AWP changes divided by the average AWP of the forest (see equation 2, 3 & 4).

We will modify the sentence: We than quantified the changes in productivity due to changes in MAT ($SI_{MAT}$) and Q95 ($SI_{Q95}$).

*P7L28: This has to be carefully discussed. It seems to be obvious that the species choice will have the strongest influence.*

Thanks for this point. We will modify the sentences at P11 L16ff: … If species are unfavourably distributed within the forest (low $\Omega_{AWP}$), the AWP of the forest is low. Note, $\Omega_{AWP}$ is the ratio between current AWP and the highest possible AWP of the forest which can be reached due to shuffling of species identities. If AWP is low the forest will suffer from increasing temperatures, which results in negative slopes ($\Delta$AWP / $\Delta$T). These values are than divided by low AWP values (Equation 4), which results in large negative values of $SI_{MAT}$ and $SI_{Q95}$. (See Appendix B5).

*Figure 3: maybe explain somewhere (can be in the main text) how to interpret the scale from 0-1 of the Omega$_{AWP}$.*

We will add an explanation in the caption of figure 3.

*P8L12: "analysed how"*

We will reformulate the phrases.

*P8L12: Maybe recap here that AWP is your expression of productivity.*

We will replace forest productivity by (above-ground) wood production. See also reply to your comment P1 L19.

*P9L1: "specific value combination of forest properties" ==> rephrase*

We will write: …with a specific set of forest properties which…

*P9L5/6 & 12-14: Certainly analyzing all possible combinations of species and structures etc as done in the Forest Factory approach is valuable but this approach will also generate a huge number of stands which are highly unrealistic and that will never be found in reality. So the discussion could be more balanced here highlighting that you also produce quite a lot of "non-sense" forests as well.*

This is an important point of the forest factory approach.

All forest patches could exist in reality, as every tree has a positive productivity and enough space for its crown. It is not possible that "non-sense" forests are generated as the used method prevents that a light demanding species occurs below a closed canopy or that forests are overcrowded. We will revise the text to makes this clearer.

*P9L10: "FORMIND"*

We will replace the phrase.

*P10L2-3: This sentence needs to be rewritten for clarity*

We will remove this sentence.

*P10L11: RCP2.6*

We will correct this.

*P10L15-28: I had the feeling this section needs to be rewritten for clarity and logical connection to the preceding sections.*

Thanks for this comment. We will reformulate the section in the following way:

To characterize the annual cycles of temperature we select two variables: mean annual temperature (MAT) and inter annual temperature amplitude (Q95). Both variables can be varied independently. In case of higher MAT we observe an elongation of the vegetation period. This leads to higher forest productivity, if other resources are sufficiently available (Luo, 2007). This explains why $SI_{MAT}$ is often positive. However, warmer summer temperatures can also lead to a decline in wood production due to an increase in respiration. In case of increasing Q95, more days with extreme temperatures will occur in a year. Thus, an increase of one $^{\circ}C^{-1}$ of Q95 will increase respiration more strongly compared to an increase of one $^{\circ} C^{-1}$ of MAT. Hence, the increase of Q95 has normally negative effects on the productivity (negative SI values).

*P11L3-12: Also here maybe some rewriting is needed to better link the paragraph to the rest of the discussion.*

We will modify the two paragraphs of section 4.2.

*P13L5: I find the conclusions too short and too close to the results. The conclusion in my view should clarify: Why do your results matter? What do we learn?*

We will extent the conclusion and include implications of our study for future forest modelling and forest management.

*P13L12: I find the supporting material organized in a complicated way. Can you not provide the text plus the associated figures and then another piece of text etc. The online material is not meant to be read as one text but one should find things quickly.*

We will rearrange the text and figures as suggested.

*Figure A2: I am pretty sure you are showing the annual precipitation sum and not the ''mean'' here.*

Correct. We will revise the sentence.

*P15L4/5: I do not understand this: it seems mean (1.5%) is outside of the interquartile range?*

The unit is \%${\circ}C{-1}$ . This means that on average forest increase their productivity by one \% if temperature increases by ${\circ}C$ . The interquantile range of 95\% (Q95) is used only to quantify the interannual temperature variability.

We will modify the sentences to make this clearer:

*Figure A3: hec should be ha}*

*P16l5: "the first plot" ==>not very precise. Do you mean plot 3a)? Then also include small letters in the plots!*

We will follow your suggestions.

*P16L6: I would always refer to $\Omega_{AWP}$ and not introduce any other terms such as suitability etc. it is getting too complicated . . .*

We will modify text and figure.

*Figure A4: To me it looks like You are overestimating $SI_{MAT}$ quite systematically but you never really discuss this?*

Good point. It is quite difficult to evaluate the theoretical analysis by using field data, as a huge number of similar forest plots are needed, which cover a temperature gradient. In case of the German forest inventory only a few forest plots can be used for such an analysis. For spruce and beach monocultures we were able to select enough similar plots. Further, these inventory plots provide no direct measurements of climate or LAI, which would be needed for an appropriate evaluation. We therefore used altitude as proxy for temperature (although other environmental variables like precipitation or soil attributes could change as well with altitude) and basal area as proxy for LAI. The reason for the overestimates of $SI_{MAT}$ is not clear, but might be a result of the differences between $SI_{MAT}$-calculations based on field data and the calculation of $SI_{MAT}$ based on the forest factory approach.

We will add some sentences regarding this point to the Appendix.

 We will revise the sentence.

*Figure b3: I wonder why there are so distinct patterns of values with y=0 and also x=-100 in panel a and y=-8 (or so) and x=-100. Is this an artifact?*

The vertical dots occur as all forests with SI values smaller than -100\% ${\circ}C{^{-1}}$ where set to -100\% ${\circ}C{^{-1}}$ for this graphic. To be consistent, we will remove them, also as they were removed for the final analysis, which is presented in the paper. See page 6, line 4 of the manuscript: …*In our analysis we exclude all forests stands for which AWP is negative if the temperature rises by 1* ${\circ}C{-1}$ *( 2\% of all stands).*

I assume the horizontal structure is an artifact of the boosted regression tree algorithm. We will reformulate the corresponding sentences.

We will reformulate these phrases and add the "b)".

*Figure B6: The layout of this figure makes it very hard to see which color overlays the other . . . so do most "brownish" species rather follow the green line or the dark blue for the left-side of the bell-shaped lines in panel a)?*

We will revise the figure based on a modified color-palette and enlarge the graphic.

*Figure B7: Is nowhere referred to in the text. What is the unit of the y-axis?*

It will be referred in the text and the text will be reformulated and we will revise the figure.

*Figure B8: The $SI_{MAT}$ values are so small, is that correct?*

We forgot conversion into \% (multiplication by 100, as done in all the other graphics). We will also add the unit to the y-axes.

**Reviewer 2 (anonymus)**

Compared to the announced text changes in our reply in the interactive discussion we slightly changed several text fragments to improve readability, English and grammar (especially in the three new sections: 4.2 Implications for forest management, 4.3 Implications for global vegetation modelling and the new Conclusion). We mark major changes in our reply in purple.

*The authors use an individual-based forest modelling approach to isolate the effect of five forest structure (LAI, maximum stand height, and canopy stratification) and composition (functional diversity and its optimal distribution) parameters on the overall sensitivity of simulated wood production to increasing mean annual temperature and seasonal amplitude in a European temperate setting. The model is integrated hundred of times over a single year and forest stand using synthetic climate scenarios with perturbed temperature to simulate an ensemble of productivities for thousands of stands with a range of initial conditions of structure and composition, representative of different stand development stages. This synthetic dataset, broadly consistent with ecological rules observed in the real world, is then analysed using a statistical regression method in order to quantify the relative effect of the five structural and compositional parameters on the simulated temperature sensitivity of forest productivity. Their results indicate that an index of optimal species distribution of the trees in a forest stand – the ratio of actual productivity of a forest stand to the maximum productivity achieved by changing only the species of the trees whilst keeping the same stand structure – explains ? 88% of the temperature sensitivity of forest productivity simulated by the model.*

*Among the remaining four parameters, forest height (a proxy of stand development stage) is the most important variable in explaining the rest of the fractional variance of temperature sensitivities. Thus the authors conclude that the sensitivity of plant productivity in temperate forests to changes in temperature is driven by forest structure and species diversity. I believe that the main scientific finding is of interest for the wider biogeosciences communities.*

*The overall modelling experiment seems appropriated to disentangle the relative importance of forest structure and composition properties on patterns of temperature sensitivity of temperate forest productivity in the model world. However, the manuscript is very difficult to follow at times and the discussion of the main findings is simply too thin. I recommend major revisions in order to improve (i) the readability and English of the manuscript and (ii) to better discuss the underlying mechanisms and implications of the findings for the wider ecological theory.*

Thank you very much for your review. We will revise our manuscript carefully including your recommendations.

We will enhance the readability and we will send the revised manuscript to native speakers to improve the English and the grammar.

We further will add two new sections discussing the implications of our results for forest management and global vegetation modelling (see new section 4.2 and 4.3) and enlarge the conclusion (see appendix R3). Finally, we will carefully revise the Discussion including more details of the underlying mechanisms (Please note that some sections of the former discussion are now in section 3.1).

*SPECIFIC COMMENTS*

*Regarding the first point above, I have the following comments:*

*-Introduction*

*It is informative but the English and grammar need revision.*

As explained above, we will revise the manuscript regarding English and grammar using the help of a native speaker.

*-Method*

*This section tends to be redundant with the opening paragraph and appendices. I recommend reorganizing it and make it more concise. It is not clear how the stands at different stages of development were initialised. Were the structure, composition and development stage randomly generated or did you apply some spin-up? This has implications for the realised productivity when computing the $?\_AWP$ index. Why exactly 370,170 stands?*

We did not perform any spin-off. Trees are "planted" with a certain size derived from the stem size distribution considering that each tree of a forest has a positive productivity and enough space for its crown. The forest factory creates forests, which are based on 15 different stem size distributions and 256 species mixtures. 100 forest patches with a size of 400m² of each combination were built (in total 384,000). To generate these 100 forests of each combination, we randomly selected a tree from a stem size distribution, assigned a species and checked if the tree has enough space and a positive productivity. In a few cases not all species of the mixture could be placed within a forest by the algorithm, so we rejected such forests. We end up with 370,170 forest stands. We will reformulate the text in 2.2 of the Methods section.

*In Line 28 of page 4 it is said "We end up with five climate scenario sets of one-year length that differ in precipitation and radiation." But since these scenarios are derived from five different real years, they should also differ in absolute temperature values? As I understand it, within each synthetic scenario, radiation and precipitation are the same and only temperature changes at the specified steps. Among the five scenarios, the absolute values of all variables should be different. Please explain better this part to the reader.*

Thanks for mentioning this point. The absolute temperature values differ between the 5 time series. We will explain this point in more detail.

*In Fig. 2, what does the shading in the middle panels mean? The meaning of H_forest is not explained in the caption.*

Thanks for mentioning this point. The shading represents the standard deviation obtained from the analysis of five different time series (with the same modification of MAT and Q95). We will rewrite the text and explain $H_{forest}$.

*-Results and Discussion*

*I found many passages in these sections very difficult to follow. There are result statements with no reference to the figures or tables that leave the reader guessing the corresponding figures. The results section is rather short and most of*

We will move figure 5 und 6 (and the corresponding text) from the Discussion into the Result section. We further will expand the Discussion by two new sections, which will discuss the implications for forest management and vegetation modelling (see 4.2 Implications for forest management and 4.3 Implications for global vegetation modelling). Finally we will carefully revise the results and discussion to improve the text and link the statements with the corresponding figures.

*Figure B1 shows a long negative tail in the distribution of the obtained sensitivities. The authors focus exclusively on the positive sensitivities and neglect negative values, despite declaring in the introduction that responses can be both positive and negative. Why the simulated sensitivities are so asymmetrical and the negative values are not discussed?*

We will explain the large negative values and the long negative tail of the sensitivity values in more detail. However, we don´t know the reason for the asymmetric distribution. In addition, we will modify figure 5 and replace forest C by an old-growth forest with negative SI-values.

*In the discussion I miss a more complete explanation of the underlying ecophysiological and metabolic mechanisms (e.g., Figs 5 and 6). Also, there is no discussion of the potential limitations of the model and the modelling experiments performed here. For instance, autotrophic respiration seems to be a critical factor affecting the response of net productivity to changes in temperature in the model. How well is this process represented in the model version used? Would you get the same result if you account for uncertainties in parameterisation of this process?*

Thanks for mentioning these points. The forest factory is based on the well-established and often applied forest gap model FORMIND. The parametrization was used and discussed in several previous studies (Bohn et al 2014, Bohn et al 2017, Rödig et al. 2017). We will make this clearer in the Method section. We further will enlarge the section "4.1. The study-design" and enlarge the explanations in the Appendix discussing photosynthesis and respiration rates of single trees (Figure B7). In general, the parameters with the highest uncertainty within the FORMIND-model are establishment and mortality, which are not used in this study.

*Finally, what are the implications of the main finding for the wider ecosystem and climate modelling communities that usually rely on global models that have no explicit forest structure? Any recommendation?*

This is a good point. Thank you. We will add a new section, which discusses the implication of our analysis on global vegetation modelling (see 4.3 Implications for global vegetation modelling).

*-Conclusion Rather brief. Here the authors could wrap-up the wider implications of their main findings.*

We will extend the conclusions to include implications of our study for future forest modelling and forest management. (See also  new conclusion).

*-Figures*

*The figures are excellent but the captions are not sufficiently informative. Please improve the captions.*

We will revise the captions.

*Fig 3 could be merged with Fig B2. The latter is important to understand the overall result.*

We will do this.

*-English and grammar.*

*There are many grammar errors and typos through the text. Please revise English and correct typos.*

We will send the revised manuscript to an language expert to improve the English and grammar.

**List of major changes**

- Revised section 2.2 forest factory approach
- Revision of most captions of the figures
- Revision of former section 4.2 The influence of forest structure on temperature sensitivity - now section 3.1.1.
- Revision of former section 4.3 The effect of species composition on temperature sensitivity - now 3.1.2
- Extention of the discussion of the study design (section 4.1)
- Two new sections 4.2 Implications for forest management and 4.3 Implications for global vegetation modelling
- Revision of the conclusion.

[revised manuscript text omitted]

**2   Method**

To analyse the effect of temperature on the productivity of forest stands, we apply the "forest factory" model approach (Bohn and Huth, 2017). [c11]The forest factory generates 370,170 different forest stands (see section 2.1) and [c12]allows the estimat[c13]ion of above ground wood production (AWP) under various climate [c14]time series (see section 2.2). The 320 scenarios differ
* * *
[c3]

[c4]

[c5] *Text added.*

[c6]

[c1]

[c2]

[c3]

[c4]

[c5]

[c6]

[c7] *Text added.*

[c8]

[c9]

[c10] *Text added.*

[c11]

[c12] *Text added.*

[c13]

[c14]

in mean annual temperature (MAT) and annual temperature amplitude (Q95). Finally, we calculate the forest stand-specific sensitivity of productivity against temperature change ($SI_{MAT}$ and $SI_{Q95}$) as the relative change of [c15]wood production per temperature change of $1\,°C$ (see section 2.2). To relate these sensitivities to forest structure and species composition, we characterize every forest stand with five properties (see section 2.4). We analyse the influence of the five forest properties on temperature sensitivity using boosted regression trees (see section 2.5). Finally, we analyse which combination of forest properties results in the highest sensitivity values for different successional stages (see section 2.6).

**2.1 The forest factory approach**

The forest factory [c1] creates forest patches which is based on 15 different stem size distributions [c2]and 256 species mixtures.[c3]100 forests patches of each combination are built.

[revised manuscript text omitted]

   [c3]In [c4]even-aged forest[c5]s [c6], all trees have the same height and receive full light (e.g., [c7] Fig. 5[c8], forest C). Such forests
10  show a bell-shaped relationship between forest height and temperature sensitivity (Fig. 5 SI values for 100% available light depending on tree height). [c9]

   [c13]In case of a forest consisting of trees with different heights [c14]smaller trees receive less light due to shading. Note that, even if trees reveive less light, the bell-shaped relationship between tree height and productivity persists (Fig. 5). Two cases will be discussed [c15](assuming identical LAI as forest C, Fig. 5 [c16]). [c17]In the first case all trees have not yet reached their
15  maximal SI-values (Fig. 5, forest A,); and [c18]in the second case all trees [c19]already passed their maximal SI-values (Fig. 5, forest B). In the case of forest A, trees in the shade of larger trees always have lower SI-values if they belong to the same species (see Appendix B4). Hence, the temperature sensitivity level of this forest is lower than the sensitivity of an even-aged forest, whose trees have the same size as the largest tree in forest A (Fig. 5,[c20] tree 1). Hence, if maximal SI-Values are not reached, increasing height heterogeneity decreases SI-values of a forest.
* * *
[c1]
[c2] *Text added.*
[c3]
[c4] *Text added.*
[c5] *Text added.*
[c6]
[c7]
[c8] *Text added.*
[c9]
[c13]
[c14]
[c15] *Text added.*
[c16] *Text added.*
[c17]
[c18]
[c19]
[c20] *Text added.*

[Figure]

**Figure 5.** Analysis of $SI_{MAT}$ values of single trees within three different forests. The diagram shows the calculated $SI_{MAT}$ value of individual trees for every combination of tree height and available light [c10](for pinus sylvestris between $SI_{MAT}$ [c11]-levels of 6.5 and - 6.5; other species show similar patterns). The dots indicate the different trees of the three forest examples [c12]The white dots belong to trees with the corresponding number of forest A, gray dots belong to the trees of forest B and dark gray dots belong to forest C. Note that in the case of forest C, all trees have the same height and the same light, so that all three dots are at the same place in the diagram.

In forest B [c21](Fig. 5), SI-values of the shaded trees can be similar (or even higher) than the SI-value of the largest trees in the forest [c22](SI-values of tree 1 show similar levels as tree 2, 3 and 4 in forest B, Fig. 5). [c23]Hence, if maximal SI-values are passed, increasing tree height heterogeneity results in similar [c24] (or even [c25]more positive) temperature sensitivity levels compared to an even-aged forest [c26] trees [c27](an even-aged forest consisting only of trees similar to tree 1 of forest B in Fig. 5).

5     These general considerations explain the change from low levels of height heterogeneity in young forests to a more heterogeneous structure [c1]in the analysis of those forests, which will benefit from increasing temperature (see Fig. 4 d).

**3.1.2    The effect of species composition on temperature sensitivity**

In this study, we use the new index $\Omega_{AWP}$ called [c2]species distribution index (Bohn and Huth, 2017). $\Omega_{AWP}$ [c3] is the ratio between current AWP and the highest possible AWP of the forest which can be reached due to shuffling of species identi-

10 ties. Its huge importance on forest temperature sensitivity might be illustrated by the following considerations: If species are
* * *
[c21] *Text added.*

[c22] *Text added.*

[c23]

[c24]

[c25]

[c26]

[c27] *Text added.*

[c1]

[c2]

[c3]

[Figure]

**Figure 6.** Graphic (a) shows which species [c1]have the highest productivity ( $\Omega_{AWP}$ [c2] 
[revised manuscript text omitted]

* * *
[c1]
[c2]
[c3]
[c4]
[c5]
[c6]
[c7]
[c8]
[c9]
[c10]
[c11]
[c12]
[c13] *Text added.*

such a development, one could plant below the closed canopy of even-aged, pioneer trees a climx-species-rich understory that will build the the canopy of the mature forest. This study highlights that forest structure and species composition are both relevant [c14]in understanding the temperature sensitivity of [c15]wood production.

*Copyright statement.* TEXT

**1 Introduction**

Climate change alters [c1]wood production by modifying [c2]the rates of photosynthesis and respiration rates of trees (Barber et al., 2000; Luo, 2007; Peñuelas and Filella, 2009; Reyer et al., 2014). Changes in forest productivity have been observed in past decades all over the world (Nemani et al., 2003; Boisvenue and Running, 2006; Seddon et al., 2016). [c3]The carbon stock of forests and their role as carbon sink are therefore changing. These [c4]findings have stimulated discussions about whether forest

10 management strategies can be adapted to reduce forest vulnerability to climate change, to support recovery after extreme events and [c5]foster the carbon sink function of forests (Spittlehouse and Stewart, 2004; Spittlehouse, 2005; Bonan, 2008).

[c6]Wood production is influenced by several factors [c7], such as $CO_2$ fertilization, nitrogen deposition, precipitation, and temperature. (Barford et al., 2001)[c8]. [c9] For instance, rising CO2 increases water use efficiency of forests (Keenan et al., 2013) [c10], which could compensate negative effects of climate change on European forest growth (whereas, with constant $CO_2$ at

15 350ppm, forest growth declines on several sites due to climate change - see Reyer et al. 2014). Another important process is the fertilization (De Vries et al., 2006, 2009)[c11]. Due to depositions of nitrogen in the second half of the last century wood production had increased in European forests (Solberg et al., 2009). However, temperature modifies photosynthesis, respiration and growth rates of trees (Dillon et al., 2010; Piao et al., 2010; Wang et al., 2011; Jeong et al., 2011; Heskel et al., 2016). In the temperate biome, positive effects on [c12]wood production (Bontemps et al., 2010; Delpierre et al., 2009; Pan et al., 2013;

20 McMahon et al., 2010, e.g.) as well as negative ones have been found (Barber et al., 2000; Jump et al., 2006; Charru et al., 2010, e.g.). However, it remains unclear why forests react differently to temperature change.
* * *
[c14]
[c15]
[c1]
[c2] *Text added.*
[c3] *Text added.*
[c4]
[c5]
[c6]
[c7]
[c8]
[c9] *Text added.*
[c10] *Text added.*
[c11] *Text added.*
[c12]

[revised manuscript text omitted]

    [c3]In [c4]even-aged forest[c5]s [c6], all trees have the same height and receive full light (e.g., [c7] Fig. 5[c8], forest C). Such forests

10   show a bell-shaped relationship between forest height and temperature sensitivity (Fig. 5 SI values for 100% available light depending on tree height). [c9]

    [c13]In case of a forest consisting of trees with different heights [c14]smaller trees receive less light due to shading. Note that, even if trees reveive less light, the bell-shaped relationship between tree height and productivity persists (Fig. 5). Two cases will be discussed [c15](assuming identical LAI as forest C, Fig. 5 [c16]). [c17]In the first case all trees have not yet reached their

15   maximal SI-values (Fig. 5, forest A,); and [c18]in the second case all trees [c19]already passed their maximal SI-values (Fig. 5, forest B). In the case of forest A, trees in the shade of larger trees always have lower SI-values if they belong to the same species (see Appendix B4). Hence, the temperature sensitivity level of this forest is lower than the sensitivity of an even-aged forest, whose trees have the same size as the largest tree in forest A (Fig. 5,[c20] tree 1). Hence, if maximal SI-Values are not reached, increasing height heterogeneity decreases SI-values of a forest.
* * *
[c1]
[c2] *Text added.*
[c3]
[c4] *Text added.*
[c5] *Text added.*
[c6]
[c7]
[c8] *Text added.*
[c9]
[c13]
[c14]
[c15] *Text added.*
[c16] *Text added.*
[c17]
[c18]
[c19]
[c20] *Text added.*

[Figure]

**Figure 5.** Analysis of $SI_{MAT}$ values of single trees within three different forests. The diagram shows the calculated $SI_{MAT}$ value of individual trees for every combination of tree height and available light [c10](for pinus sylvestris between $SI_{MAT}$ [c11]-levels of 6.5 and - 6.5; other species show similar patterns). The dots indicate the different trees of the three forest examples [c12]The white dots belong to trees with the corresponding number of forest A, gray dots belong to the trees of forest B and dark gray dots belong to forest C. Note that in the case of forest C, all trees have the same height and the same light, so that all three dots are at the same place in the diagram.

In forest B [c21](Fig. 5), SI-values of the shaded trees can be similar (or even higher) than the SI-value of the largest trees in the forest [c22](SI-values of tree 1 show similar levels as tree 2, 3 and 4 in forest B, Fig. 5). [c23]Hence, if maximal SI-values are passed, increasing tree height heterogeneity results in similar [c24] (or even [c25]more positive) temperature sensitivity levels compared to an even-aged forest [c26] trees [c27](an even-aged forest consisting only of trees similar to tree 1 of forest B in Fig. 5).

5    These general considerations explain the change from low levels of height heterogeneity in young forests to a more heterogeneous structure [c1]in the analysis of those forests, which will benefit from increasing temperature (see Fig. 4 d).

**3.1.2   The effect of species composition on temperature sensitivity**

In this study, we use the new index $\Omega_{AWP}$ called [c2]species distribution index (Bohn and Huth, 2017). $\Omega_{AWP}$ [c3] is the ratio between current AWP and the highest possible AWP of the forest which can be reached due to shuffling of species identi-

10    ties. Its huge importance on forest temperature sensitivity might be illustrated by the following considerations: If species are
* * *
[c21] *Text added.*

[c22] *Text added.*

[c23]

[c24]

[c25]

[c26]

[c27] *Text added.*

[c1]

[c2]

[c3]

[Figure]

**Figure 6.** Graphic (a) shows which species [c1]have the highest productivity ( $\Omega_{AWP}$ [c2] 
[revised manuscript text omitted]

---

## Author Response (AR2)

**Associate Editor Decision: Publish subject to minor revisions (review by editor)** (19 Jan 2018) by Sebastiaan Luyssaert

Comments to the Author:

The authors properly addressed most of the concerns raised by the reviewers. One concern needs, however, more attention. Although the authors state in their cover letter that "We will send the revised manuscript to an language expert to improve the English and grammar", I'm unsure this has happened. If it happened, the language needs to be checked again. There are still too many typos and inconsistencies which hampers the readability of the manuscript. The tenses of the verbs should follow the conventions in scientific writing. On page P7 the present tense is used "do" on P8, L1 the past tense is used. P8, L2 another present tense is used. In the methods section it is customary to use a form of the simple past tense to describe what you did in your study. Passive voice is preferred. If you are unsure search the web for "Using tenses in scientific writing".

Thank you for this comment. We sent the manuscript to a professional language expert and checked the tenses of verbs.

Install a spelling checker for Latex. Just on the first pages I spotted the following issues: P1, L16 replace "archieve" by "achieve"; P2, L1 replace "climx" by "climax"; P2, L2 delete one "the"; P2, L13 add subscript to CO2; P2, L15 add \, in Latex between 350 and ppm. Apply for all numbers followed by units.; P5, L16 replace "available enough space" by "enough space available"; P5, L19 replace "every" by "any"; P7, caption replace "with in" by "within".

We corrected these issues and sent the manuscript to a professional language expert.

There is no space limit in Biogeosciences. So, there is no need to use acronyms. Writing in full as much as possible will largely increase the readability of the manuscript.

Thank you for this comment. We omit most acronyms and keep only AWP and $\Omega_{AWP}$ and $SI_{MAT}$ & $SI_{Q95.}$

P15 Delete L10-14. The discussion should not start with a summary of the methods.

Thanks for this comment. We deleted this section

P15, L25-26 apply correct in-text citation format. Unless the author is the subject or object of the sentence the name and year should be between brackets. There is a difference between \cite and \citep in the Latex template for Biogeosciences.

We corrected this mistake.

P23, why was A4 included in the Appendix rather than the main text?

Thank you for this point. We shifted the figure and corresponding text to the Method section.

The text in the appendix as well as the figures need to be carefully checked and edited. Fig B5 remove h[!h!b]; Fig B6 check the units of the y-axis;

Thank you. We revised the text and the figures of the Appendix